# Structural basis of substrate recognition and allosteric activation of the proapoptotic mitochondrial HtrA2 protease

Emelie E. Aspholm[1,2], Jens Lidman[1,2] & Björn M. Burmann [1,2] ✉

The mitochondrial serine protease HtrA2 is a human homolog of the *Escherichia coli* Deg-proteins exhibiting chaperone and proteolytic roles. HtrA2 is involved in both apoptotic regulation via its ability to degrade inhibitor-of-apoptosis proteins (IAPs), as well as in cellular maintenance as part of the cellular protein quality control machinery, by preventing the possible toxic accumulation of aggregated proteins. In this study, we use advanced solution NMR spectroscopy methods combined with biophysical characterization and biochemical assays to elucidate the crucial role of the substrate recognizing PDZ domain. This domain regulates the protease activity of HtrA2 by triggering an intricate allosteric network involving the regulatory loops of the protease domain. We further show that divalent metal ions can both positively and negatively modulate the activity of HtrA2, leading to a refined model of HtrA2 regulation within the apoptotic pathway.

To ensure a functional proteome, cells in all kingdoms of life rely on a variety of different unfoldases and proteases to avoid the accumulation of unfolded proteins and possibly toxic protein aggregates[1–4]. Whereas the functional details for the ATP-dependent systems residing in the cytosolic compartments of the cells were subject to intense studies in recent years and became well understood, the functional and structural details of protein quality control by the ATP-independent serine–proteases of the HtrA-family (High-temperature requirement A) remain only partially understood[3,5–9]. Within humans four different HtrA proteins could be identified: whereas HtrA1, HtrA3, HtrA4 are residing in the cytosol, HtrA2 is mainly located in mitochondria under non-stress conditions[10]. It could be shown that loss of proteolytic activity of this important class of proteins is directly connected to severe diseases, including arthritis, cancer, familial ischemic cerebral small-vessel disease, and age-related macular degeneration, as well as neurodegenerative diseases such as Parkinson's and Alzheimer´s disease[3,6,8,9].

The human mitochondrial serine protease HtrA2 (Omi), a homolog of the *Escherichia coli* heat shock proteins DegP, DegQ, and DegS, is primarily residing in the mitochondrial intermembrane space performing protein quality functions[11–13]. The full-length protein consists of 458 residues, of which the mature protein lacks the 133 amino-terminal residues containing a mitochondrial targeting sequence followed by a transmembrane domain. The protein is initially attached to the mitochondrial inner membrane and undergoes subsequent maturation by proteolytic auto-processing of the transmembrane domain upon apoptotic stimuli[13]. In the resulting membrane-dissociated form, the protein consists of a serine protease domain containing a characteristic catalytic triad (S306, H198, and D228), as well as a carboxy-terminal PDZ domain (PSD-95, DLG, and ZO-1)[14], serving as a substrate recognition domain and restricting access to the active site of the enzyme (Fig. 1a, b and Supplementary Fig. 1). Furthermore, several regulatory loops, termed LA, LD, L1, L2, and L3, within the protease domain are modulating the proteolytic activity of HtrA2 possessing either activating or inhibitory roles[15]. It could also be shown that the mature protein assembles into a homotrimer of ~105 kDa, essential for its proteolytic capabilities[16]. The role of HtrA2 as a key regulator of the apoptotic stress response has been studied in-depth, revealing its role in initiating the apoptotic cascade by processing Inhibitor of Apoptosis proteins (IAPs)[17,18]. In addition, HtrA2 shares the main function of proteases within the mitochondrial and cellular protein quality

[1]Department of Chemistry and Molecular Biology, University of Gothenburg, Göteborg, Sweden. [2]Wallenberg Centre for Molecular and Translational Medicine, University of Gothenburg, Göteborg, Sweden. ✉e-mail: bjorn.marcus.burmann@gu.se

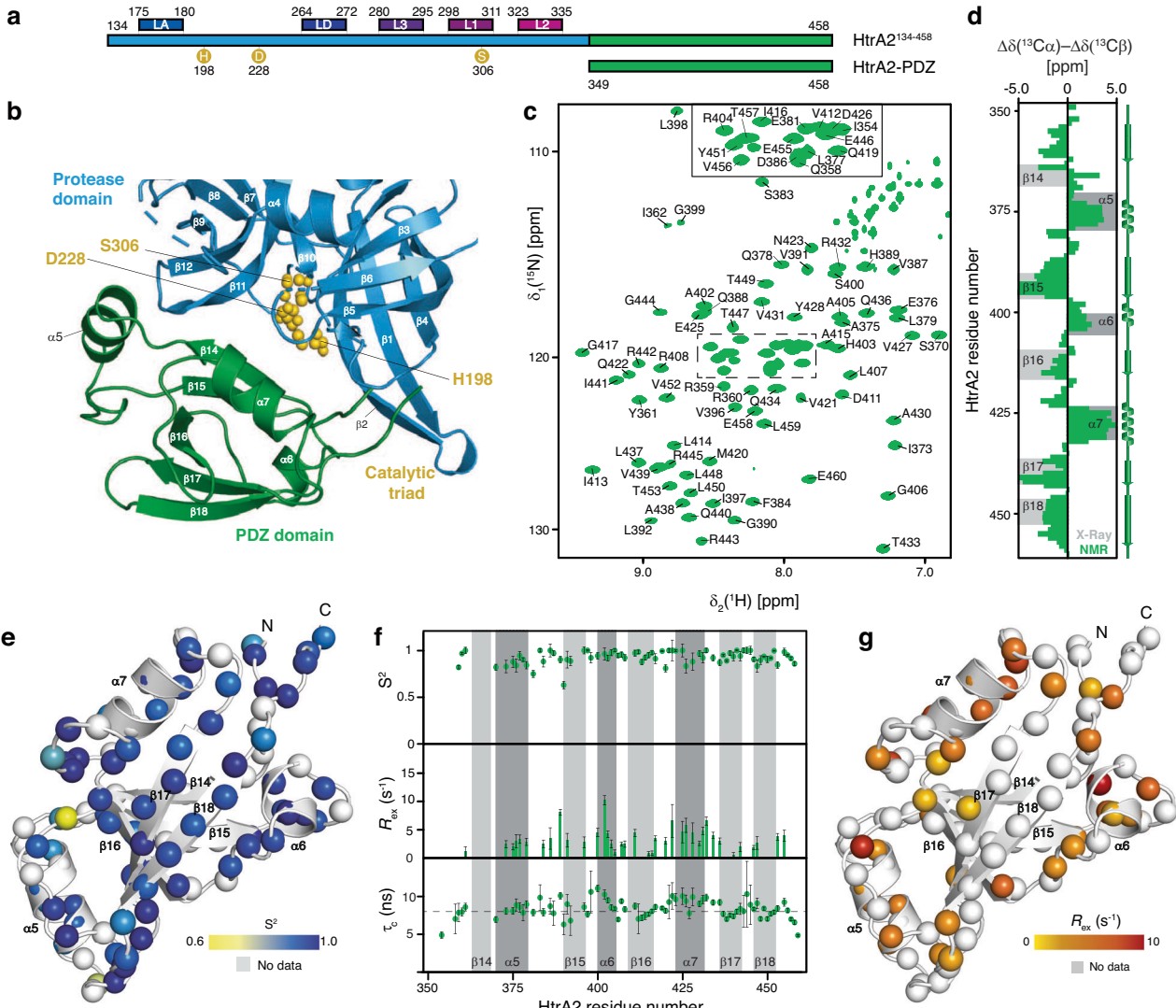

**Fig. 1 | Structure and dynamics of the HtrA2-PDZ domain in solution. a** Scheme of the used constructs indicating the domain structure of mature human HtrA2. Residues of the catalytic triad and important regulatory loops are labeled. **b** Crystal structure of the proteolytically active HtrA2 trimer (PDB-ID: 5M3N) focusing on the protease-PDZ-domain interface of a single subunit (complete trimer: Supplementary Fig. 1). Different domains (PDZ domain: green; protease domain: blue) and the catalytic triad composed of H198, D228, and S306 are indicated (gold). **c** 2D [$^{15}$N, $^{1}$H]-NMR spectrum of the [$U$-$^{15}$N,$^{13}$C]-HtrA2-PDZ domain. The sequence-specific resonance assignment based on triple-resonance experiments is shown. **d** Secondary structure elements of the HtrA2-PDZ domain in solution (green). The secondary structure elements of the HtrA2-PDZ domain within the full-length crystal structure (PDB-ID: 5M3N) are indicated in gray. **e** Generalized order

parameter ($S^2$) reporting on sub-nanosecond motions plotted on the HtrA2-PDZ structure. The backbone amide moieties of the HtrA2-PDZ domain are shown as spheres and the $S^2$ values are indicated by the yellow-to-blue gradient. **f** Analysis of $^{15}$N-backbone relaxation data with the Lipari–Szabo model-free approach. The generalized order parameter $S^2$ reports on pico- to nanosecond motions, the chemical exchange contributions $R_{ex}$ indicating micro- to millisecond motions, and the rotational correlation time $\tau_c$. Broken line represents the average value of 8.2 ns. Error bars indicate the standard fitting error obtained from the nonlinear least-squares minimization. **g** The amide moieties of the HtrA2-PDZ domain are shown as spheres. The calculated $R_{ex}$ is indicated by the yellow-to-red gradient. Source data are provided as a Source Data File.

control network: the prevention of accumulation of protein aggregates leading to cellular malfunction[8,9]. As mitochondrial dysfunction as a consequence of protein accumulation is closely linked to neurodegeneration, a possible prominent neuroprotective function of HtrA2 has been hypothesized[19,20]. This view is further supported by the experimental observation that mutations in the *HTRA2* gene cause hereditary tremors which can progress into Parkinson's disease as well as that mice carrying a loss-of-function variant of HtrA2 develop Parkinson-like symptoms[21,22]. In addition, HtrA2 has also been reported to co-localize within Lewy bodies and has been shown to be able to reduce the propensity of Parkinson-related α-synuclein seeding whilst also contributing to the removal of existing α-synuclein aggregates[23,24].

Despite the importance of HtrA2 in apoptosis regulation and its likely connection to neurodegeneration its functional details remain still largely elusive. Recently, it could be shown that membrane-dissociated HtrA2, prior to activation, is in an equilibrium between trimer and hexamer employing a methyl-TROSY NMR spectroscopy approach[25]. Remarkably, HtrA2 thus uses a similar sequence motif within the PDZ domain as its bacterial homolog DegP, which could be identified as part of a temperature-dependent activation switch for the bacterial protein[25,26]. Together with several available high-resolution X-ray structures[16,27,28], a picture emerges indicating an important role for the PDZ domain in the activation of HtrA2 proteolytic activity.

Here we set out to specify the role of the HtrA2-PDZ domain for the modulation of HtrA2 proteolytic function. By combining solution

NMR spectroscopy, biochemistry, and fluorescence cleavage assays, we show that the inherent dynamics of the PDZ domain play a crucial role for regulating the proteolytic function of the full-length protein triggering an activation cascade of the regulatory loops embedded within the protease domain employing an extended allosteric network. Furthermore, we identify divalent metal ions as important functional modulators, positive or negative, of HtrA2 proteolysis by modulating its activation through targeting the PDZ domain. After initially focusing on the isolated PDZ domain, we establish our observations within the full-length protein exploiting the methyl-TROSY approach, especially suited for large molecular complexes[29]. By combining this powerful technique with paramagnetic relaxation enhancement (PRE) experiments, we were able to identify a previously unknown role of the amino-terminal helix α1 in the activation cascade of HtrA2. Together with recent observations of regulation by transient oligomerization[25] and interprotomer cooperativity[30], our results of divalent ion modulated PDZ dynamics as well as the analysis of methyl NOE patterns and dynamics in full-length HtrA2 lead to a refined picture of the individual components underlying HtrA2 regulation, highlighting important aspects of regulating cellular protein quality control to prevent uncontrolled apoptosis.

## Results

### Isolated HtrA2-PDZ domain in solution

To investigate the structural plasticity and inherent dynamics of the HtrA2-PDZ domain, a construct consisting of the isolated HtrA2-PDZ domain (residues 349–458) was designed (Fig. 1a and Supplementary Fig. 1a). We obtained a nearly complete (97%) assignment of the PDZ domain by sequence-specific backbone and side-chain resonance assignment experiments (Fig. 1c and Supplementary Fig. 2a). Comparison of the secondary structural elements of the HtrA2-PDZ domain in solution as derived from the $C_\alpha$ and $C_\beta$ combined secondary shifts shows a large similarity to the secondary structure elements observed in the crystal structure of the HtrA2-PDZ domain and indicates overall a stably folded domain (Fig. 1d and Supplementary Fig. 2b, c)[16]. The main differences occur in strand β14, which in solution, indicated by the secondary chemical shift analysis (Fig. 1d and Supplementary Fig. 2b), is slightly shifted, and based on the extent of the obtained secondary chemical shift values appears to be only partially populated (~50%) (Fig. 1d). In addition, this segment shows broadened signals in the [$^{15}$N, $^1$H]-NMR spectrum although the non-solvent-exchangeable side-chain carbon resonances could be completely assigned (Supplemental Fig. S2d). This behavior could point to either elevated solvent exchange under the used conditions and/or dynamics on the millisecond timescale for this region of the protein, indicative of either structural heterogeneity and/or exchange processes on the micro- to millisecond NMR timescale[31]. This observation is further supported by the diminished signal intensity in the [$^{15}$N, $^1$H]-NMR spectrum for residues within adjacent strand β15 (Supplemental Fig. S2e), pointing to a similar process and suggesting that the interaction between these two strands is only transiently formed. This can be readily explained by the fact that within the full-length protein the strand β14 locks the PDZ domain onto the protease domain rendering the HtrA2 protease inactive and that these interdomain contacts are broken upon activation[25,32]. Further, we determined the average rotational correlation time ($\tau_c$) for the PDZ domain to 8.2 ns (Fig. 1f), which is in line with the expected theoretical $\tau_c$ value for globular proteins of this molecular weight (~12.1 kDa).

To further assess the overall stability of the PDZ domain we characterized the methyl–methyl NOE networks (Supplementary Fig. 3) within this domain. Since methyl-bearing residues are well distributed throughout the entire domain, we could thereby acquire an overview of the stabilizing intradomain contacts. We observed that the methyl–methyl NOEs identified are in line with the PDZ domain, exhibiting an overall stable fold in agreement with the HtrA2 X-ray

structures previously reported[16,27]. For instance, structural elements that are in close vicinity in the crystal structure such as β16 and β17 or β17 and β18 show several methyl–methyl NOE contacts, whereas α6 and α7 show only a limited number of NOEs to adjacent β-strands (Supplementary Fig. 3b, c). Interestingly, we could observe that the α5 helix displayed no methyl–methyl NOEs to any other structural element within the PDZ domain, indicating that this part of the PDZ domain might not stably stack against the other structural elements of the domain (Supplementary Fig. 3c, f). Having a more detailed look into the amide as well as the carbon NOEs of the α5 helix revealed that this segment is well stabilized within itself forming a short helix perfectly in line with the observed secondary chemical shifts (Supplementary Fig. 3d, e). In contrast, only a limited number of weak NOE contacts to other segments of the domain could be identified, indicating its rather loose attachment leading us to hypothesize that this helix could be an important trigger within the PDZ domain possibly involved in the activation cascade of HtrA2.

### Dynamics of the isolated HtrA2-PDZ domain

In order to elucidate the dynamic properties of the PDZ domain, we analyzed the backbone dynamics of the HtrA2-PDZ domain at a concentration of 500 μM by standard NMR relaxation experiments[33]. The $^{15}$N{$^1$H}-NOE (hetNOE) and $^{15}$N longitudinal $R_1$ relaxation rates report on fast motions on the pico- to nanosecond timescale. For HtrA2-PDZ, we observed a stable hetNOE ratio profile, on average 0.8, with a sharp decrease towards the carboxy-terminus, suggesting that the HtrA2-PDZ domain is a stably folded domain with increased flexibility at the carboxy-terminus, which is devoid of stable secondary structure elements. The $R_1$ rate constants show a similar tendency as the hetNOE data (Supplementary Fig. 4a). We also explored the motions in the micro- to millisecond timescale by measuring the $^{15}$N transverse relaxation rates $R_{2\beta}$ and $R_2$ as derived from the $R_{1\rho}$ rates ($R_{2(R1\rho)}$). The $R_{2(R1\rho)}$ rate constants of HtrA2-PDZ, reporting on motions on the lower microsecond timescale (the used spin-lock radio frequency field of 2000 Hz levels out all exchange contribution ($R_{ex}$) much slower than 80 μs[33]), indicate a similar behavior to that observed within the hetNOE and $R_1$ rate measurements, and we observed mainly planar $R_{2(R1\rho)}$ rate constants (Supplementary Fig. 4b). The $R_{2\beta}$ rate constants, however, show a notable increase in the regions between α5–β15 and β17–β18, suggesting dynamics within these segments on the higher micro- to millisecond timescale, pointing to a possible conformational exchange occurring in these loop regions (Supplementary Fig. 4b). To rule out any contribution of a large anisotropic diffusion tensor to the observed behavior, we also plotted the $R_1 \cdot R_{2\beta}$ values alongside the $R_{2\beta}$ rate constants, showing the same trend as the $R_{2\beta}$ rates and thus no indication of such a contribution (Supplementary Fig. 4b). Next, we analyzed the backbone relaxation data with the Lipari–Szabo model-free approach using an axially symmetric diffusion tensor (Supplementary Fig. 4c, d) to obtain the generalized order parameters ($S^2$) and the chemical exchange contributions ($R_{ex}$)[34,35]. We observed generalized order parameter values with an average of 0.9 ± 0.1 (Fig. 1e, f), indicating structural rigidity for the whole domain, including the loosely attached helix α5 on the pico- to nanosecond timescale. Nevertheless, we observed increased $R_{ex}$ terms for residues in the β15 preceding linker region, as well as in α6 and the region involving α7 and the linker region between α7 and β17, pointing to the presence of chemical exchange contributions as initially observed by the enhanced $R_{2\beta}$ rates (Fig. 1f, g).

To quantify these exchange contributions, we next employed a BEST TROSY $^{15}$N Carr-Purcell-Meiboom-Gill (CPMG) relaxation dispersion (RD) experiment[36]. The obtained relaxation dispersion profiles provide insight into protein dynamics, where non-flat profiles indicate micro- to millisecond dynamics and flat profiles their absence (Supplementary Fig. 4e, f). The CPMG data recorded at 298 K revealed micro- to millisecond dynamics within the amino-terminus, the α5

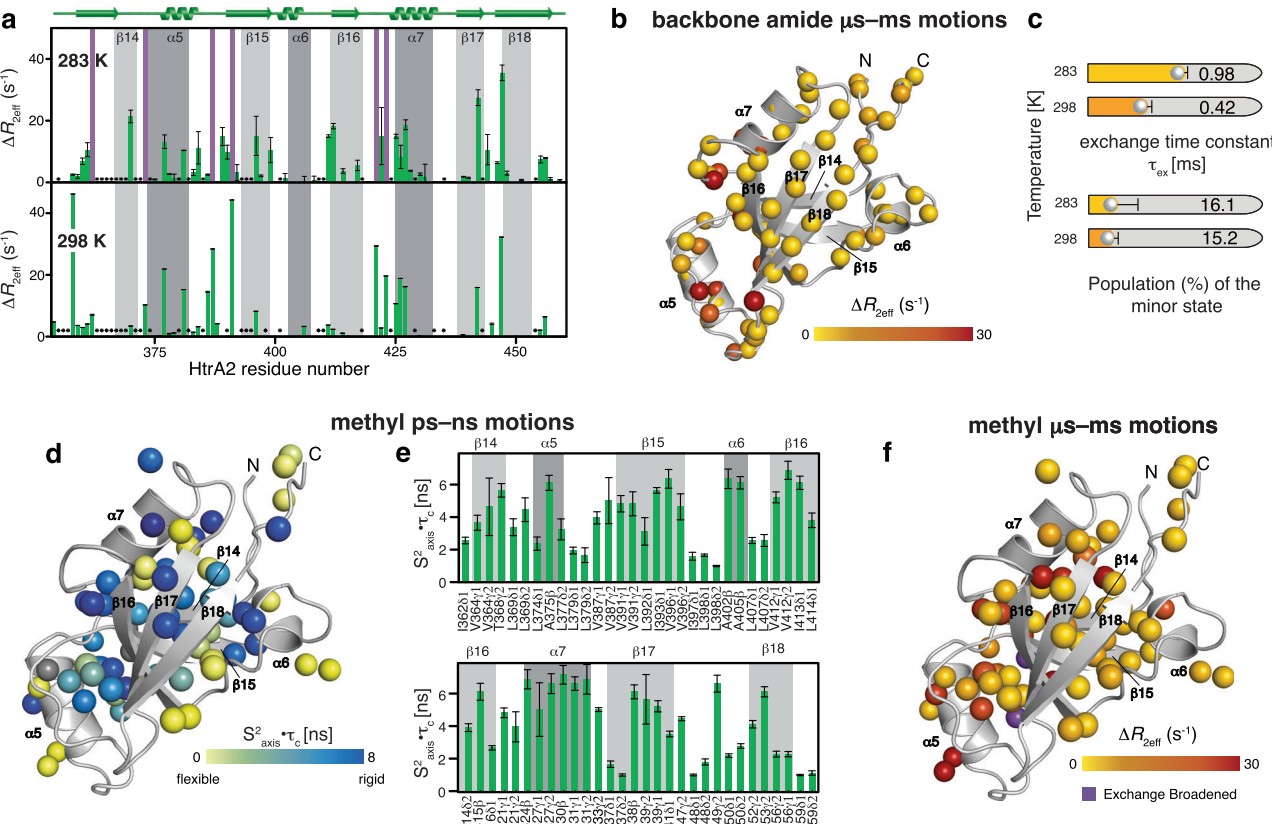

**Fig. 2 | Subtle dynamic adaptations within the PDZ domain assessed by backbone and ALVIT methyl dynamics. a** $\Delta R_{2eff}$ values for the backbone amide groups, obtained from the difference of $R_{2eff}$ at the lowest and highest CPMG frequency $\upsilon_{CPMG}$, at the indicated temperatures. Residues exchange broadened are highlighted by purple bars; unassigned residues indicated by an asterisk (*). Error bars indicate the standard fitting error obtained from the nonlinear least-squares minimization of one experiment ($n = 1$). **b** Amplitude of the CPMG relaxation dispersion profiles $\Delta R_{2eff}$ at 21.1 T of an [$U$-$^{15}$N,$^{13}$C]−PDZ sample at 298 K. **c** Rate constants of the dynamic process ($\tau_{ex}$) and the relative populations of PDZ states obtained from a global fit of the CPMG relaxation dispersion data. Error bars indicate the standard fitting error obtained from the nonlinear least-squares minimization of one experiment ($n = 1$). **d, e** Local methyl group dynamics on the pico- to nanosecond timescale probed by methyl single-quantum (SQ) and triple-quantum (TQ) relaxation experiments showing the product of the local order parameter and the overall tumbling constant, $S^2_{axis} \cdot \tau_C$. Measurements were performed on an [$U$-$^2$H, Ile-$\delta_1$-$^{13}$CH$_3$, Leu, Val-$^{13}$CH$_3$, Ala-$^{13}$CH$_3$, Thr-$\gamma_2$-$^{13}$CH$_3$]−PDZ sample at 298 K. Methyl groups (spheres) and the obtained $S^2_{axis} \cdot \tau_C$-values (yellow-to-blue gradient) (**d**). $S^2_{axis} \cdot \tau_C$-values plotted against the HtrA2-PDZ amino acid sequence (**e**). Error bars indicate the standard fitting error obtained from the nonlinear least-squares minimization of one experiment ($n = 1$). **f** Amplitude of the CPMG relaxation dispersion profiles $\Delta R_{2eff}$ at 16.4 T. Source data are provided as a Source Data File.

helix and the adjacent loop connecting to β15, as well as α7 and its preceding loop (Fig. 2a, b). By transferring the sequence-specific resonance assignment to lower temperatures (283 K) we were then able to compare this data to the initial CPMG data recorded at 298 K. At lower temperature, we observed dynamics in addition also within α7 and the β17−β18 loop, to a larger extent compared to our observations at 298 K and additionally evidenced as line-broadening (Fig. 2a and Supplementary Fig. 4e−h). Quantitative global analysis of the dispersion data showed that this motion occurred on 0.98 ± 0.005 ms (283 K) as well as 0.42 ± 0.004 ms (298 K) timescale (Fig. 2c). At the same time the population of the minor state was also slightly modulated by the temperature, decreasing from 16.1 to 15.2% (Fig. 2c).

**Methyl side-chain dynamics reveal enhanced motions within HtrA2-PDZ**

To derive the origin of the local exchange contributions identified by the backbone relaxation analysis, we exploited the increased sensitivity of the methyl groups to gain insight into the side-chain dynamics of the HtrA2-PDZ. Using an ALVIT-labeled HtrA2-PDZ sample we first determined the product of the side-chain order parameters and the correlation time of the overall molecular tumbling ($S^2_{axis} \cdot \tau_C$), reporting on the extent of the amplitude of motions on the pico- to nanosecond

NMR timescale (Fig. 2d, e). Overall, the obtained values indicated the stable fold of the domain averaging to $S^2_{axis} \cdot \tau_C = 4.3 \pm 0.5$ ns for residues residing in the core of the domain, comparable to values obtained for the PDZ domains of bacterial DegP[26]. Looking more specifically at the local distributions of the $S^2_{axis} \cdot \tau_C$ values, we observed that these values indicated increased flexibility for residues in the α5-β15-α6 region as well as in α7 and the carboxy-terminus (Fig. 2d, e). As we had already obtained indications of specific line-broadening in a 2D [$^{13}$C,$^1$H]-NMR spectrum for some methyl resonances, e.g., I373 and V364 (Supplementary Fig. 2a), we used a multiple quantum (MQ) CPMG relaxation dispersion experiment[37]. The obtained CPMG relaxation dispersion profiles report on the micro- to millisecond dynamics and showed increased rate constants for residues in the α5 to β15 region as well as for residues in the β16/α7 with a 16.6 ± 0.4% minor state population on a 1.19 ± 0.07 ms timescale (Fig. 2f and Supplementary Fig. 5a, b). The obtained data for both, the protein amide backbone as well as the methyl groups, indicate that the PDZ domain is experiencing micro- to millisecond dynamics, indicative of the same underlying process. These dynamics are manifested within the α5 helix and the adjacent linker region connecting α5 and β15, pointing to a possible importance of these inherent local dynamics for the regulation of HtrA2 activation and its protease activity.

## Oligomerization properties of the HtrA2-PDZ domain

Given the recently identified hexameric inactive form of HtrA2[25], we next assessed a potential transient oligomerization of the isolated HtrA2-PDZ domain by size exclusion chromatography coupled with multi-angle light-scattering (SEC-MALS) experiments using a range of different HtrA2-PDZ concentrations. Although the accessible range of protein concentrations was limited by solubility, MALS data for dimerization can be fitted with constrained values for the titration end point masses and lower limits of $K_D$ values can be obtained from such solubility-limited datasets[38]. Under the chosen experimental conditions, the HtrA2-PDZ domain showed only a minute tendency to dimerize with an estimated lower limit of a $K_D$ of $13 \pm 1.65$ mM (Supplementary Fig. 5c, d). This weak transient interaction is about three orders of magnitude weaker than the recently reported trimer-to-hexamer equilibrium for the full-length protein by NMR spectroscopy with ~20 µM[25]. This difference can be likely attributed to the presumably cooperative nature of the interaction between the three PDZ:PDZ interfaces within the hexameric species[25].

## Modulation of the PDZ domain by divalent ions facilitates HtrA2s proteolytic activity

We hypothesized that due to HtrA2s role in apoptotic signaling its proteolytic function might be modulated by divalent cations, such as $Ca^{2+}$, as bacterial HtrA proteins have previously been shown to be modulated by divalent ions[39,40]. Upon initially adding $Mg^{2+}$ and $Ca^{2+}$ ions to the HtrA2-PDZ NMR buffer, we were able to detect localized chemical shift changes within the PDZ domain, indicative of an ability of the domain to bind divalent ions. A detailed analysis of the resulting chemical shift perturbations for each residue revealed that the regions with the most pronounced chemical shift changes were located within the loop region between α5 and β15, as well as the loop region preceding α7. We then extended our analysis by also testing $Cu^{2+}$ and $Zn^{2+}$, previously reported as inhibitors for bacterial HtrAs[39,40], detecting similar chemical shift perturbations towards HtrA2-PDZ (Supplementary Fig. 6a–d). Upon addition of $CuCl_2$ as well as $ZnSO_4$ we also observed a signal attenuation in the NMR spectra to about 60% and 80%, respectively. Whereas this effect can be attributed for $ZnSO_4$ to a possible oligomerization, in case of $CuCl_2$ it is likely caused by enhanced relaxation due to the presence of paramagnetic $Cu^{2+}$ leading to paramagnetic relaxation enhancement (PRE)[41]. Therefore, the observed effect in the presence of $Cu^{2+}$ reflects rather the solvent accessibility of the different amide moieties as the effect was most pronounced for the loop regions in general (Supplementary Fig. 6c). Coupled with our observations of enhanced dynamics on the micro- to millisecond timescale of both the side-chain methyl groups and the backbone in the regions of HtrA2-PDZ where the divalent metal ions bind, we hypothesized that these inherent dynamics might play a functional role in the modulation of the protease function of HtrA2 by divalent cations. Using the MQ CPMG on the methyl groups for HtrA2-PDZ samples supplemented with either $CaCl_2$ or $ZnSO_4$, we observed however only minor modulations of the inherent dynamics compared to the HtrA2-PDZ in the absence of divalent cations (Supplementary Fig. 6e–h). We next used the NMR line-shape analysis tool TITAN to determine the metal affinity to the HtrA2-PDZ domain from titration series[42]. We were able to determine $K_D$ values for $Ca^{2+}$ and $Zn^{2+}$ in the lower millimolar range ($4.1 \pm 0.5$ mM and $1.6 \pm 0.3$ mM for $Ca^{2+}$ and $Zn^{2+}$, respectively) and the dissociation rate $k_{off}$ to $84.3 \pm 29$ s$^{-1}$, and $52.8 \pm 15.3$ s$^{-1}$ for $Ca^{2+}$ and $Zn^{2+}$, respectively (Supplementary Figs. 7 and 8). We were not able to reliably fit the data from the $Mg^{2+}$ and $Cu^{2+}$ titration series.

To assess the potential functional role of the divalent cations, we subsequently tested the proteolytic activity of mature HtrA2 toward β-casein in the presence of either $Mg^{2+}$, $Ca^{2+}$, $Cu^{2+}$, or $Zn^{2+}$. The activity of HtrA2 toward β-casein was increased when $Ca^{2+}$ was supplemented, and to a lesser extent also in the presence of $Mg^{2+}$, compared to the activity of the protease in the standard assay buffer (Fig. 3a, b and Supplementary Fig. 9). To rule out any effects of the divalent ions on the β-casein directly, we also performed controls in the absence of HtrA2, resulting in no cleavage, which clearly attributes the observed enhancement of HtrA2 to a modulation of its proteolytic activity by $Ca^{2+}$ and $Mg^{2+}$ (Supplementary Fig. 10). In contrast, the addition of $Cu^{2+}$ and $Zn^{2+}$ strongly inhibited HtrA2 protease activity (Fig. 3a, b and Supplementary Fig. 9). The observed inhibitory effect of $Zn^{2+}$ and $Cu^{2+}$ agree with previously reported effects of these ions on related HtrA proteins[39,40].

Previously, a hexameric species of HtrA2 has been characterized as a closed inactive state with reduced substrate affinity compared to the canonical trimeric species of HtrA2[25]. Based on this previous finding and the signal attenuation observed when titrating $ZnSO_4$ toward HtrA2-PDZ, we reasoned that the different divalent ions might influence the oligomeric equilibrium of HtrA2. Therefore, we performed SEC-MALS experiments with the HtrA2-PDZ domain in the presence of the four metal ions used in the proteolytic assay. While the addition of $Mg^{2+}$, $Cu^{2+}$, and $Ca^{2+}$ did not result in a notably increased dimerization rate of the HtrA2-PDZ domain compared to the non-supplemented buffer, the addition of $Zn^{2+}$ markedly increased the propensity ~50-fold with which the HtrA2-PDZ domains dimerized (Supplementary Fig. 11 and Supplementary Table 1).

## Structural basis of the interaction between HtrA2-PDZ and an activating peptide

It has been previously demonstrated that HtrA2 can be activated by the interaction between its PDZ domain and target substrates or custom-made peptides designed for optimized interaction with the HtrA2-PDZ domain, similar to bacterial DegS and DegP[43–45]. We looked more closely at this activation step by analyzing the interaction between the isolated HtrA2-PDZ domain and a customized activator peptide, DD-PDZOpt[25]. We probed the interaction via titration experiments focusing on the backbone using 2D [$^{15}$N, $^1$H]-NMR spectra, as well as by an ALVIT methyl-labeled PDZ sample by recording 2D [$^{13}$C, $^1$H]-NMR spectra (Supplementary Fig. 12a–f). While we observed only minor chemical shift changes on the backbone amide moieties, we were able to see the effects of the activating peptide binding more pronounced on the methyl groups. We detected significant chemical shift changes in the regions of β14, β15, and α7 (Supplementary Fig. 12e, f), which correlates well with previous observations of a HtrA2 peptide binding cleft comprising these regions of the PDZ domain[25,46]. To obtain detailed insight, we then used TITAN[42] to analyze the titration data of HtrA2-PDZ and DD-PDZOpt. Employing a two-state ligand binding model, we determined the dissociation constant $K_D$ to $3.18 \pm 0.9$ µM, with a dissociation rate $k_{off}$ of $48.5 \pm 2.9$ s$^{-1}$ (Supplementary Fig. 12g), in good agreement with a reported $K_D$ of 7.5 µM between wild-type HtrA2 and a related 13-residue activating peptide[47]. Based on the obtained CSPs we docked the DD-PDZOpt peptide onto the isolated PDZ domain by HADDOCK[48,49] (Supplementary Fig. 12h and Supplementary Table 2), revealing a virtual identical arrangement as previously determined by X-ray crystallography for a related activating peptide (PDB-ID: 2PZD; Supplementary Fig. 12i). Finally, we investigated if the binding influences the transient dimerization of the HtrA2-PDZ domain. The SEC-MALS analysis resulted in a similar dimerization constant, $10.3 \pm 2.5$ mM, as observed for the isolated PDZ, ruling out any contribution to the monomer–dimer equilibrium (Supplementary Fig. 11i, j).

## Dissecting the role of divalent ions in a coupled enzymatic assay

Next, we assessed the proteolytic efficiency of HtrA2 in coupled enzymatic fluorescence-based cleavage assays of the H2Opt peptide, a self-quenching fluorescent substrate peptide, in the presence of the activating peptide DD-PDZOpt. In accordance with previous studies[25,44,47], we observed that the proteolytic efficiency of HtrA2

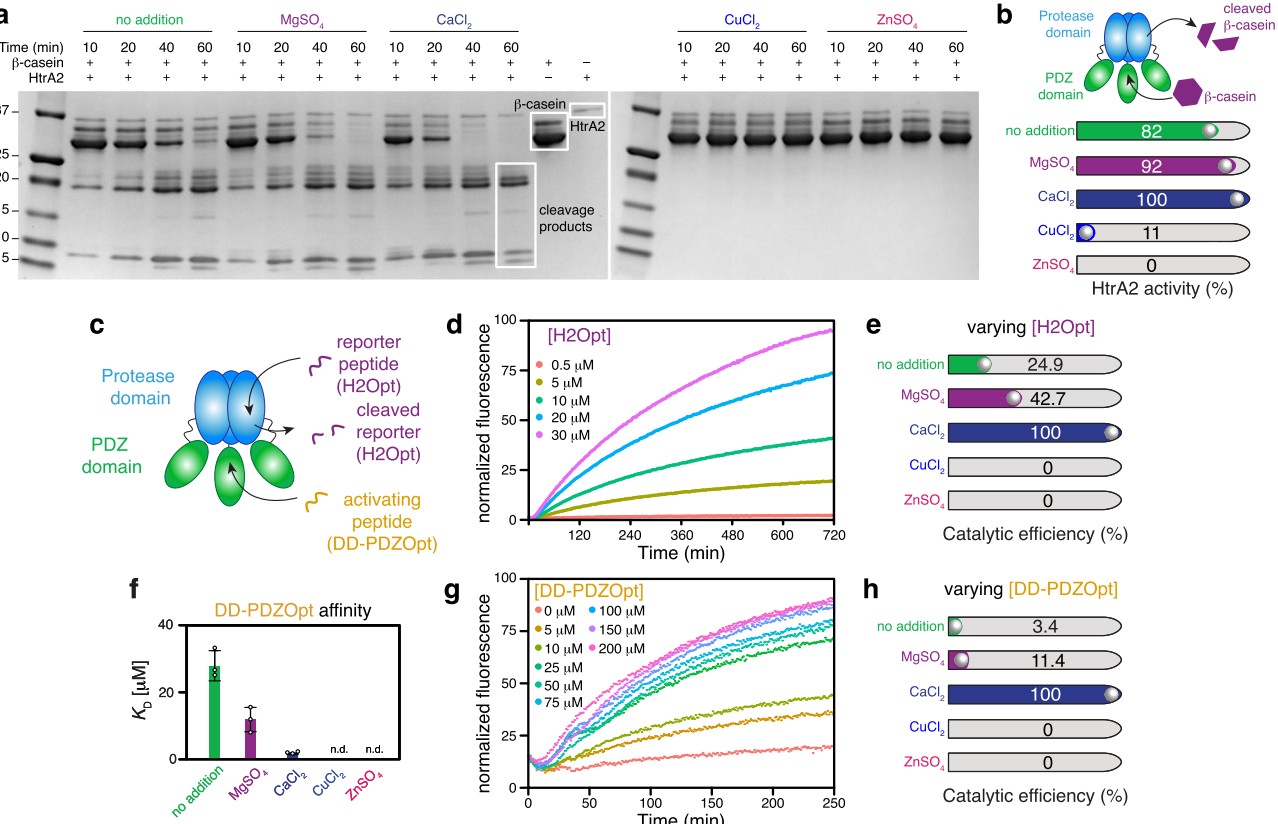

**Fig. 3 | Divalent metal ions modulate the proteolytic activity of HtrA2.**
**a** Proteolytic assay using β-casein shows that Mg²⁺ and Ca²⁺ enhanced the proteolytic activity of HtrA2, while supplementation of Zn²⁺ and Cu²⁺ strongly inhibited proteolytic activity. Cleavage assays were done as biological triplicates, yielding similar results (Supplementary Fig. 9). **b** Comparison of the proteolytic efficiency of HtrA2 towards β-casein without or with divalent metal ions. Cleavage is expressed as percentage of β-casein cleaved after 40 minutes of incubation with HtrA2. Bar colors correspond to the color assigned to each metal ion in (**a**). **c** Scheme of the fluorescence assay using an activating peptide and a reporter peptide.
**d** Fluorescence assay showing activity of HtrA2 with an HtrA2 activating peptide (DD-PDZOpt, 50 μM) using varying reporter peptide concentrations as indicated.

Data is representative of three replicate experiments. **e** Analysis of the catalytic efficiency ($k_{cat}/K_M$) of the assay shown in (**d**) in the absence or presence of divalent metal ions as indicated. **f** Determination of the apparent $K_D$ ($K_{D,app}$) of the activator peptide using varying peptide concentrations. n.d. indicates not determined. Values are the averages of $n = 3$ individual repeats, and error bars indicate the SD of these replicates. **g** Fluorescence assay showing activity of HtrA2 using varying HtrA2 activating peptide concentrations as indicated with a fixed reporter peptide concentration (20 μM). Data are representative of three replicate experiments. **h** Analysis of the catalytic efficiency ($k_{cat}/K_M$) of the assay shown in (**g**) with or without divalent metal ions. Source data are provided as a Source Data File.

towards the fluorescent substrate was greatly enhanced when the activating peptide was present (Fig. 3g). We extracted kinetic parameters for wild-type HtrA2 using two variants of the fluorescence assay: either using a fixed H2Opt concentration and varying the concentration of DD-PDZOpt or alternatively using a fixed concentration of DD-PDZOpt and varying the concentration of the H2Opt peptide. Based on the assay with varying substrate (H2Opt) concentration, we determined a $k_{cat}/K_m$ of ~85 M⁻¹ s⁻¹ in standard assay buffer devoid of additional metal ions (Fig. 3d, Supplementary Fig. 13, and Supplementary Table 3). It should be noted that we used here a sub-saturating concentration of DD-PDZOpt (50 μM) that is not sufficient to fully activate HtrA2, for the purpose of analyzing the activation of HtrA2 in the presence of low concentrations of activating peptide. The addition of 2 mM CaCl₂ to the assay resulted in a doubling of the $v_{max}$ value and a decrease of the $K_M$ value of about 50% compared to the HtrA2 activity in our standard assay buffer (no metals supplemented), indicating that the enzyme affinity to the substrate is notably enhanced in the presence of Ca²⁺ and as a result, Ca²⁺ binding increased the $k_{cat}/K_M$ value about fourfold (Fig. 3e). Addition of Mg²⁺ lowered the $K_M$ value of HtrA2 to about 25% of that in standard assay buffer, but the $v_{max}$ was markedly decreased, resulting in a $k_{cat}/k_M$ increase of almost twofold (Fig. 3e). Performing the experiment with 2 mM ZnSO₄ or 2 mM CuCl₂

supplemented showed, as expected from the β-casein assay, no detectable protease activity (Fig. 3e and Supplementary Fig. 13).

Having established the effect of calcium ions as an uncompetitive activator, as it increases the velocity of the reaction and at the same time decreases the $K_M$, we wanted to clarify if it affects the activation and/or the proteolysis itself. To assess this question, we next used a fixed concentration of substrate (H2Opt) and varying concentrations of activating peptide DD-PDZOpt to extract kinetic parameters under variable concentrations of the activating peptide. We extracted the apparent $K_{D,app}$ and $v_{max}$ from the experimental data observing that Ca²⁺ also in this instance lowered the $K_{D,app}$ but that the $v_{max}$ remained at about the same rate as in standard assay buffer, and $k_{cat}/K_M$ of Ca²⁺-bound HtrA2 increased ~13-fold compared to HtrA2 in standard assay buffer (Supplementary Fig. 14 and Supplementary Table 4). Mg²⁺-bound HtrA2 had a decrease in $K_{D,app}$ to about 50% of HtrA2 in standard assay buffer but a lowered $v_{max}$ and as such a $k_{cat}/K_M$ of about 50% compared to HtrA2 in standard buffer. The addition of CuCl₂ would inhibit HtrA2 but the inhibition could be relieved by concentrations of DD-PDZOpt above ~100 μM (Supplementary Fig. 14d), while Zn²⁺ addition completely abolished HtrA2 activity at all concentrations of activating peptide tested. It has previously been suggested for other HtrA proteins that Zn²⁺ may increase oligomer stabilization and Zn²⁺

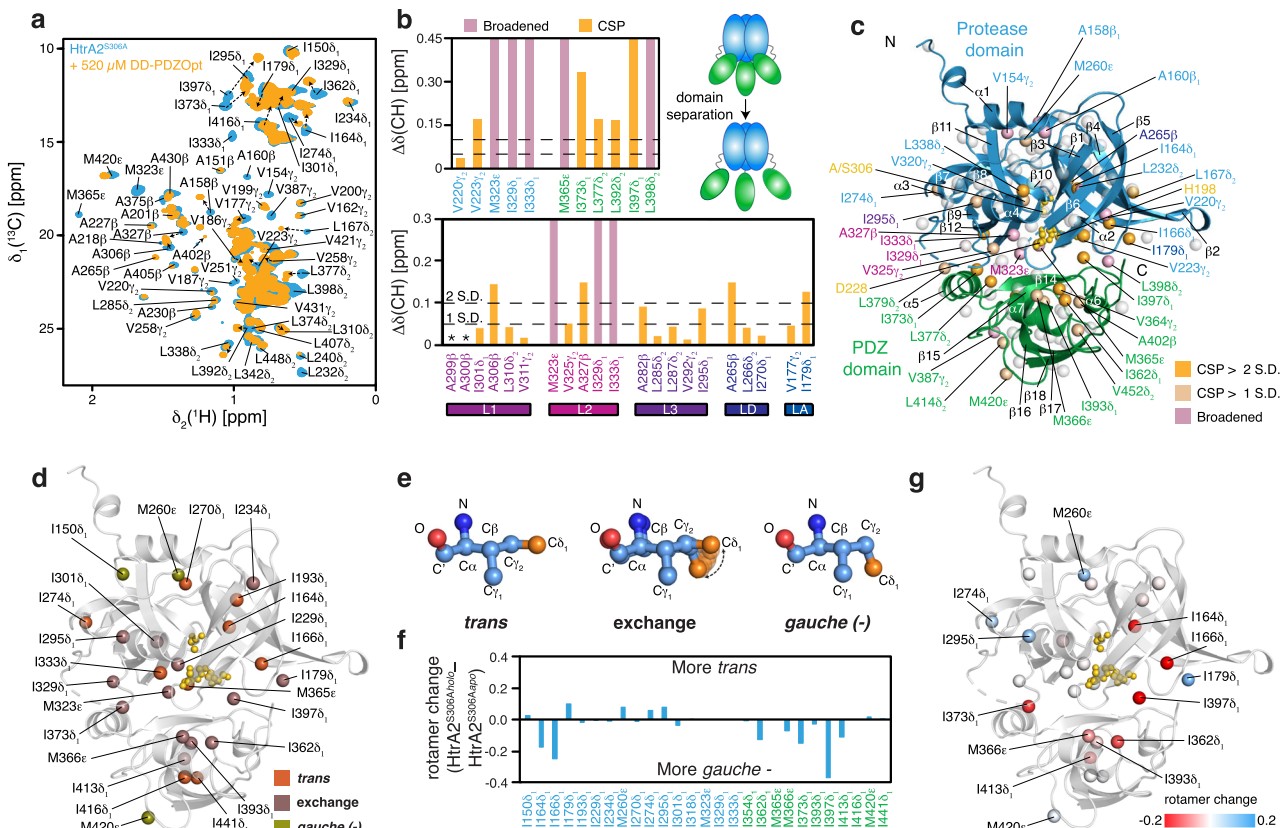

**Fig. 4 | Localized chemical shift changes delineate the allosteric activation of HtrA2. a** 2D [$^{13}$C, $^{1}$H]-NMR spectra showing titration of DD-PDZOpt toward an 80 μM sample of MALVI$^{proS}$-labeled HtrA2$^{S306A}$. Arrows indicate large CSPs. **b** Upper panel shows residues belonging to the protease domain (blue) or the PDZ domain (green) showing significant CSPs (>1 SD or 2 SD indicated by the broken lines) and line-broadening indicative of PDZ–protease domain separation. Lower panel shows the impact of the titration of DD-PDZOpt on the different regulatory loops of HtrA2 modulating its proteolytic function. Asterisks mark unassigned residues. **c** Significant CSPs and broadened residues following DD-PDZOpt titration highlighted onto the HtrA2 monomer. Color scheme used for annotating the methyl groups is analogous to (**b**). **d** Rotameric states of isoleucine and methionine residues as determined by the methyl $^{13}$C shifts in the closed inactive state of HtrA2 (PDB-ID: 5M3N). The catalytic triad residues are shown in gold. **e** Schematic view of the *trans* and *gauche (−)* rotameric states of the isoleucine side-chain. **f** Changes in rotamer states between the holo and apo form of HtrA2$^{S306A}$. Residues belonging to the protease domain (blue), residues belonging to the PDZ domain (green) are indicated. **g** Rotameric changes as seen in (**f**) plotted onto the structure of HtrA2 (PDB-ID:5M3N). Red spheres show residues in a more *gauche (−)* conformation while blue spheres show residues in a more *trans* conformation. The catalytic triad residues are shown in gold. Source data are provided as a Source Data File.

ions have been found to bind to the hexameric form of *Synechocystis* HtrA homolog A[39,50]. In light of the SEC-MALS results indicating that the isolated PDZ domain is prone to dimerize in the presence of Zn$^{2+}$ but not in the presence of any other of the tested ions as well as the binding preference of activating peptides to trimeric rather than hexameric HtrA2, this provides a possible explanation for our observations in the proteolytic assays: HtrA2 potentially forms hexamers via the PDZ domain in the presence of Zn$^{2+}$, preventing efficient binding of the activating peptide and thus keeping HtrA2 in an inactive state. Our experimental data with Cu$^{2+}$ showed that the inhibition caused by Cu$^{2+}$ could be relieved by increasing concentrations of DD-PDZOpt (Supplementary Fig. 14), pointing to competitive inhibition by Cu$^{2+}$ with the activating peptide, in line with our SEC-MALS data showing no indication that the copper ions influence the trimer-to-hexamer equilibrium of HtrA2.

### Elucidating the allosteric regulation of HtrA2$^{S306A}$ using methyl-TROSY

For expanding our characterizations also to the full-length protein, we used the catalytically inactive trimeric HtrA2$^{S306A}$ variant to suppress any eventual self-cleavage at the high protein concentrations needed for NMR studies[25]. By using previously reported assignments for ILVM$^{proR}$ (referring to methyl group labeling of Ile-δ$_1$, Leu-δ$_1$ (*proR*), Val-γ$_1$ (*proR*), and Met-ε) HtrA2$^{S306A}$ (see refs. 25,30) as well as our PDZ side-

chain assignment, we were able to complete the sequence-specific assignment to ~92% of the methyl groups of a MALVI$^{proS}$ (referring to methyl group labeling of Met-ε, Ala-β, Leu-δ$_2$ (*proS*), Val-γ$_2$ (*proS*), and Ile-δ$_1$) labeled HtrA2$^{S306A}$ (Supplementary Fig. 15). We then used this sample to monitor the binding between full-length mature trimeric HtrA2$^{S306A}$ (Supplementary Fig. 1b) and DD-PDZOpt to eventually elucidate the distinct steps underlying its allosteric activation. Also here, we observed significant chemical shift changes as well as line-broadening of residues located in the β$_{14}$ and β$_{15}$ strands of the PDZ domain where the peptide binding cleft is located, as well as in the β$_{11}$ and β$_{12}$ strands of the proteolytic domain (Fig. 4a–c and Supplementary Fig. 16a). These regions are in close contact in the closed HtrA2 conformation, where the proteolytic domain stacks against the PDZ domain, restricting access to the catalytic site[16,25]. We observed significant chemical shift changes within the regulatory loops, pointing to an activation cascade of L2-L3-LD-L1 leading to a correct arrangement of the catalytic triad upon binding of the activating peptide to the PDZ domain (Fig. 5a and Supplementary Fig. 16b, c)[27]. In addition to the CSPs, line-broadening of residues in helix α1 of the protease domain was observed in a similar manner as for the PDZ-domain protease domain interface, which is in general indicative for either conformational heterogeneity and/or micro- to millisecond dynamics. As previous studies confirmed that the effects in the case of the interdomain interface reflect on the detachment of the PDZ domain[25], the

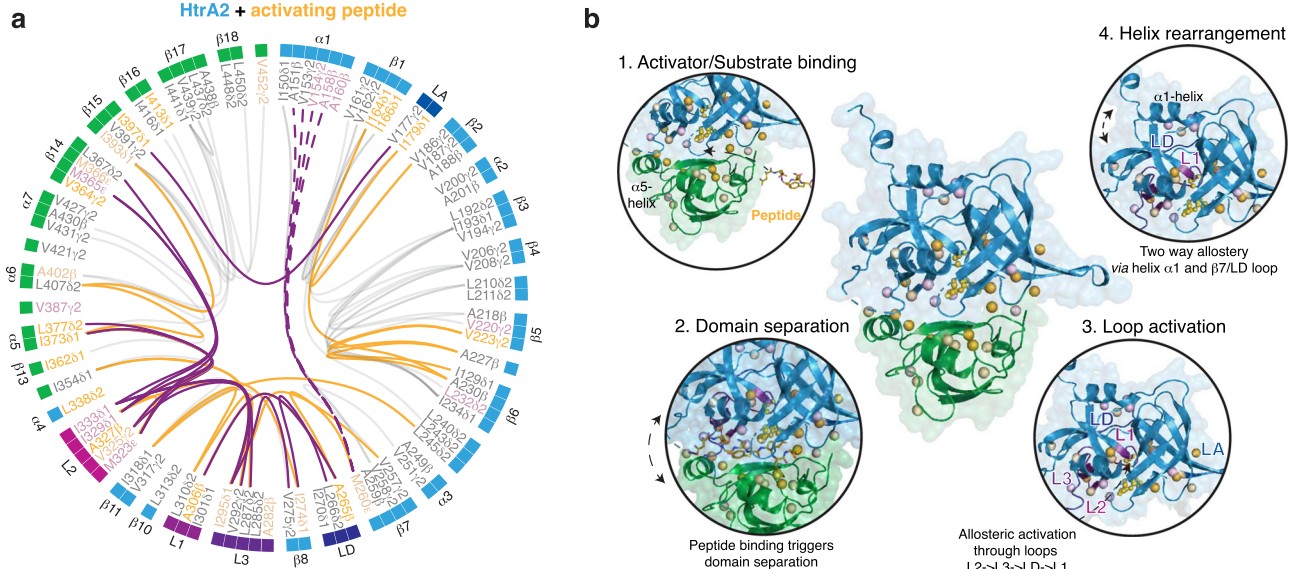

**Fig. 5 | Distinct steps underlying the allosteric activation of HtrA2. a** Flareplot representation visualizing the changes, based on the altered methyl NOE interaction patterns upon DD-PDZOpt interaction (see individual representations of the apo and holo-state in Supplementary Fig. 16). Purple lines represent proposed allosteric coupling of the activation loops, whereas the broken lines represent the putative coupled detachment of helix α1. Yellow lines represent altered NOE contacts only observed in the presence of the activating peptide in line with local structural rearrangements required for the allosteric coupling. **b** Model of the allosteric modulation of the HtrA2 monomer upon DD-PDZOpt binding involving distinct regulatory steps, leading to the activation of the proteolytic function (for details see main text).

observation here for helix α1 could suggest that also this helix experiences dynamical changes upon substrate binding, pointing to a similar detachment. The effect on α1 is likely triggered by rearrangements in the LD loop and strand β7 harboring the central M260, stacking closely to helix α1.

The local structural arrangement of amino acids side-chains can be crucial in substrate binding events with side-chain rotamer changes commonly occurring during substrate binding leading to rearrangements and possible allosteric regulation[51]. We therefore compared the methyl side-chain rotamer states, extracted from the isoleucine and methionine $^{13}$C chemical shifts, of the apo form of HtrA2$^{S306A}$ and its holo form bound to the activating peptide DD-PDZOpt (Fig. 4d–g). Particularly at the domain interface several methyl groups are in an exchange between *trans* and *gauche (−)* rotameric states in the closed apo HtrA2 state (Fig. 4d, e). Upon activation, we observed several residues whose rotameric states changed more towards the *gauche (−)* rotamer at the interdomain interface, e.g., I362, M366, I373, and I397 (Fig. 4f, g). Furthermore, two isoleucine residues in the proximity of the active site, I164 and I166, had an ~20% increased *gauche (−)* population, pointing to local subtle structural adaptations underlying HtrA2 activation (Fig. 4f, g).

To obtain a more detailed picture of the functional consequences of DD-PDZOpt binding to the PDZ domain, we next characterized the methyl–methyl NOEs within the MALVI$^{proS}$-HtrA2$^{S306A}$ in the absence of the activating peptide (Supplementary Fig. 16b). The identified NOEs were consistent with the previously reported crystal structures of HtrA2 [[16,27]] and revealed a network of contacts stabilizing the regulatory loops within the protease domain, whereas especially the L2 loop was stabilized via characteristic interdomain NOEs to β14 as reported before[25] (Supplementary Fig. 1e). In addition, we could identify additional NOEs between helix α5 and the L2 loop as well as between β15 and the LA loop (Supplementary Fig. 16b, d). We then analyzed the NOE patterns in the peptide-bound state of MALVI$^{proS}$-HtrA2$^{S306A}$. As expected from the CSPs we also see altered NOE patterns, particularly at the amino-terminus as well as in the vicinity of the regulatory loops (Supplementary Fig. 16c). Looking particularly at I150 we can observe that this residue is involved in an intricate NOE network within the apo state, which is largely diminished in the peptide-bound

state (Supplementary Fig. 16d), which together with the previously discussed line-broadening of several residues involved in a stabilizing NOE network of helix α1 point to structural changes in this region upon activation. In line with the earlier observation of positive cooperativity upon activator peptide binding[30], we also observed changes in the interprotomer NOEs, particularly around the crucial I270, I274, and I150 residues, suggesting local rearrangements at the monomer interfaces (Supplementary Fig. 16d). Other regions of changes in NOE patterns were in the vicinity of the regulatory loops suggesting together with the observed subtle adaptations of the side-chain rotamers a switch from the inactive to active conformation around the catalytic site (Supplementary Fig. 16c). Importantly, we also could detect highly similar NOE patterns for several methyl groups experiencing only minute changes, as highlighted for a local cluster around I333 (Supplementary Fig. 16c, e). This observation rules out that we were detecting a reduced number of NOE cross peaks simply due to sensitivity issues.

In summary, based on the observed methyl CSPs throughout the HtrA2 protein in combination with the altered NOE networks (Fig. 5a), we were able to identify distinctive steps underlying HtrA2 activation (Fig. 5b). Binding of the activator peptide leads to a transition to the HtrA2 open state by detaching the PDZ domain from the protease domain. Particularly, the release of the loop L2 from this domain enables the triggering of the rearrangement cascade L2-L3-LD-L1. This allosteric signaling in the final step leads to perturbations of the amino-terminal α1 helix, and whilst the observed data might indicate a detachment of the α1 helix, based on the observed broadening of the methyl resonances and weakened NOE contacts for I150 no definite conclusions can be drawn, yet.

To also obtain insight into the dynamical adaptations and to probe the relative domain motions, we used the $S^2_{axis} \cdot \tau_C$ of the MALVI$^{proS}$-HtrA2$^{S306A}$ apo state and the peptide-bound MALVI$^{proS}$-HtrA2$^{S306A}$. In the apo state we obtained values of $38 \pm 19$ ns and $37 \pm 24$ ns for the protease and PDZ domain (Supplementary Fig. 17a, c), respectively, in good agreement with a previous study[25] showing that the domains are tightly coupled in the apo state. Analyzing HtrA2$^{S306A}$ in the presence of the DD-PDZOpt peptide yielded decreased values of $29 \pm 20$ ns and $24 \pm 21$ ns for the protease and PDZ

domain, respectively (Supplementary Fig. 17b, d). It must be noted that due to line-broadening and signal overlap fewer resonances could be analyzed compared to the apo state thus the difference is slightly less pronounced if the same residues are compared: $34 \pm 13$ ns and $35 \pm 19$ ns for protease domain and PDZ in the apo state in comparison to $29 \pm 20$ ns and $24 \pm 21$ ns in the peptide-bound state (Supplementary Fig. 17d). Nevertheless, the obtained values are in line with the decoupling of the PDZ domain as evidenced by the on average lower values for this domain. Although these values indicate an enhanced flexibility of this domain, values around 6.5 ns, as obtained for the isolated PDZ domain (Fig. 2d, e), would be expected for a completely decoupled movement, pointing to a still restricted mobility after detachment of the PDZ.

Next, we quantified the micro- to millisecond dynamics of MALVI$^{proS}$-HtrA2$^{S306A}$ by measuring MQ CPMG relaxation rates at 25 and 750 Hz to assess the differences in dynamics of the apo state and the peptide-bound state (Supplementary Fig. 17e–g). On average, we observe $\Delta R_{2\mathrm{eff}}$ values of $8.5\,\mathrm{s}^{-1}$, with helix α1, parts of the regulatory loops in particular the LA loop, and helix α5 showing the largest extent of conformational exchange (Supplementary Fig. 17e). This observation suggests that parts involved in the activation cascade could experience micro- to millisecond dynamics in the apo state in line with our initial hypothesis of an important trigger function of helix α5 within the PDZ as well as the suggested role of helix α1 in the activation cascade (Fig. 5b).

Looking at the MQ rates at 750 Hz, we obtained an average value of $31.8 \pm 14.6\,\mathrm{s}^{-1}$ for the apo state in line with previous results[25], whereas using only a subset of values for the apo state and the peptide-bound state for direct comparison yielded values of $27 \pm 14.7\,\mathrm{s}^{-1}$ and $16.4 \pm 7.8\,\mathrm{s}^{-1}$, respectively. As under these high CPMG pulsing values (750 Hz) only minimal contributions from the exchange between different states with lifetimes larger than -1.3 ms can be expected, this marked difference for the peptide-bound state indicate a decrease on the pico- to nanosecond timescale dynamics in good agreement with the same trend of the $S^2_{\mathrm{axis}}\bullet\tau_C$ values.

We reasoned that structural frustration could be the reason for the observation of conformational exchange contributions within the apo state of HtrA2 as it has lately been suggested to be a potential underlying cause for local microsecond dynamics[26,52]. The analysis of the local structural frustration[53] highlighted several hot spots of highly frustrated regions such as the helices α1 and α5 besides the LA loop. These regions matched well with the identified segments experiencing micro- to millisecond conformational exchange in the apo state as detected by the multi-quantum CPMG experiment (Supplementary Fig. 17h–j).

## Divalent cations are not sufficient for HtrA2 activation

Having established the potential activation cascade of HtrA2 upon DD-PDZOpt binding, we wondered if the interaction with divalent cations would already be sufficient to trigger the first steps of this activation, namely the domain separation. Titrating the different divalent ions towards MALVI$^{proS}$-HtrA2$^{S306A}$ led to localized signal attenuations for all four tested ions but no significant CSPs (Supplementary Fig. 18a, b). Whereas in the case of CaCl$_2$ and MgSO$_4$ no clear indication of the domain detachment was observable, the global signal reduction upon ZnSO$_4$ addition pointed to the formation of a larger HtrA2 species. Addressing this possible oligomerization by SEC-MALS revealed a multimerization $K_D$ ranging from 0.5 to 1 mM for the absence and presence of CaCl$_2$, respectively, in agreement with previous observations[25] (Supplementary Fig. 18c–f and Supplementary Table 5). The $K_D$ could not be determined in the presence of ZnSO$_4$, but the data for the isolated PDZ domain as well as the observed strong signal attenuation in the titration of MALVI$^{proS}$-HtrA2$^{S306A}$ suggest the formation of a multimeric species or even aggregates under these conditions.

To map the binding site(s) of calcium more precisely within full-length mature HtrA2$^{S306A}$, we used gadolinium (Gd$^{2+}$) which can substitute calcium in Ca$^{2+}$ binding sites and whose strong paramagnetic relaxation enhancement (PRE) effect enables mapping of Ca$^{2+}$ binding sites[54,55]. We observed extensive line-broadening of residues 412–430 (comprising the β16 and α7 regions) and residues 329–369 (β14) (Supplementary Fig. 19a, b). Using MIB2, a web-based tool for the prediction of metal binding sites in proteins, on the wild-type HtrA2 structure, we identified potential calcium-binding sites[56]. These sites include as a top hit a region in the α7 helix and the preceding loop region of the PDZ domain comprising residues Q419, Q422, N423, E425, and D426 which correlates well with our gadolinium titration data (Supplementary Fig. 19c). Employing the same in silico approach for identifying theoretical copper binding sites, revealed two potential binding sites partially overlapping with the activator peptide binding groove, in line with our experimental observation that Cu$^{2+}$-ions can be displaced by increasing the activator peptide concentration (Supplementary Fig. 19d, e).

Thus, Ca$^{2+}$ promotes stronger binding of activating peptide/substrate to the PDZ domain, leading to enhanced activation of HtrA2 once in the presence of substrate. This hypothesis is based on (i) our titration data of Ca$^{2+}$ towards full-length HtrA2 did not reveal any of the chemical shift perturbations associated with HtrA2 activation as seen upon the addition of the activating peptide DD-PDZOpt and (ii) our kinetic data from the fluorescence assays suggest that addition of Ca$^{2+}$ leads to an increased affinity to the substrate as evidenced by a reduced $K_D$ value.

## PRE-measurements suggest enhanced mobility of helix α1 upon activation

To further clarify the potential detachment of helix α1 and the contribution of Ca$^{2+}$ to the HtrA2 proteolytic activity, we constructed a single cysteine mutant, HtrA2$^{S306A, S145C}$, enabling the measurement of PREs by side-specific attachment of a paramagnetic spin label (MTSL). We then compared the PRE effects on HtrA2 in sample buffer (no metals), in buffer containing CaCl$_2$ or after the addition of the activating DD-PDZOpt peptide (Fig. 6a–c). We initially used the apo form of HtrA2 to map the basal mobility of helix α1. We observed mainly localized signal attenuations for a subset of resonances, which is consistent with an attached helix α1 as the measured $S^2_{\mathrm{axis}}\bullet\tau_c$ values of methyl-bearing residues 150–160 are en par with the average values observed for the complete protein (Supplementary Fig. 20). Upon closer inspection we mainly observed a PRE-dependent line-broadening for I150 and other residues in the direct vicinity of the MTSL-labeling site besides residues at the outside of the interdomain interface such as I274 and I295 and solvent exposed methyl groups of helix α5 in the PDZ domain (Fig. 6d). These localized effects point to the basal degree of flexibility likely due to movements of the unstructured part of helix α1 preceding I150 thus confirming the closed conformation of HtrA2. Upon addition of Ca$^{2+}$ ions we could observe a more pronounced PRE-induced line-broadening compared to the initial experiment with apo HtrA2 (Fig. 6b, e and Supplementary Fig. 20). We reason that this observation is due to a slight destabilization of the domain interface leading to partial opening as evidenced by additional PRE-dependent line-broadening for helix α5 and the L2 loop, as several residues partially buried at the interdomain interface show PRE effects, e.g., I333 and I373 (Fig. 6e). These more pronounced PRE effects thus clarify the molecular basis of the observed effect of calcium-binding to the PDZ domain: shifting HtrA2 toward a pre-open state, enabling easier access for the activating peptide to its binding site buried at the PDZ:protease domain interface resulting in the observed reduction in dissociation constant (Fig. 3f).

The addition of the activating peptide to HtrA2 leads to notably larger PRE effects. We observed in this state PRE-dependent line-broadening also in extended patches of the PDZ domain, in the

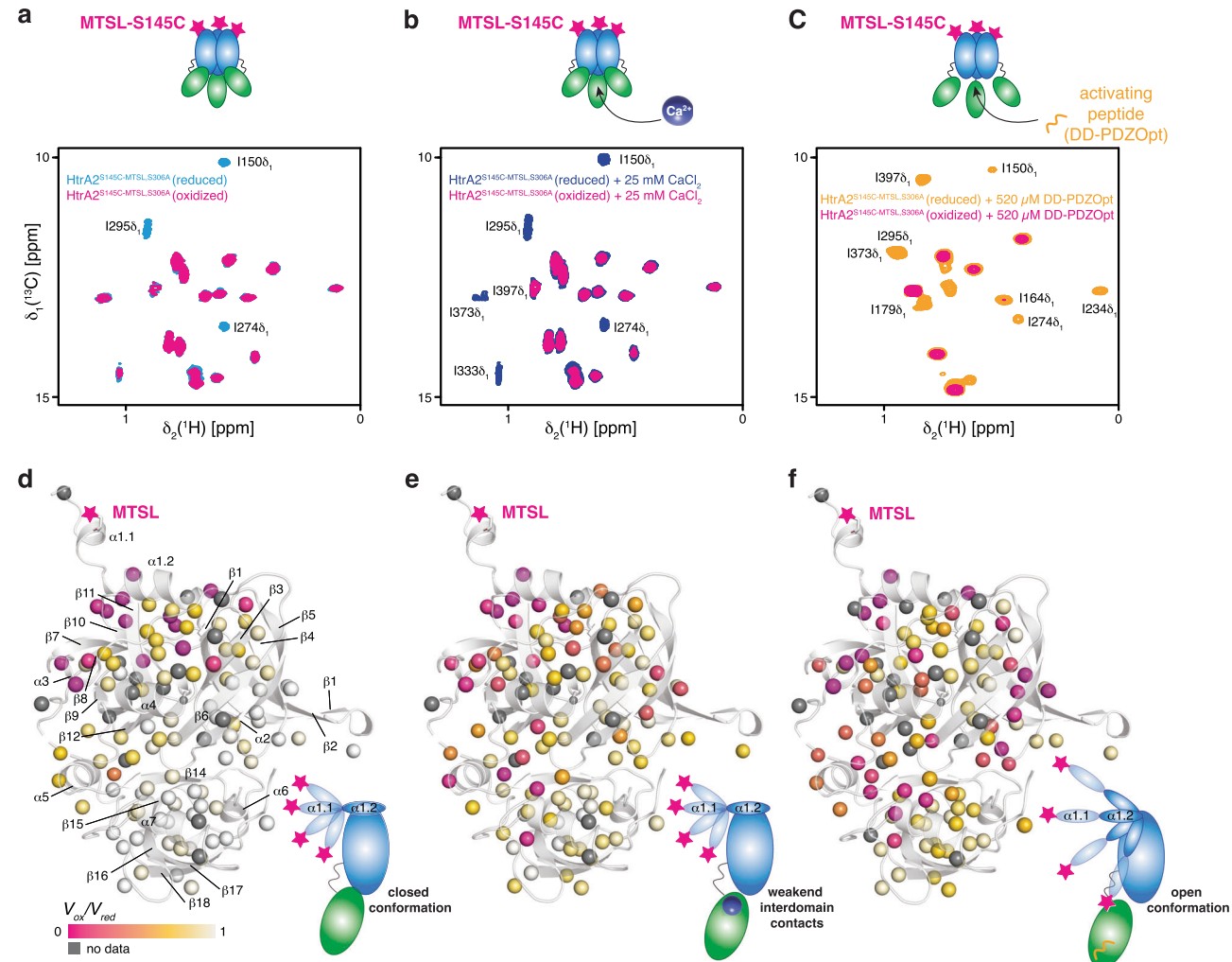

**Fig. 6 | PRE analysis of the mobility of helix α1. a–c** 2D [$^{13}$C,$^1$H]-NMR of the Ile-region of MTSL-spin-labeled MALVI$^{proS}$-HtrA2$^{S306A, S145C}$ in the apo state (blue, **a**), in the presence of 25 mM CaCl$_2$ (dark blue, **b**), and in the presence of 520 μM DD-PDZOpt (yellow, **c**) in the oxidized state and after addition of 5 mM ascorbic acid in the spin label reduced state (magenta). **d–f** PRE effect of a spin label on MALVI$^{proS}$-HtrA2$^{S306A, S145C}$. The PRE intensities detected for the different HtrA2 states (**d–f**) are indicated by the color gradient on the methyl groups depicted as spheres (PDB-ID: 5M3N). Insets illustrate the degree of mobility of the helix α1 under the different conditions.

activator peptide binding cleft (β13–β14), as well as in the protease domain, mainly located to β4–β5 region comprising the LA loop, in close vicinity to the catalytic triad and in the now opened interdomain interface. Thus, the PRE data support the conclusions drawn from our metal titration experiments (Fig. 6c and Supplementary Fig. 20b), which indicate that calcium-binding alone is not sufficient for HtrA2 transition into an active state. Furthermore, our PRE data clearly shows that helix α1 is affected when HtrA2 transitions into an active state upon binding an activating peptide (Fig. 6c, f and Supplementary Fig. 20). Our PRE data shows that the PDZ domain is notably affected when a saturating amount of activating peptide is titrated to HtrA2, evidenced by the large signal attenuations. This suggests that the PDZ domain in the activated state of HtrA2 is spatially closer to the paramagnetic spin label than in the closed conformation. Since the peptide binding cleft, which is closely stacked to the protease domain in the closed conformation, shows among the largest signal attenuations changes (Supplementary Fig. 20c), it suggests that the PDZ domain could shift its position relative to the HtrA2 protease domain converting the protein from a closed inactive to an open conformation. Although our data suggests an increased degree of mobility for helix α1, also the conformational flexibility of the undocked PDZ likely contributes to the observed enhanced PRE effects. This is in line with a splayed outwards conformation of the PDZ towards the amino-terminus that was recently demonstrated for HtrA2 when bound to its natural substrate Apollon and has also been observed previously for a bacterial HtrA protein[57–59]. Although the observed effects are perfectly in line with a proposed model of activation upon detachment of the PDZ from the protease domain, based on recent observations by the Kay group of a more pronounced CSP of the characteristic I150 methyl resonance upon substrate binding to the active site[60], the open state characterized here most likely represents a still catalytically inactive state and only upon substrate binding to the catalytic site the open fully active state can be populated.

## Discussion

It has previously been demonstrated that the proteolytic activity of HtrA2 is modulated both by binding of activating peptides to its regulatory PDZ domain, as well as by its propensity to form inhibitory hexamers[25,44,61]. Here, we show that HtrA2 activity can be additionally modulated by divalent cations via binding to the PDZ domain in the proximity of helices α5 and α7, where metals such as zinc have inhibitory effects in contrast to magnesium and especially calcium leading to an enhanced HtrA2 activity (Fig. 3 and Supplementary Figs. 13 and 14). Our data suggest that binding of metals to the PDZ domain also modulates the dynamics of helix α5 of the PDZ domain, an important structural element docking the PDZ on top of the protease domain in

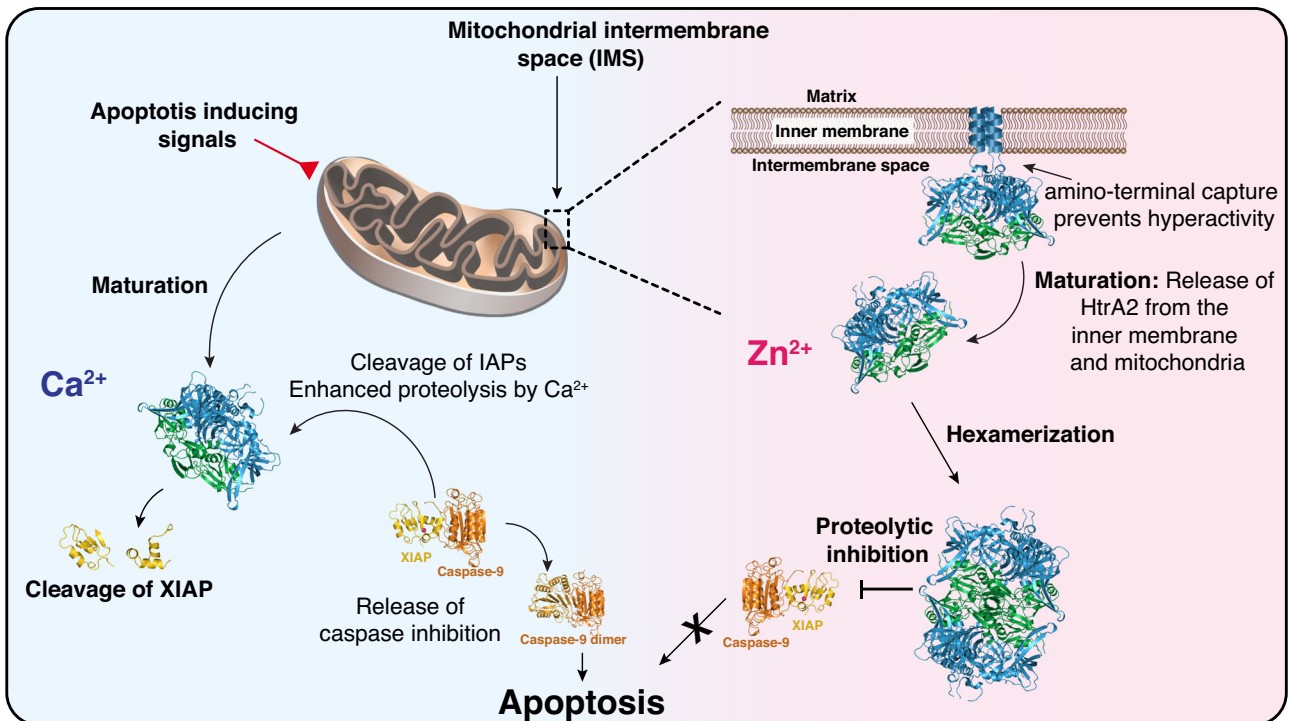

**Fig. 7 | Influence of divalent cations on the functional repertoire of the HtrA2 protease.** In its membrane-attached form, HtrA2 performs protein quality control functions within the mitochondrial intermembrane space (IMS). Upon apoptosis-inducing signals HtrA2 matures, leading to the cleaving of its amino-terminus and secretion from the mitochondrion into the cytoplasm. In the presence of elevated $Zn^{2+}$ levels, the equilibrium of HtrA2 is shifted towards the hexameric inactive state, impairing apoptotic signaling. Under normal apoptotic conditions, the release of HtrA2 from the mitochondrion is coupled to an increase of the cytosolic $Ca^{2+}$ concentration, which further activates the proteolytic function of HtrA2, leading to the cleavage of IAPs and the unleashing of the proapoptotic caspase function, triggering apoptosis. The following crystal structures were used: full-length HtrA2 (PDB-ID: 5M3N), caspase-9:XIAP complex (PDB-ID: 1NW9) as well as the caspase-9 dimer (PDB-ID: 1JXQ).

the non-activated state and potentially acting as a trigger for the domain separation. Calcium ions support the activation of HtrA2 without being themselves sufficient for this important initial step of the activation cascade, as the addition of $CaCl_2$ alone is not resulting in the characteristic CSPs associated with activation of HtrA2 seen in the presence of an activating peptide and the PREs indicate only a partial opening of the binding cleft (Figs. 4a–c and 6 and Supplementary Figs. 18 and 20). Another important feature in this still partially closed complex is the stabilization facilitated by the β14 and β15 in the PDZ domain harboring important methyl-bearing residues locking the closed conformation in solution as identified by us and the Kay lab[25]. Among these residues particularly I393 appears to be a crucial contact point as on the one hand we observe a large change in its rotameric state upon domain separation and on the other hand as mutations in the directly adjacent helix α6, such as G399S or R405W, impair HtrA2 activation completely supposedly freezing HtrA2 in the inactive state[22,62].

The identified basis of the activation of HtrA2 as a multistep event starts with substrate recognition via the PDZ domain, triggering the separation of the PDZ domain and the protease domain, subsequently leading to a rearrangement of the regulatory loops within the protease domain to allow access to the active site (Fig. 5b). A previously not characterized role could be identified for the amino-terminal helix α1 by a local structural rearrangement leading to a possible detachment as evidenced by the extended PRE effects observed for the activator bound HtrA2 (Fig. 6). Even though the exact functional consequences of this feature remain elusive, several previous observations underline the importance of this amino-terminal element: (i) at the far-amino-terminus is an IAP-interacting motif through which supposedly HtrA2 function can be over-activated upon IAP-binding[43,44], (ii) the amino-terminus together with helix α1 are involved in inter-subunit

communication and therefore important for the positive cooperativity of HtrA2[16,30], (iii) some Parkinson's disease associated mutations as well as a putative phosphorylation site reside within this helix[63,64].

The influence of divalent metal ions on the activity of HtrA2 raises the question of which biological mechanisms these metals might be involved in. In the case of calcium, it is established that calcium levels are rapidly increasing during apoptosis and that calcium can accumulate both in the cytosol and inside the mitochondria, either as controlled calcium signaling or due to pathological conditions of the cell[65,66]. Therefore, an involvement of $Ca^{2+}$ in the HtrA2-mediated progression of apoptosis by enhancing the proteolytic activity of HtrA2 against inhibitor-of-apoptosis proteins, such as the well-known HtrA2 target XIAP, seems highly likely[13,67]. Similarly, the inhibitory effect of zinc on HtrA2 activity might also influence the role of HtrA2 in apoptosis. Zinc is well-known for its ability to suppress apoptosis, and it has been observed in breast and pancreatic cancers that malignant cells accumulate zinc[68,69]. Interestingly, zinc homeostasis is also known to be affected in prostate cancers where both too high and too low levels of zinc may increase the risk of developing prostate cancer. Surprisingly, inhibition of HtrA2 has also been shown to induce apoptosis in prostate cancer cell lines in a $Zn^{2+}$-dependent manner together with zinc chelators and a putative drug candidate against prostate cancer, but the exact mechanism remains unclear[70]. Based on our findings in this study, we thus propose a refined mechanism of how HtrA2 activity is modulated by $Ca^{2+}$ and $Zn^{2+}$ ions (Fig. 7).

## Methods
### Protein preparation
A construct for full-length mature human HtrA2[133–458] without the active site serine (S306A) (pET21b-HtrA2) was purchased from

GenScript optimized for *Escherichia coli* codon usage. All other protein constructs were made employing restriction-free cloning[71] (Supplementary Tables 6 and 7). Constructs were expressed in the *E. coli* BL21 (λDE3) strain as amino-terminally SUMO-His₆-tagged proteins in a pET28b vector or carboxy-terminally His₆-tagged proteins in a pET21b vector (Supplementary Table 6). Cells were grown at 310 K in 1 L medium containing either 50 µg/ml kanamycin (pET28b) or 100 µg/ml ampicillin (pET21b) until an OD$_{600}$ 0.6–0.8 was reached, induced with 0.4 mM IPTG and the protein was expressed at 298 K for an additional 16–18 h. Cells were harvested by centrifugation at 277 K, 4500×*g*, 20 min. Cells were subsequently resuspended in 25 ml of purification buffer, 50 mM HEPES pH 7.3, 0.5 M NaCl for HtrA2-PDZ or 50 mM HEPES pH 8, 0.5 M NaCl for full-length HtrA2$^{S306A}$, wild-type HtrA2, and HtrA2$^{S306A, S145C}$, supplemented with 5 mM imidazole. Prior to cell lysis, ½ cOmplete EDTA-free protease inhibitor (Roche), 100 units of DNase (ArcticZymes) and MgCl₂ to a final concentration of 5 mM were added. Protease inhibitor was omitted when purifying wild-type HtrA2. Cells were lysed by three passes through an Emulsiflex (Avestin), and cell debris was removed by centrifugation at 20,000×*g*, 277 K, for 45 min. Proteins were purified with a Ni²⁺-HisTrap HP (GE Healthcare) column using purification buffer and eluted with 500 mM imidazole. The protein fractions were isolated and dialyzed against purification buffer together with human SENP1 (Addgene #16356) for removal of the SUMO-His₆-tag at 277 K overnight[72]. Cleaved SUMO-tag and residual SENP1 were separated from the proteins by running the solution over a Ni²⁺-HisTrap HP column. Collected flow-through and wash fractions of cleaved protein were concentrated using Vivaspin centrifugal concentrations (MWCO: 5000, 10,000, or 50,000; Sartorius) before being applied to gel filtration columns (Superdex 75 increase or Superdex 200 increase, GE Healthcare) equilibrated with PBS pH 7.4 (sample buffer for HtrA2-PDZ), 25 mM Tris-HCl pH 8, 0.1 M NaCl, 1 mM TCEP (sample buffer for full-length HtrA2$^{S306A}$ and wild-type HtrA2) or 25 mM Tris-HCl pH 8, 0.1 M NaCl, 10 mM DTT for HtrA2$^{S145C}$.

### Isotope labeling

Isotope labeling of proteins was achieved by expressing proteins in M9 minimal media with the relevant isotopes supplemented to the media (H₂O supplemented with (¹⁵NH₄)Cl and *D*-(*U*-¹³C)-glucose for uniform double labeling [*U*-¹³C,¹⁵N], or D₂O supplemented with (¹⁵NH₄)Cl yielding [*U*-²H,¹⁵N]-labeled proteins and with (¹⁵NH₄)Cl and *D*-(*U*-²H,¹³C)-glucose yielding [*U*-²H,¹³C,¹⁵N]-labeled proteins[73]. For specific methyl group labeling of ALVIT-PDZ, D₂O-based M9 media supplemented with (¹⁵NH₄)Cl, *D*-(*U*-²H,¹²C)-glucose, 2% of Bioexpress rich medium, 50 mg/L 2-ketobutyric acid-4-¹³C,3,3-*d₂* sodium salt hydrate (isoleucine), 85 mg/L 2-keto-3-(methyl-*d₃*)-butyric acid-4-¹³C₄, 3-d sodium salt (valine/leucine), 50 mg/L [*U*-²H, γ2-¹³CH₃] threonine and 100 mg/L *d₅*-glycine (threonine), and 50 mg/L 2-[²H], 3-[¹³C] L-alanine (alanine) were added 1 h prior to induction[74–76]. For MALVI$^{proS}$–labeled HtrA2$^{S306A}$, we supplemented the medium with (¹⁵NH₄)Cl, *D*-(¹²C,²H)-glucose, 2% (v/v) Bioexpress cell growth media (98% [*U*-²H,¹⁵N]), 50 mg/L 2-ketobutyric acid-4-¹³C,3,3-*d₂* sodium salt hydrate (isoleucine), 50 mg/L [*U*-²H,¹³CH₃] methionine, 50 mg/L 2-[²H], 3-[¹³C] L-alanine, and 4 vials/L DLAM-LV$^{proS}$-kit (2-(¹³C)-methyl-4-(D₃)-acetolactate (valine/leucine *proS* methyl group only) added 1 h prior to induction[77]. For specific LV-HtrA2$^{S306A}$, the medium was supplemented with (¹⁵NH₄)Cl, *D*-(¹³C,²H)-glucose 2-keto-3-(methyl-*d₃*)-butyric acid-1,2,3,4-¹³C₄, 3-²H added 1 h prior to induction. For V-HtrA2$^{S306A}$, the medium was supplemented with (¹⁵NH₄)Cl, *D*-(¹³C,²H)-glucose, 85 mg/L 2-keto-3-(methyl-*d₃*)-butyric acid-4-¹³C₄, 3-d sodium salt and 40 mg/L of L-leucine-*d₁₀* added 1 h prior induction[78].

Bioexpress, threonine, alanine, and glycine were purchased from Cambridge Isotope Laboratories. The DLAM-LV$^{proS}$-kit was purchased from NMR-Bio and all other isotopes from Sigma Aldrich/Merck.

### NMR spectroscopy

NMR experiments were performed at 298 K if not stated otherwise on Bruker Avance III HD 700-, 800-, and 900-MHz spectrometers running Topspin 3.6 (Bruker Biospin), equipped with cryogenically cooled triple-resonance probes. Sequence-specific assignments of the backbone resonances were made using 2D [¹⁵N,¹H]-TROSY-HSQC, 3D HNCO, 3D HNCA and 3D HNCACB as well as 3D CBCACONH experiments for [*U*-¹⁵N,¹³C] HtrA2-PDZ[79,80]. Aliphatic side-chain resonance assignments for HtrA2-PDZ were achieved by 3D (H)C(CCO)NH, 3D H(CCCO)NH, and based on 2D [¹³C, ¹H]-HSQC spectra with/without constant time (CT) version by 3D H(C)CH-TOCSY and 3D (H)CCH-TOCSY experiments on [*U*-¹⁵N,¹³C] HtrA2-PDZ[80]. In addition, the following NOESY-type experiments with the indicated mixing times were performed: 3D ¹H$_{all}$-¹H$_{amide}$-¹⁵N$_{amide}$ SOFAST NOESY with 200-ms mixing time on a [*U*-¹⁵N,¹³C] HtrA2-PDZ sample[81] as well as a 3D ¹³C$_{methyl}$-¹³C$_{methyl}$-¹H$_{methyl}$ HMQC NOESY and a ¹H$_{methyl}$-¹³C$_{methyl}$-¹H$_{methyl}$ NOESY with 600 and 120 ms mixing times on an ALVIT-PDZ sample[82,83], respectively. NMR data were processed using the mddNMR2.6 and NMRPipe software packages[84,85]. Resonance assignments and spectral analyses were done in CARA[86]. Secondary chemical shifts were calculated relative to the random coil values by the POTENCI algorithm[87]. Further, a weighting function with weights 1-2-1 for residues (*i-1*)-i-(*i*+1) was applied to the raw data[38,88]. The chemical shift changes of the amide moiety or the methyl groups were calculated as follows:

$$\Delta\delta(\text{HN}) = \sqrt{\left(\Delta\delta(^1\text{H})\right)^2 + \left(\Delta\delta(^{15}\text{N})/5\right)^2} \quad (1)$$

$$\Delta\delta(\text{HC}) = \sqrt{\left(\Delta\delta(^1\text{H})\right)^2 + \left(\Delta\delta(^{13}\text{C})/4\right)^2} \quad (2)$$

For the quantitative analysis of signal intensities, the amplitudes were corrected by differences in the ¹H-90° pulse length, the number of scans, and the dilution factor[89].

### Methyl group assignments

Based on the determined sequence-specific assignment of the methyl groups of the HtrA2-PDZ reported here together with the recently reported ILVM$^{proR}$-methyl assignment for HtrA2$^{S306A}$[25], we transferred the assignments and subsequently confirmed as well as completed the MALVI$^{proS}$-methyl groups of HtrA2$^{S306A}$ by an experimental approach using V-HtrA2$^{S306A}$ and LV-HtrA2$^{S306A}$ samples to identify leucine and valine residues unambiguously, an HMBC-HMQC experiment of the LV-HtrA2$^{S306A}$ to connect leucine/valine geminal pairs, and a 3D ¹³C$_{Methyl}$–¹³C$_{Methyl}$¹H$_{Methyl}$ SOFAST NOESY experiment of the MALVI$^{proS}$-HtrA2$^{S306A}$ with mixing times of 50 ms and 300 ms, respectively[81,90]. This approach yielded the following large degree of assignment (~92%): Alaβ (21/30), Ileδ₁ (22/22), Leuδ₂ (35/37), Metε (5/5), and Valγ2 (37/37). Methyl NOE interaction patterns for ALVIT-PDZ and MALVI$^{proS}$-HtrA2$^{S306A}$, respectively, were subsequently generated and visualized using Flareplot (https://gpcrviz.github.io/flareplot/).

### Analysis of the rotameric equilibria of the methyl groups

We used the obtained Ile (δ₁) and Met (ε) ¹³C chemical shifts to deduce the rotameric equilibria of the Ile χ₂, and Met χ₃ angles, employing a previously outlined approach[91,92]. As the ¹³C chemical shifts of methyl groups are reportedly directly dependent on the side-chain rotamer, this approach yields insight into the rotameric state of the different methyl groups[93–96]. The population of the *trans* rotameric state ($p_{trans}$) for each residue was calculated according to the chemical shift values of the methyl ¹³C signals ($\delta_{obs}$: ppm)[93,94] using Eqs. (3–4):

$$^{13}C_\varepsilon Met : \delta_{obs} = 15.9 + 3.6\, p_{trans} \quad (3)$$

$$^{13}C_\delta Ile : \delta_{obs} = 9.3 + 5.5\, p_{trans} \quad (4)$$

The equations yielded $p_{trans}$ values ranging from 1 (all *trans*) to 0 (all *gauche (–)*, respectively. In the analysis residues with values ranging between 0.75 and 1 were considered to be in the *trans* conformation and 0–0.25 as to be in the *gauche (–)* conformation.

## NMR backbone dynamics

For the analysis of the dynamic properties of HtrA2-PDZ, the following relaxation experiments were measured: $^{15}N\{^1H\}$-NOE, $R_1(^{15}N)$, $R_{1\rho}(^{15}N)$ and TROSY for rotational correlation times (TRACT) for determining $R_{2\alpha}(^{15}N)$ and $R_{2\beta}(^{15}N)$[97,98]. Steady-state heteronuclear 2D $^{15}N\{^1H\}$-NOE experiments were recorded using a 5 s $^1H$-saturation time for the NOE experiment and the equivalent recovery time for the reference experiment, each preceded by an additional 2.0 s recovery time. $R_1$ measurements were performed with delays of 400, 600, 800, 1200, 1600, 1800, and 2400 ms. $R_{1\rho}$ data measurements were performed by recording delays of 0, 15, 25, 35, 45, 55, and 65 ms. The TROSY ($R_{2\beta}$) and were determined by recording relaxation delay points such as 4, 10, 20, 30, 40 ms as well as 0.4, 2, 4, 8,16 ms, respectively. Nonlinear least square fits of relaxation data were done with PINT[99] using an in-house analysis pipeline, as recently described[100]. $R_{2(R1\rho)}$ ($^{15}N$) values were derived from $R_{1\rho}$ using Eq. (5):

$$R_2 = \frac{R_{1\rho}}{\sin^2\theta} - \frac{R_1}{\tan^2\theta} \tag{5}$$

with $\theta = \tan^{-1}(\omega/\Omega)$, where $\omega$ is the spin-lock field strength (2 kHz) and $\Omega$ is the offset from the $^{15}N$ carrier frequency[88,101].

Error bars for $R_1(^{15}N)$, $R_{1\rho}$ ($^{15}N$), $R_{2\alpha}(^{15}N)$, and $R_{2\beta}(^{15}N)$ were calculated by a Monte Carlo simulation embedded within PINT[99], and for $R_{2(R1\rho)}$ ($^{15}N$) by error propagation. Error bars for the $^{15}N\{^1H\}$-NOE were calculated from the spectral noise. Analysis of the obtained relaxation rates was performed with Tensor2 on the NMRbox web server[102,103]. Residues with an $R_{2(R1\rho)}/R_1$ rate quotient above 30 were removed from the analysis with Tensor2 and a prolate axially symmetric diffusion tensor was used to fit the relaxation data to the HtrA2-PDZ structure (PDB-ID: 1LCY). To quantify the observed exchange contributions additionally, a BEST TROSY $^{15}N$ CPMG experiment using constant time relaxation periods ($T$) of 40 ms employing CPMG frequencies ranging from 25 to 1500 Hz were used at magnetic field strengths of 16.4 T (700 MHz proton frequency) and 21.1 T (900 MHz proton frequency)[36]. $R_{2,eff}$, the effective transverse relaxation rate was calculated according to the following equation:

$$R_{2,eff} = -\left(\frac{1}{T}\right)\ln\left(\frac{I}{I_0}\right) \tag{6}$$

where $I$ (or $I_0$) are the intensities with and without the presence of a constant time relaxation interval of duration $T$, during which a variable number of $^{15}N$ 180° pulses are applied leading to $\nu_{CPMG} = 1/(2\delta)$, where $\delta$ is the time between successive pulses. Data were processed in nmrPipe[66]. Dispersion data were obtained using an in-house extension of PINT 2.1[99], where the respective intensities for each data point at both fields were extracted separately, and subsequently the effective transverse relaxation rates were fitted globally to a two-site exchange model.

## NMR side-chain dynamics

Experiments were performed on a [$U$-$^2H$, Ile-$\delta_1$-$^{13}CH_3$, Leu,Val-$^{13}CH_3$, Ala-$^{13}CH_3$, Thr-$\gamma_2$-$^{13}CH_3$]]–PDZ (ALVIT-PDZ) sample, with stochastic labeling one methyl isotopomer of the valine and leucine groups, at a temperature of 298 K in 99.9% $D_2O$ based NMR buffer[104]. Side-chain methyl order parameters ($S^2_{axis}{}^*\tau_C$) were determined by cross-correlated relaxation experiments[105,106]. Single- (SQ) and triple-quantum (TQ) $^1H$–$^{13}C$ experiments were collected at a series of delay times at a concentration of 500 μM. Ratios of the peak intensities were

determined in PINT[99] and subsequently fitted by in-house written routines for thirteen values ranging between 2 and 45 ms using the following equation where $T$ is the relaxation delay time and $\delta$ a factor to account for coupling due to relaxation with external protons:

$$\left|\frac{I_a}{I_b}\right| = \frac{3}{4}\frac{\eta\tanh\left(\sqrt{\eta^2 + \delta^2}T\right)}{\sqrt{\eta^2 + \delta^2} - \delta\tanh\left(\sqrt{\eta^2 + \delta^2}T\right)} \tag{7}$$

$S^2_{axis}{}^*\tau_C$ values were determined using Eq. (5) from the fitted $\eta$ values and the separately determined rotational correlation times adjusted for the change in viscosity in 100% $D_2O$ ($\tau_c$)[107], for the two different states at 298 K:

$$\eta \approx \frac{9}{10}\left(\frac{\mu_0}{4\pi}\right)\left[P_2(\cos\Theta_{axis,HH})\right]^2\frac{S^2_{axis}\gamma_H^4\hbar^2\tau_c}{r_{HH}^6} \tag{8}$$

where $\mu_0$ is the vacuum permittivity constant, $\gamma_H$ the gyromagnetic ratio of the proton spin, $r_{HH}$ is the distance between pairs of methyl protons (1.813 Å), $S^2_{axis}$ is the generalized order parameter describing the amplitude of motion of the methyl threefold axis, $\Theta_{axis,HH}$ is the angle between the methyl symmetry axis and a vector between a pair of methyl protons (90°), and $P_2(x) = \frac{1}{2}(3x^2\text{-}1)$. Finally, the product of the methyl order parameter and the overall correlation time constant, $S^2_{axis}\bullet\tau_C$, was determined. Multiple quantum (MQ) methyl relaxation dispersion experiments were recorded as a series of 2D datasets using constant time relaxation periods ($T$) of 40 ms (16.4 T (700 MHz)/18.8 T (800 MHz)) and CPMG (Carr-Purcell-Meiboom-Gill) frequencies ranging from 25 to 750 Hz[37]. Data were processed in nmrPipe[66], and peak intensities were extracted with PINT[99]. Dispersion data were fitted numerically to a two-site exchange model using the program ChemEx (available at https://github.com/gbouvignies/chemex/releases).

## Generation of the structural model

The calculation of the structural model between the HtrA2-PDZ and the DD-PDZOpt peptide was based on docking the peptide using the obtained chemical shift perturbation data. For the docking, the HAD-DOCK web server was used[48,49]. For the PDZ structure, the chain A of the crystal structure of the HtrA2-PDZ (PDB-ID: 2PZD) was used, whereas for DD-PDZOpt (DDGQYYFV) an unfolded peptide was generated by using XPLOR-NIH[108]. The active residues were defined as residues bearing methyl groups with a chemical shift change of more than two standard deviations corrected to zero[109] in the titration of DD-PDZOpt to ALVIT-PDZ (Supplementary Fig. 12f) as well as all peptide residues. Passive residues were automatically defined by HADDOCK, and standard docking parameters were used. Haddock clustered 173 out of 200 calculated structures into 11 clusters (Supplementary Table 2).

## Spin labeling of HtrA2$^{S145C}$

HtrA2$^{S145C}$ was purified as described above and stored in 20 mM Tris-HCl pH 8, 0.1 M NaCl supplemented with 10 mM DTT. Spin labeling of HtrA2$^{S145C}$ with MTSL (1-Oxyl-2,2,5,5-tetramethyl-Δ3-pyrroline-3-methyl)-methanethiosulfonate; Toronto Research Chemicals) was done according to published protocols[88,110]. The protein was buffer exchanged into sample buffer without DTT immediately prior to spin labeling using a Zeba spin desalting column 7 kDA MWCO (Thermo-Fisher). A ten-fold molar excess of MTSL dissolved in acetonitrile was added to the protein solution and the mixture was incubated overnight at room temperature in the dark. Unreacted MTSL was subsequently removed by exchanging the buffer to sample buffer (20 mM Tris-HCl pH 8, 0.1 M NaCl). The spin label was reduced by adding ascorbate dissolved in sample buffer (500 mM) directly into the NMR tube to a final concentration of 5 mM.

## Paramagnetic relaxation enhancement (PRE)

A 2D [$^{13}$C,$^1$H]-NMR spectrum of spin-labeled MALVI$^{proS}$-HtrA2$^{S145C, S306A}$ was initially measured in the paramagnetic state, and after the addition of 5 mM final concentration of ascorbate to the sample, the diamagnetic reference was measured. The intensities of well-resolved methyl resonances were extracted at 40 °C with PINT[99]. The experiments were also performed in the presence of 5 mM CaCl$_2$ or DD-PDZOpt.

## SEC-MALS

SEC-MALS experiments were performed using a Superdex Increase 200 10/300 GL column (GE Healthcare) on an Agilent 1260 HPLC Infinity II at RT (~297 K). HtrA2-PDZ experiments were run in PBS pH 7.4 or in HBS (20 mM HEPES, 150 mM NaCl) pH 7.4 when divalent metal ions were supplemented. Experiments with HtrA2$^{S306A}$ were run in HBS pH 7.4. Protein elution was monitored by three detectors in series, namely, an Agilent multi-wavelength absorbance detector (absorbance at 280 nm and 254 nm), a Wyatt miniDAWN TREOS multi-angle light-scattering (MALS) detector, and a Wyatt Optilab rEX differential refractive index (dRI) detector. The column was pre-equilibrated overnight in the running buffer to obtain stable baseline signals from the detectors before data collection. Molar mass, elution concentration, and mass distributions of the samples were calculated using the ASTRA 7.1.3 software (Wyatt Technology). A BSA solution (2–4 mg/ml), purchased from Sigma Aldrich and directly used without further purification, was used to calibrate inter-detector delay volumes, band-broadening corrections, and light-scattering detector normalization using standard protocols within ASTRA 7.1.3. To assess the oligomerization state of HtrA2-PDZ and HtrA2$^{S306A}$, increasing elution concentrations of the proteins were used to fit the dissociation constant ($K_D$) assuming a fast monomer–dimer equilibrium:

$$M_W = 2M - M \frac{-K_D + \sqrt{K_D^2 + 8[M]K_D}}{4[M]} \qquad (9)$$

where $M$ is the molecular mass of the monomer and $[M]$ the molar concentration of the sample (in terms of monomer) as it passes through the MALS detector after eluting form the column[111]. The concentration was obtained by the absorbance signal after band-broadening correction, using the MALS detector as the reference instrument. In all, 95% confidence intervals were determined from the fitting error in GraphPad Prism version 9.1.0.

## Gadolinium titration

To identify the calcium-binding sites in full-length HtrA2 we resorted to exploiting the relaxation enhancement effect of paramagnetic gadolinium (Gd$^{3+}$)[54]. Gd$^{3+}$ can be readily substituted into Ca$^{2+}$ binding sites, thus providing a detailed picture of the interacting residues[55,112]. We titrated increasing amounts of gadolinium(III)-trifluoromethanesulfonate (Gd(OTF)$_3$; (Merck)) to MALVI$^{proS}$-HtrA2$^{S306A}$ and measured at each step a 2D [$^{13}$C,$^1$H]-NMR spectrum. Data were processed in nmrPipe[66] and peak intensities were extracted with PINT[99].

## Cleavage assay

Overall, 1 mg/mL β-casein (Sigma Aldrich) was incubated with 623 nM (monomer concentration) of HtrA2 at 310 K in assay buffer (20 mM HEPES pH 7.4) supplemented with 2 mM of either MgSO$_4$, CaCl$_2$, CuCl$_2$, or ZnSO$_4$, to a total volume of 101 μL. Samples were taken at 10-, 20-, 40- and 60-min intervals and the cleavage patterns were analyzed by SDS-PAGE using 4–20% MiniProtean® TGX™ (Bio-Rad) gels stained with SimplyBlue SafeStain (Thermo Scientific). Quantification analysis of band intensities was made using the Image Lab software (Bio-Rad).

## Fluorescence cleavage assays

The cleavage assay with varying DD-PDZOpt concentrations was performed using 20 μM of the fluorescent substrate peptide H2Opt (Mca-IRRVSYSF{Lys(Dnp)}KK (GenScript)) with 510 nM wild-type HtrA2 (monomeric concentration) in 20 mM HEPES pH 7.4 supplemented with 1 mM EDTA and increasing activator peptide concentrations (DD-PDZOpt: DDGQYYFV (GenScript)) at 313 K. For assays including 2 mM divalent metal ions, EDTA was omitted from the buffer. The reactions were monitored using a FLUOstar Optima plate reader using $\lambda_{ex}$: 320 nm and $\lambda_{em}$: 405 nm. The concentration of cleaved product from fluorescence intensity was determined by calculating the fluorescence intensity difference between an internal control of uncleaved H2Opt and the maximum fluorescence intensity obtained at complete peptide cleavage during the assay, which could then be used to determine the amount of cleaved peptide. Catalytic rates were calculated by extraction of initial reaction rates from the fluorescence assay curves. Calculations of the steady-state kinetic parameters $v_{max}$ and $K_m$ were performed by fitting the data to the Michaelis–Menten equation (substrate vs. reaction rate) with GraphPad Prism version 9.1.0 (Eq. (10)):

$$v = \frac{v_{max}[peptide]}{K_m + [peptide]} \qquad (10)$$

where $v$ denotes the reaction velocity, and $[peptide]$ denotes the DD-PDZOpt concentration. Following a previously outlined approach[25], we fitted the cleavage rate of H2Opt as a function of the DD-PDZOpt concentration (5, 10, 25, 50, 75, 100, 150, 200, and 300 μM) to a standard one-site binding model approximating the total peptide concentration by the free peptide concentration:

$$k = \left( \frac{k_{max}[DD - PDZOpt]}{K_{D,app} + [DD - PDZOpt]} \right) + k_0 \qquad (11)$$

where $[DD - PDZOpt]$ denotes the activating peptide concentration, $k$ the cleavage rate, $k_0$ the basal substrate cleavage rate and $k_{max}$ the maximum cleavage rate in the fully peptide-bound form, $K_{D,app}$ is the apparent dissociation constant.

The assay with varying substrate concentration was carried out using 510 nM of wild-type HtrA2 (monomeric concentration), 50 μM DD-PDZOpt and five concentrations of the H2Opt peptide (0.5, 5, 10, 20, and 30 μM). Buffer and temperature conditions as well as plate reader settings were as stated above for the activation peptide concentration variation assay. Calculations of the steady-state kinetic parameters $v_{max}$ and $K_m$ were determined using the webtool InterferENZY[113]. The $k_{cat}$ value was calculated as shown in Eq. (12).

$$k_{cat} = \frac{v_{max}}{[HtrA2]} \qquad (12)$$

where $[HtrA2]$ denotes the trimeric concentration of HtrA2 used in the kinetic assay. All variants of the fluorescence cleavage assays were run in triplicates yielding similar results.

## Reporting summary

Further information on research design is available in the Nature Portfolio Reporting Summary linked to this article.

# Data availability

All data needed to evaluate the conclusions in the paper are present in the manuscript and/or the Supplementary Materials. Uncropped gel images are provided in a Source Data file. The sequence-specific NMR resonance assignment for HtrA2-PDZ was deposited in the BioMagResBank (www.bmrb.wisc.edu) under accession code 51320 The NMR data used for the relaxation analysis, chemical shift perturbations, and

NOE networks have been tabulated and are available alongside the fluorescence data and the structural model generated by HADDOCK on Mendeley data [https://doi.org/10.17632/fgpcmp2kkg.1]. Structure coordinates used in this study are available for the RCSB Protein Data Bank (https://www.rcsb.org) under accession codes 1JXG, 1LCY, 1NW9, 2PZD, and 5M3N. Source data are provided with this paper.

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

## Acknowledgements

The Swedish NMR Centre of the University of Gothenburg is acknowledged for spectrometer time. The authors thank H. Fremlén for help with graphic design. B.M.B. gratefully acknowledges an EMBO Young Investigator Fellowship as well as funding from the Swedish Research Council (Starting Grant 2016-04721; Consolidator Grant 2020-00466), the Swedish Cancer Foundation (2019-0415 and 2022-2490), and the Knut och Alice Wallenberg Foundation through a Wallenberg Academy Fellowship (2016.0163 and 2020.0300) as well as through the Wallenberg Centre for Molecular and Translational Medicine, University of Gothenburg, Sweden. This study made use of NMRbox: National Center for Biomolecular NMR Data Processing and Analysis, a Biomedical Technology Research Resource (BTRR), which is supported by NIH grant P41GM111135 (NIGMS).

## Author contributions

B.M.B. conceived the study and designed the experiments together with E.E.A. E.E.A. performed all experimental work. E.E.A., J.L., and B.M.B. analyzed and discussed the data. E.E.A., J.L., and B.M.B. wrote jointly the manuscript.

## Funding

## Competing interests

The authors declare no competing interests.
