## [Peer Review File · Nature Communications]

Structural basis of substrate recognition and allosteric activation of the pro-apoptotic mitochondrial HtrA2 protease.REVIEWER COMMENTS

Reviewer #1 (Remarks to the Author):

The manuscript focuses on the in-depth structural analysis the PDZ domain of human HtrA2 protease. Using solution NMR and HtrA2 enzyme activity assays, in the presence of activating peptides and divalent cations, the authors pinpoint key structural elements in the PDZ domain relevant for HtrA2 activation by divalent metals and by the HtrA2 activating peptide - DD-PDZOp. Finally, the authors show that interaction between full-length HtrA2 (S306A active site mutant) with DD-PDZOp in the PDZ domain results in the rearrangements of loops L2-L3-LD-L1, followed by changes in residues located in $\beta 7$ and $\alpha 1$ of the catalytic domain. The manuscript includes a thorough study, combining NMR studies with SEC-MALS and enzyme activity assays, with the aim to identify correlations between enzyme activation by metal ions and by DD-PDZOp, changes in HtrA2 oligomerization and structural rearrangements. The mechanisms underlying activation of the full-length protein by metal ions remain unclear in the current version of the manuscript. Also, it is not discussed if or how the structural rearrangements alter the HtrA2 catalytic triad.

Overall, the presentation of all the results on the manuscript would benefit from a stronger and clearer focus on the structural changes induced by DD-PDZOp that result in enzyme activation.

It is not clear why the authors only compare their data with the structure of the catalytically inactive full-length HtrA2 (S306A active site mutant). The structure of HtrA2 with the wild-type catalytic triad has been determined (doi: 10.1038/cddis.2017.487) and showed a rearrangement of the regulatory loops leading to small changes in the contacts between the PDZ and the protease domains both within each subunit and across the inter-subunit interfaces. These changes affect the LD loop and the first helix of the PDZ domain (corresponding to $\alpha 5$ in this manuscript)

Additionally, there are a few issues that should be revised to improve the current version of the manuscript:

1)In Figure 1a, it would be informative to include the secondary structure elements that are discussed throughout the manuscript. The amino acid sequence and secondary structure/activation loops could also be presented in the supplementary information;

2)In Figure 1a and 1e the white labels are difficult to read;

3)Figure 1b is not very informative. Maybe a close-up of the interactions between the PDZ and the catalytic domain would be helpful for readers who are not familiar with HtrA2 structure; moreover, to discuss allostery, the interactions across subunits should be presented and discussed;

4)Could the structure of the isolated PDZ domain shown in Figure 1e be superposed with the PDZ domain in the structure of full-length HtrA2? Are differences identified when the structures of the catalytically inactive (S306A) and the active enzymes are compared?

5)In the enzyme activity assays shown in Figure 5, controls without HtrA2 are missing to evaluate if the metal ions affect beta-casein degradation;

6)Concerning the effects of metal ions on PDZ domain and full-length HtrA2 oligomerization, it is unclear why ions like copper and zinc are tested. Have the authors seen if the tested ions modify PDZ or HtrA2 folding and secondary structure leading to protein aggregation? How are the authors distinguishing protein oligomerization from unspecific aggregation?

7)In the materials and methods section, the preparation of catalytically inactive HtrA2 is described. However, the description of the expression and purification of the active enzyme used in enzyme activity assays is missing.

Reviewer #2 (Remarks to the Author):

The manuscript describes a study on HtrA2, a protease linked to severe human neurological diseases. Multimerization, the allosteric mechanism and activation of HtrA2 is linked to a PDZ domain, which plays an important role in substrate binding and oligomerization. Studies on dynamics of the PDZ domain are central in the current study.

An interesting finding in the current study is that divalent metal ions, suggested to be linked to HtrA2 activity before, play a crucial role in HtrA2 oligomerization and activity. Missing however, how, where and how strong the metals bind to the PDZ domains, how this leads oligomerization, nature of oligomer interactions, as well as explanations how mutations related to PDZ may disrupt HtrA2 activities. Though PDZ dynamics has been analyzed in atomic detail, the overall description condenses to a global oligomerization effect of PDZ domains, for which it has not been made clear that the detailed understanding of PDZ dynamics was necessary. The NMR, metal binding, and HtrA2 metal (in)activation studies are certainly done thoroughly. However, In the current manuscript these read very much as unrelated observations: an integral picture is missing.

Additional comments:

Fig. 1 needs clarifications

1.c indicated secondary structure cartoon, xray or nmr-based?

1.d relevance?.

1.f Tauc, S2 and Rex variations largely coincide. A product S2 x Tauc appears more constant, casting doubts on splitting them. Could the Tauc variance be due to an incorrect relaxation model, and thus the Rex in reality even larger than indicated?

1.f shows limited variation in S2 order parameter, but much more differences in Rex (and Tauc..). Suggestion: figure with Rex over the PDZ structure may be more informative on local dynamics than the relatively small variations in S2.

1.9 description in text lacking.

The data in Fig. S2c and S2d are very nice! Therefore I am a bit surprised by the low data quality in the R1 measurements in S2a.

Fig. S2.a Any explanation why R1 error bars are so large and why there are fewer R1 data than HetNOE data? Generally R1's are easier and more reliable to measure than hetNOEs. Accurate R1's would also assist in R1rho analysis of eqn. 3 to compensate their R1 contributions due to off-resonance effects.

Global R2 effects in Fig. 2a 283 K. Could this be aggregation? What would be the concentration dependence?

Fig. S2d. Given Tauc 7.9 ns (data Fig. 1f), any explanation that in S2.d S2 x Tauc (ns) can be higher than this? Fraction of dimers with a dimerization Kd of 13 mM, can only be small.

Fig. S6i Significance of fit questionable: data points only beginning of the fitted curve. Curve cannot be checked as an appropriate model. How errors in data affect fitted parameters can not be judged. Was it a constrained fit?

Fig. S9c and Fig. S1d. It is not clear, what is meant with 'affected methyls being affected'. Clarify: did NOEs disappear, become weaker, due to what?

Signal attenuation due to Cu²⁺ can be understood, given its paramagnetic properties, at least globally, but why locally, and why for Zn²⁺ in a similar region? Idem Fig. S5b,d,f and S10e,f. How were the dotted lines for dimer and monomer MMs established?

Metal binding: Concentration of metal ions? Affinities?

Is it binding, or are noted effects in NMR indirect, due to aggregation? Evidence of dimerization? Clarify better the effect of metal binding on oligomerization: directly between oligomers or

indirectly (how?).

Minor:

MALS 'dimerization': bit confusing term after oligimerization into inactive hexameric forms of HtrA2. Is it indeed dimerization of PDZ monomers into dimers?

Fig. S2.f CSP of amideprotons vs temperature. Appears not discussed. Is there a non-trivial explanation? Else skip.

Possible effect of Zn²⁺ on HtrA2: check DOI:10.1002/pros.22922

Fig. S5 a,c,e,g and Fig S10c,d dotted lines not visible as dotted

Reviewer #3 (Remarks to the Author):

This paper describes studies of the protein HtrA2, an allosteric protease, using solution-state NMR, biochemistry and functional assays.

I have taken much time to evaluate this paper in pretty much detail, and, overall, my impression is not good. The paper certainly represents a lot of work. It required lengthy assignments (including methyls), dynamics measurements, many titrations, functional assays, SEC-MALS etc. This is positive.

But the issue I see is that the story that derives from the data is not clear, and I mean this two-fold. First, it is not quite clear to me what is the take-home message: dimerization? Metal-interaction? Dynamics? And second, I think that the conclusions drawn from the data do not necessarily derive from the data, and in some instances data are in contradiction with one another. I furthermore see a number of instances where the paper has technical problems, and experiments that need to be re-done or more thoroughly analysed.

In my list below, I try to address the various main "topics"/statements that are made in the paper. Then I have a section on technical issues, then a section on readability of figures and text, and lastly a number of smaller comments.

I think that the paper needs to be completely re-thought and rewritten. In the current status I do not recommend Nat. Comm. to publish the paper, and I would encourage the authors not to submit it elsewhere in the present state.

I am sorry not to be more positive about it, as I realize the very substantial work that has gone into this study. I just think that it is a pity publishing it in this form.

Dimerization.

It is found by SEC-MALS (Supplementary Figure 3) that the affinity for dimer formation is very weak, with a K_d of at least 13 mM. Up to a concentration of 0.9 mM, i.e. higher than what was used in most NMR experiments, the population of dimer in equilibrium with monomer is 6% at most.

The authors then also look at this question of dimer formation by methyl-directed relaxation measurements, which provide the product of local methyl order parameter and overall correlation time constant (S2.tau_c). They did this experiment at 0.5 mM and 0.8 mM, and report that the values at 0.8 mM are about 1.5 times higher (overall) than at 0.5 mM. Based on these findings they conclude that "This points to an increased rotational correlation time of about 20–30% for the isolated HtrA2-PDZs, that can be readily attributed to the transient dimer formation, thus perfectly in line with the weak interactions observed for the full-length HtrA2 protein 24."

I disagree with their assessment of the dimer equilibrium, and that their assessment must be incorrect is even in their own data: looking at the SEC-MALS data (Supp Fig 3A), which show a very modest concentration dependent, and on the analysis thereof (Supp Fig 3B), it is clear that increasing the concentration from 0.5 to 0.8 mM can never lead to effects as large as what they derive from the methyl-relaxation measurements. Looking at Supp Fig 3B I would estimate that the population of dimer increases by about 2%. Yet, their methyl-relaxation data point to a +50% increase of $S_2 \times \tau_c$. Note that this sentence here (page 13) is not correct:

"This points to an increased rotational correlation time of about 20–30% for the isolated HtrA2-PDZs " The increase is 50%, not 20-30%. And again, we compare these 50% to a 2 or 3 % from SEC-MALS.

Something is here really off, and my conclusion is that the experimental data contradict each other.

Possibilities that I see are:

- Errors in the execution or analysis of the methyl-relaxation data
- Viscosity effects in the NMR sample

Have the authors seen any concentration-dependent chemical-shift changes? If there is a dimer-formation that is as strong as the methyl relaxation data imply (i.e. more than an order of magnitude larger than what SEC-MALS says), then there shall be chemical-shift changes in the dimerization interface.

I would recommend that the authors investigate the origins of their discrepancies and possibly eliminate the dimer-part altogether or do additional experiments.

The proposed allosteric effects upon peptide binding:

The section " Elucidating the allosteric regulation [...]" aims to decipher how peptide binding induces structural changes in full-length HtrA2. The section concludes saying "In summary, these different sets of CSPs throughout the HtrA2 protein enabled us to decipher distinctive steps underlying HtrA2 activation " The steps that the authors propose are depicted in Fig. 4E.

I am not convinced that this conclusion can be deduced from the data. In this section, the following data are presented:

- Addition of DD-PDZOpt to full-length protein and observation of methyl chemical-shift changes. The chemical shift changes are spread across the protein E.g. residues 154, 158, 420, located on opposite sides of the protein.
- A methyl-methyl NOESY spectrum of MALVIproS- HtrA2S306A in the absence of the peptide. To my understanding, this sample, namely HtrA2S306A without peptide, is the one for which a crystal structure is already available (Fig. 1B, PDB 1LCY). Thus, I do not quite see the added benefit of collecting a NOESY spectrum. (I assume that the NOESYs are in agreement with the structure. Please explicitly state this.)

What has not been recorded, but what would have been much more interesting is a NOESY spectrum with the peptide. From such an experiment, and ideally from a proper structure calculation, one might derive a structure of the complex with the peptide, and build mechanistic models upon it.

Currently, a rather elaborate mechanistic model (Fig. 4E) is built essentially on chemical-shift changes, which are spread all over. Maybe their model is right, but I wonder where it comes from, and I do not really see how it may be built upon the observations.

Consequently, I naturally disagree with the opening sentence of the paragraph entitled "Divalent cations are not sufficient for HtrA2 activation ", which is: "Having established the activation cascade of HtrA2 upon DD-PDZOpt binding ", and in the Discussion section the model of Figure 4E is again discussed, and I still think there that this model does not really derive from the data.

The discussion states that "helix a1 which rearranges or even detaches from the rest of the protein", but I do not see data that support this assessment. I assume the authors refer to chemical-shift changes upon addition of peptide (data reported in Figure 4C, which are rooted in data reported in Supp Figure 9A). The only data there are that the peaks become undetectable. This is certainly interesting, and possibly it indicates something like a detachment, but it could be other things. It should, at the least, be made clear what the assessment is based on, and also

make sure that a non-NMR-expert can understand that assumptions and possible other interpretations.

Effects of metal ions:

This study has analysed how metals affect HtrA2 from multiple angles: (i) ^1H - ^{15}N and methyl ^1H - ^{13}C NMR chemical shift changes upon addition of metals to the PDZ domain (Supp Fig 4A-E), (ii) Changes in dynamics in PDZ (result: no changes in dynamics as seen by CPMG RD experiment; stated on page 14 and Supp Fig 4F-H), (iii) analysis of dimerization equilibrium by SEC-MALS (Supp Fig 5; Fig 3C; Supp Fig 10 C-F for full-length HtrA2), (iv) casein proteolysis experiments with the full-length HtrA2 (Fig 3E-G); as outlined in my technical criticism in a section below, I do not find the activity data convincing, (v) NMR chemical-shift data with full-length HtrA2 (Supp Fig. 10 A+B).

The conclusions about metal interaction and possible "activation" of the protein are as follows:

1) two parts of the PDZ domain (in isolation) bind metals, around residue 385-390 and around 420. (Supp Fig 4A-D). This is particularly visible for the case of Ca^{2+} . Importantly, this effect does not seem to be found for methyl groups of the isolated PDZ, which suggest that there are no rearrangements of the tertiary structure, and it is not found in methyl-detected spectra of the full-length protein.

Cu^{2+} has effects, but they are presumably trivial solvent-PRE effects.

2) the dynamics of PDZ does not seem to change (CPMG data)

3) Zn, but not Ca, Cu or Mg, induces dimer formation of the PDZ domain

4) Cu and Zn lead to abolition of enzymatic activity (Fig 3B). In light of the fact that the NMR methyl signals in these two states are also very weak (Supp Fig 10B) I wonder if this may be trivially explained by aggregation of the protein.

5) NMR data of the full-length protein (Supp Fig 10) do not allow to make clear statements on whether/where metal ions bind. The methyl spectra do not show any shifts. But one may argue that even in the isolated PDZ the methyls do not show shifts. But maybe they do show intensity increase upon addition of Ca (just like for the backbone, Supp Fig 4C)? How can one explain intensity increases in the backbone (i.e. smaller R_2 for the backbone), but intensity decreases (larger R_2) for the methyls?

Overall, I have a hard time making sense of the metal-ion data. Something happens to the loop preceding α_5 (Ca binding in PDZ alone, but maybe not in full-length?) and there is a Ca activation of full-length protein, but the link is not clear. And there is clearly no tertiary-structure rearrangement.

In the paragraph "Divalent cations are not sufficient for HtrA2 activation" it is stated that "the global signal reduction upon ZnSO_4 addition pointed to oligomerization". I am not sure that this interpretation can be derived from the data. In Supp. Figure 10B it seems indeed that effectively all residues have strongly reduced intensity. But then a few lines below it is also said that the protein aggregates in the presence of zinc. There are always multiple reasons why a protein might aggregate, and I don't think that one can learn much from it.

The role of helix α_5 as a "switch":

The authors propose that helix α_5 may act as a "switch" (stated explicitly in the Discussion), and that it is "loosely attached" (page 10).

For me it was difficult to know what is meant here. Why would helix α_5 be a switch? A switch is something that changes when doing something to the protein (e.g. when adding ions). But it does not seem to bind

Open technical points that need clarification:

I am not very positive about the enzymatic assays and their analysis and different points need to be addressed there:

(1) Most striking to me was Supp Fig 7E. Clearly, increasing the concentration of an activating peptide does not change anything to the activity within error bars. Nonetheless, a Michaelis-Menten type model was fitted to the data (right). The model does provide some values, which are also stated there, but I think they are meaningless. This kind of fit makes sense only if one can actually see a dependency on concentration. There also does not seem to be any correlation between the raw data of Supp Fig 7C and 7E. 7C shows a clear concentration dependence, 7E does not. To my understanding these two panels supposedly go together.

(2) What is the initial drop in fluorescence? I assume that it has to do with equilibration. Please clarify. And what is the region of this curve that has been used for the fits?

(3) the data shown in Supp Fig. 7D and 7F are supposed to report similar things, where 7F shows the velocities of the fluorescence traces. (What exactly? Initial slope?) When looking at 7D I find essentially no catalytic activity (rates even negative for some!) for all concentrations up to 100 μM , and positive slopes only for the three highest concentrations. Very clearly, this is not what panel 7F shows. As said above for the fit of panel 7E, I would also be very cautious with the fit of 7F.

(4) The fits shown in Fig 3E are, in my opinion, equally problematic. Fitting the concentration dependence of Fig 3C (with CaCl_2) simply makes no sense, and even without addition (top panel) I would question the data. As an aside, the error bar of v_{max} (0.37 M/s) is incorrect; I assume it should read as 0.37×10^{-9} M/s.

To conclude, the casein data must be redone totally, in my opinion, as triplicates at least, and making clear what is done exactly.

In Supplementary Figure 3A, it is rather surprising that the MALS-estimated mass is low at the beginning of the SEC peak and then increases. Generally, the first part of a SEC peak contains the highest-molecular-weight species. This is also seen in SEC-MALS profiles in Supplementary Figure 10C. How can this be explained? And consequently, how shall one understand the masses that are used for the determination of the self-association process (Supp Fig. 3B and 5).

Along these lines, please provide error estimates for the molecular weights in the above-mentioned graphs (Supp Fig 3B etc).

The NMR spectra of proteins as large as HtrA2 are what they are, and generally have signal overlap, even when using advanced isotope-labeling schemes. It is clear from Supp Figure 8B that even with the best-resolved labeling scheme (Val-proS) there is overlap. For example, the peaks of the following residues appear clearly overlapped: 266, 379, 325, 210, 243, 270, 441, 179, 393, and several more. This is at least the impression one gets from Supp Fig. 8B. Yet, data for all these residues are plotted in Supp Fig 9. In my view, the data of Supp Figure 9A are, thus, to be taken with caution. Could the authors please provide figures that more convincingly show how one can obtain data for these residues from 2D spectra?

Note that Val325-gamma2 is assigned to two different peaks (!), which is physically impossible.

The authors use NOESY data to propose that helix alpha-5 is somewhat "flexibly attached". The analysis in the present state is not really convincing to me, and should be done more rigorously. Generally speaking, a lack of a NOE peak does not really unambiguously establish that two sites are far apart. The peak may be of weak amplitude, e.g. because of line broadening of the peak. The analysis does not seem to take into consideration the distances, or at least it is not shown.

The secondary chemical shifts are used to propose that beta-strand 14 is populated only to "~50 %". It is not clear to me how this estimation has been done. The analysis of CA-CB secondary chemical shifts has been popular some time ago, but I would argue that it is more precise to take into consideration as many shifts as possible and use more elaborate analyses, such as CheSPI

(Mulder & co) or TALOS-N (Bax & co).

In Figure 1D I see just a few bars, some pointing to b-strand, some not, but it is not clear to me how the authors obtained any estimate of the population of b-strand from such data.

Figure 3C (dimer population) needs error bars. I would claim that the differences in populations are negligible for all except the one with Zn. Likewise Fig 3B and 3F need error bars.

Something is odd in supp Figure 4D, where the chemical-shift perturbation is plotted onto the structure. The second one, with Ca, is red at many residues, but when looking at the raw data (panel B) this does not seem to match. For example, I see many red spheres in helix alpha5, but the CSPs are very small. As a comparison, in other panels (e.g. the first one, with Mg) I see much larger shifts than in the second panel, e.g. for residues 385-390, yet in the structure there is not a single red sphere. It is not a very good sign if the raw data are essentially decoupled from their final representation.

Further comments for clarification:

Page 17 states: "Upon CuCl₂ addition, we observed an inhibition of the activity of HtrA2, but this effect could be partially alleviated by drastically increasing the activator peptide concentration, a tendency not observed for Zn²⁺ (Supplementary Fig. 7c-f). This behavior is consistent with our SEC-MALS data, which indicates that the suppression mechanism exhibited by Zn²⁺ on the HtrA2 activity differs from the Cu²⁺ inhibitory mechanism. "

I have a hard time following their reasoning. Besides the fact that I am not convinced by any of the activity data (Supp Fig 7C-F), as outlined above (Open technical points), I do not quite follow what they mean here. Zn leads to more dimerization according to Fig S5. And how is this in line with the finding that using more activator peptide can NOT lead to activation in the presence of Zn? This needs more explanations.

The Discussion section would greatly benefit from making clear and unambiguous references to what the authors mean when they refer to their findings. E.g. it is stated that there is a "switch region encompassing helix a5, where metals such as zinc have inhibitory effects". As stated above, it is not clear on which data this assessment is based; there are no chemical-shift changes and no changes in CPMG data in alpha5 with metals. If the Discussion makes this point, it must be much clearer, in my opinion.

The authors use the term "slow timescale". This is sometimes seen in the literature but in my view is poor practice. A time scale can be long or short (and it needs to be specified relative to what it is long or short), but a time scale cannot be "slow".

Please state at the beginning of section "Elucidating the allosteric regulation of HtrA2S306A using methyl-TROSY" that this sample is trimeric and corresponds to the structure shown in Figure 1B, or clarify otherwise.

The following publication shall be cited in the context of the activation of HtrA proteins (and their PDZ domain) by peptides M. Meltzer et al., "Allosteric activation of HtrA protease DegP by stress signals during bacterial protein quality control," *Angew. Chemie - Int. Ed.*, vol. 47, no. 7, pp. 1332-1334, 2008, doi: 10.1002/anie.200703273.

It is very difficult for the reader to evaluate statements that the data are in "perfect agreement"

with previous data. This is used e.g. on page 17: "Using the fluorescence data, we calculated a k_{cat}/K_m of 6,500 M⁻¹ s⁻¹ in perfect agreement with previous reports 46 " Please be specific what this means, and avoid words like "perfect agreement", as it is certainly not "perfect".

Make figures more readable:

It would be extremely helpful for the reader to systematically indicate the location of secondary structure elements in plots such as Supp Fig. 4B, 4G, 4H, 6C, 6F, 9A, 10B, and Fig. 2E, and possibly there are similar instances. I found it difficult to follow the arguments put forward in the text that make reference to certain structure elements, without being able to immediately see those elements in the plots. A reader who actually cares currently constantly needs to switch between several figures to understand the naming and location of residues/secondary structures. Similarly, it would also be tremendously useful to indicate the secondary structure element names in representations of the structure, such as in Fig. 4C and Supp Fig 4D (and maybe others). Additionally, I found it difficult to read the graphical representation of Fig 4C, because the spheres are floating freely in space. For example, I tried to find the contact between alpha1 and beta7, e.g. to M260, which is supposed to be in beta7. But in the figure the sphere called "M260" floats somewhere close to the helix, and I do not see any beta-strand in the vicinity.

Figure panels such as Supp Fig 10 b are almost useless, because they are too small. What exactly can I learn e.g. about the intensity drop induced by the presence of Zn? I see a general drop everywhere (presumably the protein just aggregates, according to the main text), to a bar that is barely one millimeter long.

If the authors want to make a point from these data, then they should be plotted such that they are visible, AND plot the effects on the structure. If not, then they could just as well be dropped. Supp Fig 10 has another problem: the color code of panel b is not clear. I can understand that the purple plot refers to Mg, the red one to Zn (well, I guess so), but then what are the two blue ones? The order of plots is clearly changed between panels A and B, and the two blue-tones in A are not the same ones as those in B, so it is impossible to know what is Cu and what is Ca. Based on the intensity drop, I assume that the third panel in B is the fourth panel of A (Cu).

Please make more axis ticks on residue-wise plots. Currently there are ticks every 50 (!) residues e.g. in Figure 1. This makes it really tedious to find out the identity of residue belonging to the different bars. Please make sure that this is improved, eg Fig 1d 1f, 2a, and possibly others.

Figure 4D and Supplementary Figures 9B and 9C are not clear to me. What we are seeing are NOE contacts between methyls, shown as lines in the Flare-plots. So far so good. And then we additionally see somehow (not clear to me whether this can be seen in the Flare plots) those residues that have chemical-shift perturbation upon addition of peptide. (Is this encoded in the color? Can one read in this figure which of the two methyls that is connected by a line has a significant CSP?) What this figure implies is that the NOE network changes because a given methyl group has a chemical-shift perturbation. (Is this right?) But this implicit conclusion is not necessarily valid. Even if the structure does not undergo significant changes that would lead to disruption of contacts between methyls, one may have CSP. Chemical shifts are just rather complicated. To me it is unclear how the NOE network in the apo state (could basically be obtained from the crystal structure) together with CSPs (which do not directly translate to structure) can tell us anything about structural changes in the complex or of "allosteric coupling of the activating loops" (caption 4d).

In plots of effects on the structure (e.g. Fig 4C), only the residues with significant effects are shown but not all residues. It would be useful to see where methyl probes are which do NOT have effects.

Figure 2C: the axis label is wrong. Either it is an exchange rate constant (with units of inverse seconds or inverse milliseconds), or it is an exchange time constants (in seconds or milliseconds). Now it says that it is a rate constant, but the unit is ms.

The population, plotted in the "ski" plot below needs error bars. I am not sure I understand the reason why the rate constant is shown as bars, and the populations as "skis". Similarly, why not show bar plots in all other occurrences where currently these "skis" are used?

REVIEWER COMMENTS **our comments**

We would like to thank the reviewers for their careful reading of the manuscript and their extremely constructive comments to improve the work. We have studied each point raised by the reviewers and have revised the manuscript accordingly. Furthermore, we have conducted additional experiments to validate our mechanistic hypothesis, which is now significantly more founded by experiments compared to our previous submission. Below is our detailed point-by-point response to the reviewer's comments.

Reviewer #1 (Remarks to the Author):

The manuscript focuses on the in-depth structural analysis the PDZ domain of human HtrA2 protease. Using solution NMR and HtrA2 enzyme activity assays, in the presence of activating peptides and divalent cations, the authors pinpoint key structural elements in the PDZ domain relevant for HtrA2 activation by divalent metals and by the HtrA2 activating peptide - DD-PDZOp. Finally, the authors show that interaction between full-length HtrA2 (S306A active site mutant) with DD-PDZOp in the PDZ domain results in the rearrangements of loops L2–L3–LD–L1, followed by changes in residues located in $\beta 7$ and $\alpha 1$ of the catalytic domain. The manuscript includes a thorough study, combining NMR studies with SEC-MALS and enzyme activity assays, with the aim to identify correlations between enzyme activation by metal ions and by DD-PDZOp, changes in HtrA2 oligomerization and structural rearrangements.

The mechanisms underlying activation of the full-length protein by metal ions remain unclear in the current version of the manuscript. Also, it is not discussed if or how the structural rearrangements alter the HtrA2 catalytic triad.

We appreciate the reviewer to pointing out some unclarities in our previous version of the MS. We believe that particularly our new PRE data provide now mechanistic insight into the role of calcium, facilitating higher affinity to the activating peptide by partially opening the binding interface between the PDZ and protease domain. We also have now included a brief discussion about how the catalytic triad might be affected, as we only have insight into the effects of adjacent methyl groups and cannot observe the participating amino acids directly, we used the following phrasing discussing the effect of calcium ions:

Upon addition of Ca^{2+} ions we could observe a more pronounced PRE-induced line-broadening compared to the initial experiment with apo HtrA2 (**Fig. 6b, e** and **Supplementary Fig. S20**). Based on the observed effects we reason that this observation is due to a slight destabilization of at the PDZ:protease domain interface resulting in the observed reduction in dissociation constant (**Fig. 3f**).

Addition of the activating peptide to HtrA2 lead to notably larger PRE effects. We observed in this state PRE-dependent line-broadening also in extended patches of the PDZ domain, in the activator peptide binding cleft ($\beta 13$ – $\beta 14$), as well as in the protease domain, mainly located to $\beta 4$ – $\beta 5$ region comprising the LA loop, in close vicinity to the catalytic triad and in the now opened interdomain interface. Thus, the PRE data support the conclusions drawn from our metal titration experiments (**Fig. 6c** and

Supplementary Figure S20b), which indicate that calcium binding alone is not sufficient for HtrA2 transition into an active state. ...

And we also state our indirect conclusions about changes in the vicinity of the active site based on the rotameric state analysis and the NOE analysis:

Furthermore, two isoleucine residues in the proximity of the active site, I164 and I166, had an ~20% increased *gauche* (-) population, pointing to local subtle structural adaptations underlying HtrA2 activation (**Fig. 4f, g**).

Other regions of changes in NOE patterns were in the vicinity of the regulatory loops suggesting together with the observed subtle adaptations of the side-chain rotamers a switch from the inactive to active conformation around the catalytic site (**Supplementary Fig. 16c**).

Overall, the presentation of all the results on the manuscript would benefit from a stronger and clearer focus on the structural changes induced by DD-PDZOp that result in enzyme activation.

We thank the reviewer for this valuable suggestion. We have followed the advice of the reviewer and now included a more detailed analysis of the structural consequences induced upon DD-PDZOp binding by providing a structural model of the PDZ peptide complex using HADDOCK (Suppl. Fig. S12) as well as analyzing the methyl NOEs for full-length HtrA2 now also in the peptide-bound state (Suppl. Figure. S16).

It is not clear why the authors only compare their data with the structure of the catalytically inactive full-length HtrA2 (S306A active site mutant). The structure of HtrA2 with the wild-type catalytic triad has been determined (doi: 10.1038/cddis.2017.487) and showed a rearrangement of the regulatory loops leading to small changes in the contacts between the PDZ and the protease domains both within each subunit and across the inter-subunit interfaces. These changes affect the LD loop and the first helix of the PDZ domain (corresponding to $\alpha 5$ in this manuscript)

We appreciate the reviewer for pointing us to the subtle differences between the inactive and active structures, which we now address in the newly introduced Supplementary Figure S1 and added the following to this figure legend:

Supplementary Figure S1. Amino acid sequence and structure of the mature form of human HtrA2 (residues 134-458). **a)** Secondary structure elements are indicated as well as the activation loops LA (blue), LD (dark blue), L3 (purple), L1 (magenta) and L2 (pink). Residues H198, D228 and S306 which comprise the active site of HtrA2 are shown in gold. The start of the PDZ-domain is indicated by the green arrow. **b)** Complete trimeric assembly of HtrA2 (PDB-ID: 5M3N) with catalytic residues shown in gold and the individual domains in

blue and green, respectively. **c)** Isolated monomer of HtrA2 (PDB-ID: 5M3N) with the regulatory loops and catalytic residues indicated by the color gradient used in panel **a**. **d, e)** Overlay of the catalytic active HtrA2 (PDB-ID: 5M3N) and the catalytically inactive HtrA2^{S306A} (PDB-ID: 1LCY) showing subtle local adaptations besides larger deviations in the active site (**d**) and the PDZ domain protease domain interface (**e**). For a detailed discussion of the differences between the wild-type HtrA2 and the catalytically inactive HtrA2^{S306A} variant the reader is referred to Merski *et al.* ¹.

While analyzing the changes observed in the chemical shifts of the methyl groups and discussing our observation, we refer now to these changes illustrated in Suppl. Fig. 1:

We observed significant chemical shift changes within the regulatory loops, pointing to an activation cascade of L2-L3-LD-L1 leading to a correct arrangement of the catalytic triad upon binding of the activating peptide to the PDZ domain (**Fig. 5a and Supplementary Fig. 16b, c**) ²⁷. In addition to the CSPs, line-broadening of residues in helix α 1 of the protease domain was observed in a similar manner as for the PDZ-domain protease domain interface, which is in general indicative for either conformational heterogeneity and/or micro- to millisecond dynamics.

Additionally, there are a few issues that should be revised to improve the current version of the manuscript:

1) In Figure 1a, it would be informative to include the secondary structure elements that are discussed throughout the manuscript. The amino acid sequence and secondary structure/activation loops could also be presented in the supplementary information;

We thank the reviewer for pointing out the lack of information. We have now added this important information into Supplementary Figure S1.

2) In Figure 1a and 1e the white labels are difficult to read;

We increased in our opinion the visibility of the labels in the current version and hope to have addressed this issue adequately.

3) Figure 1b is not very informative. Maybe a close-up of the interactions between the PDZ and the catalytic domain would be helpful for readers who are not familiar with HtrA2 structure; moreover, to discuss allostery, the interactions across subunits should be presented and discussed;

We agree and thus redesigned the figure panel accordingly. The full-length structure of the trimer is now shown in Supplementary Figure 1b.

4) Could the structure of the isolated PDZ domain shown in Figure 1e be superposed with the PDZ domain in the structure of full-length HtrA2? Are differences identified when the structures of the catalytically inactive (S306A) and the active enzymes are compared?

We believe to have addressed this in the context of comparing the active and inactive structures now shown in supplemental Figure S1.

5) In the enzyme activity assays shown in Figure 5, controls without HtrA2 are missing to evaluate if the metal ions affect beta-casein degradation;

We apologize missing these in important controls in our initial version. We have added them now in Supplemental Figure S10 showing no effects on beta-casein stability by the addition of the different ions.

6) Concerning the effects of metal ions on PDZ domain and full-length HtrA2 oligomerization, it is unclear why ions like copper and zinc are tested. Have the authors seen if the tested ions modify PDZ or HtrA2 folding and secondary structure leading to protein aggregation? How are the authors distinguishing protein oligomerization from unspecific aggregation?

We thank the reviewer for pointing out that we missed the reasoning for testing the different ions in for their effect on HtrA2. We believe to have made this now clearer by adjusting the respective passage in the text:

We hypothesized that due to HtrA2s role in the apoptotic signaling its proteolytic function might be modulated by divalent cations, such as Ca^{2+} , as bacterial HtrA proteins have previously been shown to be modulated by divalent ions^{39,40}.

We then extended our analysis by also testing Cu^{2+} and Zn^{2+} , previously reported as inhibitors for bacterial HtrAs^{39,40}, we detected similar chemical shift perturbations towards HtrA2-PDZ (**Supplementary Fig. S6a–d**).

Based on our NMR titrations we did not observe any obvious changes to neither the isolated PDZ as well as the full-length HtrA2, although we cannot rule out any influence on HtrA2 folding as we did not experimentally assess this question. We believe to have clarified and amended our analysis of the effects of the metal ions also providing affinities. Whereas calcium and copper have distinct binding sites showing opposing effects, activation, or inhibition, respectively, the presence of zinc seems to be a more indirect effect leading to multimerization.

7) In the materials and methods section, the preparation of catalytically inactive HtrA2 is described. However, the description of the expression and purification of the active enzyme used in enzyme activity assays is missing.

We have now added details for the wild-type HtrA2 as well as the now also used HtrA2^{S306A}, ^{S145C} variants.

Reviewer #2 (Remarks to the Author):

The manuscript describes a study on HtrA2, a protease linked to severe human neurological diseases. Multimerization, the allosteric mechanism and activation of HtrA2 is linked to a PDZ domain, which plays an important role in substrate binding and oligomerization. Studies on dynamics of the PDZ domain are central in the current study.

An interesting finding in the current study is that divalent metal ions, suggested to be linked to HtrA2 activity before, play a crucial role in HtrA2 oligomerization and activity.

We thank the reviewer for his/her appreciation of our work and findings.

Missing however, how, where and how strong the metals bind to the PDZ domains, how this leads oligomerization, nature of oligomer interactions, as well as explanations how mutations related to PDZ may disrupt HtrA2 activities.

We thank the reviewer for pointing us to not appropriately connecting these effects of metal ions into a functional picture by lacking important information in the initial submission. We believe to have addressed the raised issues and also discuss known mutations in the HtrA2-PDZ, which localize at the PDZ:protease domain interface in the context of our results:

Another important feature in this still partially closed complex is the stabilization facilitated by the β 14 and β 15 in the PDZ domain harboring important methyl bearing residues locking the closed conformation in solution as identified by us and the Kay lab ²⁵. Among these residues particularly I393 appears to be a crucial contact point as on the one hand we observe a large change in its rotameric state upon domain separation and on the other hand as mutations in the directly adjacent helix α 6, such as G399S or R405W, impair HtrA2 activation completely supposedly freezing HtrA2 in the inactive state ^{22,62}.

Though PDZ dynamics has been analyzed in atomic detail, the overall description condenses to a global oligomerization effect of PDZ domains, for which it has not been made clear that the detailed understanding of PDZ dynamics was necessary.

We agree with the reviewer that our initial submission lacked proper connection of these important observations failing to connect them to a coherent picture. We hope to have addressed this important point now by rewriting large parts of the manuscript outlining in essence that effects on the PDZ facilitate activation of the protease domain by allosteric signaling through the entire domain even leading to a modulation of the dynamics of the amino-terminal helix α 1 in line with our proposed detachment.

The NMR, metal binding, and HtrA2 metal (in)activation studies are certainly done thoroughly. However, In the current manuscript these read very much as unrelated observations: an integral picture is missing.

We appreciate the general supportive words of the reviewer. We believe that we have now clarified particularly the role of metal binding and the analysis of the HtrA2 in the presence of the activating peptide in our view now enabled us to address the valid point of criticism that in the initial version of the MS did not connect the different sets of data sufficiently well. We hope by rewriting large parts of the MS, we have now succeeded to develop a red thread throughout the MS, connecting the different observation into an integral picture of the activation of HtrA2.

Additional comments:

Fig. 1 needs clarifications

1.c indicated secondary structure cartoon, xray or nmr-based?

We have now stated in our view clearly in the figure legend that the indicated secondary structure is x-ray based as we believe the reviewer is referring to 1d.

1.d relevance?.

We believe this comment refers to panel 1e. We believe that this panel is relevant as it indicates regions of which are broadened beyond detection in the ^{15}N -NMR spectrum, which can be attributed as a first indication of micro- to millisecond dynamics in these regions. Nevertheless, we partially redesigned Figure 1 and thus moved this panel to Supplementary Fig. S2.

1.f Tauc, S2 and Rex variations largely coincide. A product S2 x Tauc appears more constant, casting doubts on splitting them. Could the Tauc variance be due to an incorrect relaxation model, and thus the Rex in reality even larger than indicated?

We thank the reviewer for sharing his/her valuable thoughts. As outlined below in more detail, we have now re-analyzed the complete backbone relaxation data. To investigate the possibility of errors in our analysis we also used different high resolution structures of the PDZ providing highly similar results, so that we believe to be able to exclude larger Rex values as raised by the reviewer.

1.f shows limited variation in S2 order parameter, but much more differences in Rex (and Tauc..). Suggestion: figure with Rex over the PDZ structure may be more informative on local dynamics than the relatively small variations in S2.

We agree with the reviewer and now also show in the reassembled Figure 1 also the R_{ex} contributions plotted on the structure.

1.9 description in text lacking.

We believe that the reviewer was referring to Figure 1g, which we according to the previous point now moved panel 1e and show it alongside the R_{ex} contribution (now Fig. 1g). Nevertheless, we believe that a brief amendment in the main text regarding this particular panel is sufficient:

We observed that the generalized order parameter values with an average of 0.9 ± 0.1 (**Fig. 1e, f**), indicating structural rigidity for the whole domain including the loosely attached helix $\alpha 5$ on the pico- to nanosecond timescale. Nevertheless, we observed increased R_{ex} terms for residues in the $\beta 15$ preceding linker region, as well as in $\alpha 6$ and the region involving $\alpha 7$ and the linker region between $\alpha 7$ and $\beta 17$, pointing to the presence of chemical exchange contributions as initially observed by the enhanced $R_{2\beta}$ rates (**Fig. 1f, g**).

Alongside some additional info in the figure legend:

e) Generalized order parameter (S^2) reporting on sub-nanosecond motions plotted on the HtrA2-PDZ structure. The backbone amide moieties of the HtrA2-PDZ domain are shown as spheres and the S^2 values are indicated by the yellow-to-blue gradient. **f)** Analysis of ^{15}N -backbone relaxation data with the Lipari-Szabo model-free approach. The generalized order parameter S^2 reports on pico- to nanosecond motions, the chemical exchange contributions R_{ex} indicating micro- to millisecond motions, and the rotational correlation time τ_c . Broken line represents the average value of 9.7 ns. **g)** The amide moieties

of the HtrA2-PDZ domain are shown as spheres the calculated R_{ex} is indicated by the yellow-to-red gradient.

The data in Fig. S2c and S2d are very nice! Therefore I am a bit surprised by the low data quality in the R1 measurements in S2a.

Fig. S2.a Any explanation why R1 error bars are so large and why there are fewer R1 data than HetNOE data? Generally R1's are easier and more reliable to measure than hetNOEs. Accurate R1's would also assist in R1rho analysis of eqn. 3 to compensate their R1 contributions due to off-resonance effects.

We thank the reviewer to pointing us to this issue. We have now rerun the whole relaxation analysis using an optimized in-house pipeline (Kawale & Burmann Structure 2021) compare to our initial analysis and subsequently rechecked all the input data leading to more consistent values.

Global R2 effects in Fig. 2a 283 K. Could this be aggregation? What would be the concentration dependence?

Even though we did not perform temperature depended SEC-MALS experiments, elution profiles on a standard SEC run in the cold room (~8°C) were virtually identical ruling out any obvious aggregation effects under the lower temperature conditions.

Fig. S2d. Given τ_{auc} 7.9 ns (data Fig. 1f), any explanation that in S2.d $S_2 \times \tau_{auc}$ (ns) can be higher than this? Fraction of dimers with a dimerization K_D of 13 mM, can only be small.

We completely agree with the reviewer and as we have written in our answer to the same issue raised by reviewer #3 we realized an issue in the data analysis afterwards leading us to larger $S_2 \times \tau_{auc}$ values so that currently there is only a marginal increase in the values going to 800 μ M so that we decided to drop this data and only report the SEC derived data and shorten the whole section addressing the dimerization.

Fig. S6i Significance of fit questionable: data points only beginning of the fitted curve. Curve cannot be checked as an appropriate model. How errors in data affect fitted parameters can not be judged. Was it a constrained fit?

We were a bit unsure why the reviewer pointed out this particular SEC-MALS analysis (now panels I and j in Supplemental Figure S11). We used for all data presented in the manuscript an approach previously used by us and others (e.g. Morgado et al. Nat. Comms 2017; Benfield et al. J. Biol. Chem. 2011). We also point clearly towards the limitations of the approach providing only lower limits of K_D values in the main text:

Although the accessible range of protein concentrations was limited by solubility, MALS data for dimerization can be fitted with constrained values for the titration end point masses and lower limits of K_D values can be obtained from such solubility limited data sets³⁸.

We outline the used approach in more detail in the methods section:

To assess the oligomerization state of HtrA2-PDZ and HtrA2^{S306A}, increasing elution concentrations of the proteins were used to fit the dissociation constant (K_D) assuming a fast monomer–dimer equilibrium:

$$M_W = 2M - M \frac{-K_D + \sqrt{K_D^2 + 8[M]K_D}}{4[M]} \quad (9)$$

where M is the molecular mass of the monomer and $[M]$ the molar concentration of the sample (in terms of monomer) as it passes through the MALS detector after eluting from the column¹¹⁰. The concentration was obtained by the absorbance signal after band-broadening correction, using the MALS detector as the reference instrument. 95% confidence intervals were determined from the fitting error in GraphPad Prism version 9.1.0.

And regarding the question of the reviewer about a constrained fit: We only constrained the molecular masses to the theoretical values for the different species (monomer and dimer for the PDZ-domain or trimer and hexamer for the full-length HtrA2) and provided an initial starting value of the K_D .

Fig. S9c and Fig. S1d. It is not clear, what is meant with 'affected methyls being affected'. Clarify: did NOEs disappear, become weaker, due to what?

We agree that this was ambiguously phrased and referring to chemical shift changes and thus inferring changes in NOE patters. As we now have included in addition the analysis of methyl NOEs in the peptide bound state we have a clearer view on the changes in NOE patterns, which we present in Supplementary Figure S16.

Signal attenuation due to Cu²⁺ can be understood, given its paramagnetic properties, at least globally, but why locally, and why for Zn²⁺ in a similar region?

We have now added a more detailed analysis of the metal ions, which enabled us to identify in a first step experimentally the calcium binding site, which we also confirmed with in silico analysis. The same type of in silico analysis provided two potential copper binding sites, which overlap partially with the activator peptide binding region, thus providing a feasible rationale for our observation that the copper ions can be displaced by high activator peptide present.

Idem Fig. S5b,d,f and S10e,f.

How were the dotted lines for dimer and monomer MMs established?

We now clarify in the figures that the dashed lines represent the theoretical monomer and dimer masses for the PDZ or respectively the theoretical trimer and hexamer masses for the full-length protein.

Metal binding: Concentration of metal ions? Affinities?

Is it binding, or are noted effects in NMR indirect, due to aggregation? Evidence of dimerization? Clarify better the effect of metal binding on oligomerization: directly between oligomers or indirectly (how?).

As outlined in one of the previous points, we believe to have clarified and amended our analysis of the effects of the metal ions also providing affinities. Whereas calcium and copper have distinct binding sites showing opposing effects, activation, or inhibition, respectively, the presence of zinc seems to be a more indirect effect leading to multimerization.

Minor:

MALS 'dimerization': bit confusing term after oligimerization into inactive hexameric forms of HtrA2. Is it indeed dimerization of PDZ monomers into dimers?

We agree that the used wording in this context might be misleading, therefore we have rephrased this section:

Under the chosen experimental conditions, the HtrA2-PDZ domain showed only a minute tendency to dimerize with an estimated lower limit of a K_D of 13 ± 1.65 mM (**Supplementary Fig. S5c, d**). This weak transient interaction is about three orders of magnitude weaker than the recently reported **trimer-to-hexamer** equilibrium for the full-length protein by NMR spectroscopy with ~ 20 μ M²⁵. **This difference can be likely attributed to the presumably cooperative nature of the interaction between the three PDZ:PDZ interfaces within the hexameric species**²⁵.

Fig. S2.f CSP of amideprotons vs temperature. Appears not discussed. Is there a non-trivial explanation? Else skip.

We agree that it is only supplementary information, and it is only included to show that we were able to simply transfer the assignment from the higher temperature without the need of running 3D assignment experiments therefore we have now added a reference to this item to the text and choose to keep this panel.

Possible effect of Zn²⁺ on HtrA2: check DOI:10.1002/pros.22922

We thank the reviewer for pointing us to this relevant paper. We have now included the reference in our discussion around the divalent cations:

Interestingly, zinc homeostasis is also known to be affected in prostate cancers where both too high and too low levels of zinc may increase the risk of developing prostate cancer. Surprisingly, inhibition of HtrA2 has also been shown to induce apoptosis in prostate cancer cell lines in a Zn²⁺-dependent manner together with zinc chelators and a putative drug candidate against prostate cancer, but the exact mechanism remains unclear⁶⁹.

Fig. S5 a,c,e,g and Fig S10c,d dotted lines not visible as dotted

Agreed that the size and density of the individual data points was too high so that it appears as a solid line, therefore we have adjusted the figure representation accordingly.

Reviewer #3 (Remarks to the Author):

This paper describes studies of the protein HtrA2, an allosteric protease, using solution-state NMR, biochemistry and functional assays.

I have taken much time to evaluate this paper in pretty much detail, and, overall, my impression is not good.

The paper certainly represents a lot of work. It required lengthy assignments (including methyls), dynamics measurements, many titrations, functional assays, SEC-MALS etc. This is positive.

We thank the reviewer for his/her appreciation of our efforts and the work put into the manuscript. Particularly his/her detailed comments of our short comings in the initial version of the MS were extremely helpful for us, which we hope to have now appropriately addressed as outlined in detail in the next points.

But the issue I see is that the story that derives from the data is not clear, and I mean this two-fold. First, it is not quite clear to me what is the take-home message: dimerization? Metal-interaction? Dynamics?

We hope to have now clarified this crucial point that all the experiments lead to the crucial rule in starting the activation cascade of HtrA2 from the binding of an activator/substrate to the PDZ domain and how this initial step can be modulated for example by partially opening the binding cleft for the peptide through calcium leading to higher affinity of the activator peptide and in consequence to faster cleavage under lower activator peptide concentrations. We believe although our initial mechanistic model has not been altered it is now significantly better founded on the additional experimental data clarifying in detail the role of calcium, that it only modulates the activator binding and thus indirectly activates the proteolytic activity of HtrA2 as well as the PRE data confirming the larger degree of flexibility in the activator bound state were vital additions.

And second, I think that the conclusions drawn from the data do not necessarily derive from the data, and in some instances data are in contradiction with one another.

As pointed out in the previous point, we have a significant amount of new experimental data in line with our previous mechanistic suggestions, so that we believe that our conclusions are now well supported by our data and that we were able to resolve contradictions within the previous version of the MS.

I furthermore see a number of instances where the paper has technical problems, and experiments that need to be re-done or more thoroughly analysed.

We agree with the reviewer and apologize for our short comings in the previous version. We have now reanalyzed large parts of the initial data and spotted for example an analysis error in the context of the dimerization so that we now dropped large parts of this point and believe that our data is now in itself consistent as well as consistent with our proposed activation model.

In my list below, I try to address the various main “topics”/statements that are made in the paper. Then I have a section on technical issues, then a section on readability of figures and text, and lastly a number of smaller comments.

I think that the paper needs to be completely re-thought and rewritten. In the current status I do not recommend Nat. Comm. to publish the paper, and I would encourage the authors not to submit it elsewhere in the present state.

Following the detailed comments of all three reviewers and as we believe addressing them hopefully in a complete and convincing manner that the present status of the MS is now significantly advanced making it clearer and more accessible for the reader.

I am sorry not to be more positive about it, as I realize the very substantial work that has gone into this study. I just think that it is a pity publishing it in this form.

Despite the valid points of criticism, we highly appreciate the highly constructive manner of this reviewers' comments, which enabled us to greatly improve the MS in our view.

Dimerization.

It is found by SEC-MALS (Supplementary Figure 3) that the affinity for dimer formation is very weak, with a K_d of at least 13 mM. Up to a concentration of 0.9 mM, i.e. higher than what was used in most NMR experiments, the population of dimer in equilibrium with monomer is 6% at most.

The authors then also look at this question of dimer formation by methyl-directed relaxation measurements, which provide the product of local methyl order parameter and overall correlation time constant ($S2 \cdot \tau_c$). They did this experiment at 0.5 mM and 0.8 mM, and report that the values at 0.8 mM are about 1.5 times higher (overall) than at 0.5 mM. Based on these findings they conclude that “This points to an increased rotational correlation time of about 20–30% for the isolated HtrA2-PDZs, that can be readily attributed to the transient dimer formation, thus perfectly in line with the weak interactions observed for the full-length HtrA2 protein 24.”

I disagree with their assessment of the dimer equilibrium, and that their assessment must be incorrect is even in their own data: looking at the SEC-MALS data (Supp Fig 3A), which show a very modest concentration dependent, and on the analysis thereof (Supp Fig 3B), it is clear that increasing the concentration from 0.5 to 0.8 mM can never lead to effects as large as what they derive from the methyl-relaxation measurements. Looking at Supp Fig 3B I would estimate that the population of dimer increases by about 2%. Yet, their methyl-relaxation data point to a +50% increase of $S2 \cdot \tau_c$. Note that this sentence here (page 13) is not correct: “This points to an increased rotational correlation time of about 20–30% for the isolated HtrA2-PDZs “ The increase is 50%, not 20-30%. And again, we compare these 50% to a 2 or 3 % from SEC-MALS.

Something is here really off, and my conclusion is that the experimental data contradict each other. Possibilities that I see are:

- Errors in the execution or analysis of the methyl-relaxation data
- Viscosity effects in the NMR sample

We agree with the reviewer that the data about dimerization in the mentioned section did not add up in the previous version of the MS. Although we observed some tiny concentration dependent CSPs, we realized that there was an error in the original $S2 \cdot \tau_c$ data analysis at the higher concentration and remeasuring indicated only a tiny concentration dependent modulation of these, so we decided to shorten the whole section and only include the SEC-MALS data in the MS. We apologize for initially not spotting this mistake:

Previously, a hexameric species of HtrA2 has been characterized as a closed inactive state with reduced substrate affinity compared to the trimeric species of HtrA2²⁵. Based on this and the signal attenuation observed when titrating $ZnSO_4$ towards HtrA2-PDZ, we reasoned that the different divalent ions might influence the oligomeric equilibrium of HtrA2. Therefore, we performed SEC-MALS experiments with the HtrA2-PDZ domain in the presence of the four metal ions used in the proteolytic assay. While the addition of Mg^{2+} , Cu^{2+} , and Ca^{2+} did not result in a notably increased dimerization rate of the HtrA2-PDZ domain compared to non-supplemented buffer, addition of Zn^{2+} markedly increased the propensity ~50-fold with which the HtrA2-PDZ domains dimerized (**Supplementary Fig. S11 and Supplementary Table S1**).

Have the authors seen any concentration-dependent chemical-shift changes? If there is a dimer-formation that is as strong as the methyl relaxation data imply (i.e. more than an order of magnitude larger than what SEC-MALS says), then there shall be chemical-shift changes in the dimerization interface.

I would recommend that the authors investigate the origins of their discrepancies and possibly eliminate the dimer-part altogether or do additional experiments.

As stated, in our previous point, we followed the recommendation of the reviewer and only included the SEC-MALS part of this section into the current MS.

The proposed allosteric effects upon peptide binding:

The section “ Elucidating the allosteric regulation [...]” aims to decipher how peptide binding induces structural changes in full-length HtrA2. The section concludes saying “In summary, these different sets of CSPs throughout the HtrA2 protein enabled us to decipher distinctive steps underlying HtrA2 activation “ The steps that the authors propose are depicted in Fig. 4E. I am not convinced that this conclusion can be deduced from the data. In this section, the following data are presented:

- Addition of DD-PDZOpt to full-length protein and observation of methyl chemical-shift changes. The chemical shift changes are spread across the protein E.g. residues 154, 158, 420, located on opposite sides of the protein.
- A methyl-methyl NOESY spectrum of MALVI_{proS}- HtrA2S306A in the absence of the peptide. To my understanding, this sample, namely HtrA2S306A without peptide, is the one for which a crystal structure is already available (Fig. 1B, PDB 1LCY). Thus, I do not quite see the added benefit of collecting a NOESY spectrum. (I assume that the NOESYs are in agreement with the structure. Please explicitly state this.)

We apologize for missing this important statement and following the reviewer’s suggestion we added the following:

To obtain a **more detailed** picture of the functional consequences of DD-PDZOpt binding to the PDZ domain we **next** characterized the methyl-methyl NOEs within the MALVI^{proS}-HtrA2^{S306A} in the absence of the **activating** peptide (**Supplementary Figure 16b**). **The identified NOEs were consistent with the previously reported crystal structures of HtrA2 [16,27]** and revealed a network of contacts stabilizing the regulatory loops within the protease domain, whereas especially the L2 loop was stabilized *via* interdomain NOEs to β 14 as reported before ²⁵.

What has not been recorded, but what would have been much more interesting is a NOESY spectrum with the peptide. From such an experiment, and ideally from a proper structure calculation, one might derive a structure of the complex with the peptide, and build mechanistic models upon it.

We completely agree that in our initial submission this important experiment was missing. Therefore, we have now added a detailed comparison of the NOEs in the apo as well as the peptide bound state of HtrA2, which are now shown in the newly added Supplementary Figure S16. We also followed the advice of the reviewer and derived a structural model for the peptide bound complex and the isolated PDZ, which is now presented in Supplementary Figure S12. We choose to focus for this on the isolated PDZ as we had in this state only CSPs based on the interaction of the peptide in contrast to also distant effects due to the domain separation and allosteric activation.

Currently, a rather elaborate mechanistic model (Fig. 4E) is built essentially on chemical-shift changes, which are spread all over. Maybe their model is right, but I wonder where it comes from, and I do not really see how it may be built upon the observations.

We agree that the model in the initial version of the MS might have been a bit far-fetched. We have now included several new data sets supporting this model from different angles. As the initial proposal was based solely on chemical shift changes, we have now additional NOE data for the peptide-bound species, as requested by this reviewer, included and in addition also PRE

experiments included. Furthermore, we believe to have adjusted the text in a way that the reader can more easily follow the reasoning for the proposed model.

Consequently, I naturally disagree with the opening sentence of the paragraph entitled “Divalent cations are not sufficient for HtrA2 activation”, which is: “Having established the activation cascade of HtrA2 upon DD-PDZOpt binding”, and in the Discussion section the model of Figure 4E is again discussed, and I still think there that this model does not really derive from the data.

We agree that in the previous version this assessment based solely on the CSP data was too inferred and hypothetical. Nevertheless, we believe with the additional experimental data, NOEs in the peptide-bound state, PREs and gadolinium data, which all are in line with our previously suggested model, we have addressed this important and crucial issue adequately.

The discussion states that “helix $\alpha 1$ which rearranges or even detaches from the rest of the protein”, but I do not see data that support this assessment. I assume the authors refer to chemical-shift changes upon addition of peptide (data reported in Figure 4C, which are rooted in data reported in Supp Figure 9A). The only data there are that the peaks become undetectable. This is certainly interesting, and possibly it indicates something like a detachment, but it could be other things. It should, at the least, be made clear what the assessment is based on, and also make sure that a non-NMR-expert can understand that assumptions and possible other interpretations.

We agree with the reviewer that the CSP data presented in the previous version was on its own insufficient to get a clear and unambiguous insight into the behavior of helix $\alpha 1$. Therefore, we consider the PRE experiments using the MTSL-labeled HtrA2 and thus probing the flexibility of this helix as a very important addition to the current state of the MS which is now presented in the new Fig. 6, clearly in agreement with our hypothesis of increased mobility of this helix in the activated state, suggesting its detachment.

Figure 6. PRE analysis of the mobility of helix $\alpha 1$. a-c) 2D ^{13}C , ^1H -NMR of the Ile-region of MTSL-spin-labeled MALVI^{proS}-HtrA2^{S145C, S306A} in the apo-state (blue, a), in the presence of 25 mM CaCl_2 (dark blue, b), and in the presence of 520 μM DD-PDZOpt (yellow, c) in the oxidized state and after addition of 5 mM ascorbic acid in the spin label reduced state (magenta). d-f) PRE effect of a spin label on MALVI^{proS}-HtrA2^{S145C, S306A}. The PRE intensities detected for the different HtrA2 states (d-f) are indicated by the color gradient on the methyl groups depicted as spheres (PDB-ID: 5M3N). Insets illustrate the degree of mobility of the helix $\alpha 1$ under the different conditions.

Effects of metal ions:

This study has analysed how metals affect HtrA2 from multiple angles: (i) 1H-15N and methyl 1H-13C NMR chemical shift changes upon addition of metals to the PDZ domain (Supp Fig 4A-E), (ii) Changes in dynamics in PDZ (result: no changes in dynamics as seen by CPMG RD experiment; stated on page 14 and Supp Fig 4F-H), (iii) analysis of dimerization equilibrium by SEC-MALS (Supp Fig 5; Fig 3C; Supp Fig 10 C-F for full-length HtrA2), (iv) casein proteolysis experiments with the full-length HtrA2 (Fig 3E-G); as outlined in my technical criticism in a section below, I do not find the activity data convincing, (v) NMR chemical-shift data with full-length HtrA2 (Supp Fig. 10 A+B).

The conclusions about metal interaction and possible “activation” of the protein are as follows: 1) two parts of the PDZ domain (in isolation) bind metals, around residue 385-390 and around 420. (Supp Fig 4A-D). This is particularly visible for the case of Ca^{2+} . Importantly, this effect does not seem to be found for methyl groups of the isolated PDZ, which suggest that there are no rearrangements of the tertiary structure, and it is not found in methyl-detected spectra of the full-length protein. Cu^{2+} has effects, but they are presumably trivial solvent-PRE effects.

2) the dynamics of PDZ does not seem to change (CPMG data)

3) Zn, but not Ca, Cu or Mg, induces dimer formation of the PDZ domain

4) Cu and Zn lead to abolition of enzymatic activity (Fig 3B). In light of the fact that the NMR methyl signals in these two states are also very weak (Supp Fig 10B) I wonder if this may be trivially explained by aggregation of the protein.

5) NMR data of the full-length protein (Supp Fig 10) do not allow to make clear statements on whether/where metal ions bind. The methyl spectra do not show any shifts. But one may argue that even in the isolated PDZ the methyls do not show shifts. But maybe they do show intensity increase upon addition of Ca (just like for the backbone, Supp Fig 4C)? How can one explain intensity increases in the backbone (i.e. smaller R2 for the backbone), but intensity decreases (larger R2) for the methyls?

We thank the reviewer for thoroughly pointing out inconsistencies in considering the effect of the divalent ions: Therefore, we have performed additional experiments to clarify the issues raised above:

1) We have performed solvent PRE experiments using Gd^{2+} as a Ca^{2+} surrogate, which has been shown by the Florence NMR group around the late I. Bertini to be a good tool to map calcium binding sites. This data allowed and in silico prediction using the MIB2 web tool, which can predict the metal binding site. This data is now shown in the new Supplemental Figure S19:

Supplementary Figure S19. Gd^{3+} titration reveals a Ca^{2+} binding site within the PDZ domain. a) Line broadening in response to titration of 0.5 molar equivalents of Gd^{3+} towards full-length HtrA2^{S306A} (PDB-ID: 5M3N). b) Signal attenuations following titration of Gd^{3+} towards full-length HtrA2^{S306A} plotted against the HtrA2 residue number. The grey shaded regions indicate affected regions. c) Putative binding site of Ca^{2+} to the HtrA2-PDZ domain based on our experimental data and prediction of Ca^{2+}

binding sites in wild-type HtrA2 (PDB-ID: 5M3N) using MIB2 ⁶. **d)** Putative binding sites of Cu²⁺ to the HtrA2-PDZ domain based on the prediction of Cu²⁺ binding sites in wild-type HtrA2 (PDB-ID: 5M3N) using MIB2 ⁸. **e)** Structural model of the HtrA2-PDZ is (green) DD-PDZOpt peptide (yellow) complex in comparison to the residues involved in Cu²⁺ binding as identified *in silico*. This panel represent the HADDOCK model shown in **Supplementary Fig. S12h**.

- 2) *We were able to deduce in detail the mechanistic effect of calcium based on the fluorescence cleavage assay varying the activator peptide concentrations indicating that calcium modifies the affinity towards the peptide/substrate thus leading to an activation.*
- 3) *We have performed PRE experiments using MTSL-HtrA2 in the presence of calcium in indicating a more opened peptide binding cleft (Fig. 6 (see above)), in line with our interpretation of how calcium modulates the activity cascade by reducing the KD, which we presume to be due to easier access for the activator peptide, without completely disrupting the PDZ:protease domain interface.*
- 4) *Further, based on in silico prediction the copper binds to parts of the peptide binding cleft (see above Suppl. Fig. S19), which explains our observation that you can titrate out the copper with high activating peptide concentrations (Suppl. Fig. S14).*
- 5) *Regarding the zinc oligomerization. In light of the identification of the inactive hexameric state by the Kay group (Toyama et al. PNAS 2021), this state was considered as a storage state that reduces the number of enzymes in the trimeric form that can be directly activated. Even though we agree that our data does not enable us to clearly distinguish between oligomerization and potential aggregation the effect of zinc is along the same lines, it reduces the number of activable trimers.*

Overall, I have a hard time making sense of the metal-ion data. Something happens to the loop preceding alpha5 (Ca binding in PDZ alone, but maybe not in full-length?) and there is a Ca activation of full-length protein, but the link is not clear. And there is clearly no tertiary-structure rearrangement.

We hope to have clarified this important point by the additional data now and the reasoning for the previous point. In essence calcium enhances the affinity of the activator peptide, as seen by our fluorescence assay with variable activator peptide concentration, and this is supposedly achieved by a partial opening of the peptide binding cleft as seen by more extensive effects using the MTSL-HtrA2 (Fig. 6, please see above).

In the paragraph “Divalent cations are not sufficient for HtrA2 activation” it is stated that “the global signal reduction upon ZnSO₄ addition pointed to oligomerization”. I am not sure that this interpretation can be derived from the data. In Supp. Figure 10B it seems indeed that effectively all residues have strongly reduced intensity. But then a few lines below it is also said that the protein aggregates in the presence of zinc. There are always multiple reasons why a protein might aggregate, and I don’t think that one can learn much from it.

We have rewritten this part to address this aspect more carefully as follows:

Whereas in the case of CaCl₂ and MgSO₄ no **clear** indication of the domain detachment was observable, the global signal reduction upon ZnSO₄ addition pointed to **the formation of a larger HtrA2 species**. Addressing this possible oligomerization by SEC-MALS revealed a multimerization K_D ranging from 0.5 – 1 mM for the absence and presence of CaCl₂, respectively, in agreement with previous observations ²⁵ (**Supplementary Fig. S18c–f** and **Supplementary Table 5**). **The K_D could not be**

determined in the presence of ZnSO₄, but the data for the isolated PDZ domain as well as the observed strong signal attenuation in the titration of MALVI^{proS}-HtrA2^{S306A} suggest the formation of a multimeric species or even aggregates under these conditions.

The role of helix alpha5 as a “switch”:

The authors propose that helix alpha5 may act as a “switch” (stated explicitly in the Discussion), and that it is “loosely attached” (page 10).

For me it was difficult to know what is meant here. Why would helix alpha5 be a switch? A switch is something that changes when doing something to the protein (e.g. when adding ions). But it does not seem to bind

We agree with the reviewer that our idea for referring to helix $\alpha 5$ as a switch might be a bit misleading and needed further explanation. We have slightly adjusted the text when referring to this helix, particularly we now term it trigger, which we believe might be the more appropriate term. Therefore, we rephrased parts of the results section:

In contrast only a limited number of weak NOE contacts to other segments of the domain could be identified, indicating its rather loose attachment leading us to hypothesize that this helix could be an important trigger within the PDZ domain possibly involved in the activation cascade of HtrA2.

.....

This observation suggests that parts involved in the activation cascade could experience micro- to millisecond dynamics in the apo state in line with our initial hypothesis of an important trigger function of helix $\alpha 5$ within the PDZ as well as the suggested role of helix $\alpha 1$ in the activation cascade (**Fig. 5b**).

And in the discussion:

Here, we show that HtrA2 activity can additionally be modulated by divalent cations *via* binding to the PDZ domain in the proximity of helices $\alpha 5$ and $\alpha 7$, where metals such as zinc have inhibitory effects in contrast to magnesium and especially calcium leading to an enhanced HtrA2 activity (**Fig. 3 and Supplementary Figs. S13, 14**). Our data suggests that binding of metals to the PDZ domain also modulates the dynamics of helix $\alpha 5$ of the PDZ domain, an important structural element docking the PDZ on top of the protease domain in the non-activated state and potentially acting as a trigger for the domain separation.

Open technical points that need clarification:

I am not very positive about the enzymatic assays and their analysis and different points need to be addressed there:

(1) Most striking to me was Supp Fig 7E. Clearly, increasing the concentration of an activating peptide does not change anything to the activity within error bars. Nonetheless, a Michaelis-Menten type model was fitted to the data (right). The model does provide some values, which are also stated there, but I think they are meaningless. This kind of fit makes sense only if one can actually see a dependency on concentration. There also does not seem to be any correlation between the raw data of Supp Fig 7C and 7E. 7C shows a clear concentration dependence, 7E does not. To my understanding these two panels supposedly go together.

Based on the valid critical points brought up by the reviewer we have now reanalyzed the complete fluorescence data, amended it with further experiments, and hope to have resolved this critical issue in the present version.

(2) What is the initial drop in fluorescence? I assume that it has to do with equilibration. Please clarify. And what is the region of this curve that has been used for the fits?

We thank the reviewer bring this point up. Yes, we believe that the drop is due to equilibration as well as allosteric activation as we observed previously for the related bacterial HtrA protein DegP (see Sulskis et al Sci. Adv. 2021).

(3) the data shown in Supp Fig. 7D and 7F are supposed to report similar things, where 7F shows the velocities of the fluorescence traces. (What exactly? Initial slope?) When looking at 7D I find essentially no catalytic activity (rates even negative for some!) for all concentrations up to 100 μ M, and positive slopes only for the three highest concentrations. Very clearly, this is not what panel 7F shows. As said above for the fit of panel 7E, I would also be very cautious with the fit of 7F.

We have re-done our analysis of the data and for this cleaving assay we only calculated kinetic parameters for no additive, calcium and magnesium since, as the reviewer pointed out, the copper data is insufficient for accurate analysis. We have further extended our kinetic analysis with a second cleaving assay using a sub-saturating amount of activating peptide to decipher the role of the metal ions more accurately in modulation of HtrA2 activity (Supplementary figures S13, S14 and Supplementary Tables S3 and S4).

(4) The fits shown in Fig 3E are, in my opinion, equally problematic. Fitting the concentration dependence of Fig 3C (with CaCl_2) simply makes no sense, and even without addition (top panel) I would question the data. As an aside, the error bar of v_{max} (0.37 M/s) is incorrect; I assume it should read as 0.37×10^{-9} M/s.

As we have completely redone the analysis of the fluorescence data and also added a different assay varying the substrate concentration, we believe to have resolved the valid concerns of the reviewer. Please see the following largely rewritten section of the MS:

Dissecting the role of divalent ions in a coupled enzymatic assay

Thanks for spotting this obvious error in the units, which have now fixed.

To conclude, the casein data must be redone totally, in my opinion, as triplicates at least, and making clear what is done exactly.

Thanks for pointing out this important aspect, we have run all the shown and discussed assays as triplicates and we now also clearly state it in the MS:

Figure 3. Divalent metal ions modulate the proteolytic activity of HtrA2. a) Proteolytic assay using β -casein as a substrate shows that Mg^{2+} and Ca^{2+} enhanced the proteolytic activity of HtrA2, while supplementation of Zn^{2+} and Cu^{2+} strongly inhibited proteolytic activity. Cleavage assays were done as triplicates, yielding similar results (Supplementary Fig. S9).

And in the Methods section:

All variants of the fluorescence cleavage assays were run in triplicates yielding similar results.

In Supplementary Figure 3A, it is rather surprising that the MALS-estimated mass is low at the beginning of the SEC peak and then increases. Generally, the first part of a SEC peak contains the highest-molecular-weight species. This is also seen in SEC-MALS profiles in Supplementary Figure 10C. How can this be explained?

We agree with the reviewer and believe that this issue is resolved now by our reanalysis of the raw data, changing none of the previous reported results.

And consequently, how shall one understand the masses that are used for the determination of the self-association process (Supp Fig. 3B and 5).

We now clarify in the figures that the dashed lines represent the theoretical monomer and dimer masses for the PDZ or respectively the theoretical trimer and hexamer masses for the full-length protein.

Along these lines, please provide error estimates for the molecular weights in the above-mentioned graphs (Supp Fig 3B etc).

We now provide the masses including their respective errors in Supplemental Table S1 (see below) for the PDZ and S5 for the full-length protein:

Supplementary Table S1. Calculated molecular weights as well as estimated errors determined by SEC-MALS for HtrA2-PDZ.

	No additive	CaCl ₂	MgSO ₄	CuCl ₂	ZnSO ₄	DD-PDZOpt
[PDZ] □M	MW (kDa)	MW (kDa)	MW (kDa)	MW (kDa)	MW (kDa)	MW (kDa)
100	11.7 ± 0.74	11.7 ± 0.20	12.0 ± 0.17	11.9 ± 0.26	14.1 ± 0.44	n.d
150	11.7 ± 0.69	11.8 ± 0.18	11.8 ± 0.28	12.2 ± 0.34	14.3 ± 0.24	11.8 ± 0.06
250	11.8 ± 0.15	11.9 ± 0.06	12.2 ± 0.15	12.2 ± 0.29	15.2 ± 0.24	n.d
350	n.d	n.d	n.d	n.d	n.d	12.0 ± 0.07
500	12.1 ± 0.15	12.3 ± 0.07	12.7 ± 0.18	12.7 ± 0.13	17.0 ± 0.30	12.1 ± 0.06
800	12.4 ± 0.05	12.7 ± 0.05	13.3 ± 0.15	13.3 ± 0.19	18.2 ± 0.35	n.d
900	12.6 ± 0.05	n.d	n.d	n.d	n.d	n.d

The NMR spectra of proteins as large as HtrA2 are what they are, and generally have signal overlap, even when using advanced isotope-labeling schemes. It is clear from Supp Figure 8B that even with the best-resolved labeling scheme (Val-proS) there is overlap. For example, the peaks

of the following residues appear clearly overlapped: 266, 379, 325, 210, 243, 270, 441, 179, 393, and several more. This is at least the impression one gets from Supp Fig. 8B. Yet, data for all these residues are plotted in Supp Fig 9. In my view, the data of Supp Figure 9A are, thus, to be taken with caution. Could the authors please provide figures that more convincingly show how one can obtain data for these residues from 2D spectra?

We agree with the observation of the reviewer that we still have with the proS labeling a significant amount of overlapping signals. To alleviate this issue, we used for all the analyses of the methyl spectra the program PINT, which enables us to deconvolute the overlapped peaks in our view consistently and reliably. For the reviewer's assessment we add appending review only figures showing some examples from the peak fitting and deconvolution of the methyl CPMG data of full-length HtrA2 with the experimental data shown in black and the deconvoluted peaks in the different colors:

Note that Val325-gamma2 is assigned to two different peaks (!), which is physically impossible.

Thanks for spotting this issue, the labeling has been corrected.

The authors use NOESY data to propose that helix alpha-5 is somewhat “flexibly attached”. The analysis in the present state is not really convincing to me, and should be done more rigorously. Generally speaking, a lack of a NOE peak does not really unambiguously establish that two sites are far apart. The peak may be of weak amplitude, e.g. because of line broadening of the peak. The analysis does not seem to take into consideration the distances, or at least it is not shown.

We agree with the reviewer that lack of NOE is not a clear indication of secondary structure elements not stabilized by adjacent motifs. Further we believe that our initial use of the wording flexibly attached was somewhat misleading therefore we carefully rewrote this section:

Interestingly, we could observe that the α 5 helix displayed no methyl-methyl NOEs to any other structural element within the PDZ domain, indicating that this part of the PDZ domain **might** not stably stack against the other structural elements of the domain (**Supplementary Fig. S3c, f**). Having a more detailed look into the amide as well as the carbon NOEs of the α 5 helix revealed that this segment is well stabilized within itself forming a short helix perfectly in line with the observed secondary chemical shifts (**Supplementary Fig. S3d, e**). In contrast only a **limited number of** weak NOE **contacts** to other segments of the domain could be identified, indicating its **rather loose** attachment leading us to hypothesize that this helix could be an important switch **or trigger** within the PDZ domain possibly involved in the activation cascade of HtrA2.

Nevertheless, we believe that the comparison of the observed NOEs in a qualitative manner as we did here, allowed us to deduce a general trend about its stabilization with other secondary structure elements in line with our interpretation.

The secondary chemical shifts are used to propose that beta-strand 14 is populated only to “~50 %”. It is not clear to me how this estimation has been done. The analysis of CA-CB secondary chemical shifts has been popular some time ago, but I would argue that it is more precise to take into consideration as many shifts as possible and use more elaborate analyses, such as CheSPI (Mulder & co) or TALOS-N (Bax & co).

We have now added an explanation to our reasoning of a ~50% population for β -strand 14 based on the extent of the secondary chemical shifts in line with previous publications (e.g. Burmann et al. NSMB 2013; Kawale & Burmann Comm. Biol. 2020):

Comparison of the secondary structural elements of the HtrA2-PDZ domain in solution as derived from the C_{α} and C_{β} combined secondary shifts shows a large similarity to the secondary structure elements observed in the crystal structure of the HtrA2-PDZ domain and indicates overall a stably folded domain (Fig. 1d and Supplementary Fig. S2b, c)¹⁶. The main differences occur in strand β_{14} , which in solution is slightly shifted and based on the obtained secondary chemical shifts appears to be only partially populated (~50%) based on the extent of the secondary chemical shift values (Fig. 1d).

In addition, we followed the useful suggestion of the reviewer and also used TALOS-N, which we now show in Supplementary Figure S2b, c:

b) Secondary structure elements of the HtrA2-PDZ domain in solution (green) as determined by TALOS-N². The secondary structure elements of the HtrA2-PDZ domain within the full-length crystal structure (PDB-ID:1LCY) are indicated in grey. **c)** The predicted order parameter S^2 based on backbone chemical shifts as determined by TALOS-N².

Importantly, we observed that our initial analysis agrees with the TALOS-N prediction of only short segments of β 14 predicted to be a stable strand as well as being slightly shifted towards the amino-terminus in relation to the x-ray structure.

In Figure 1D I see just a few bars, some pointing to b-strand, some not, but it is not clear to me how the authors obtained any estimate of the population of b-strand from such data.

We believe the reviewer is here referring to mainly to some inconsistencies around strand β 14 as well as α 5. As written in our answer to the previous point based on the extent of the secondary shifts, we estimated the population. We also have adjusted the cartoon in this panel so that the extent of helix α 5 and the shifted strand β 14 represent the data in a clearer way.

Figure 3C (dimer population) needs error bars. I would claim that the differences in populations are negligible for all except the one with Zn. Likewise Fig 3B and 3F need error bars.

We agree that the error was missing here, but as discussed above we dropped this data and shortened the discussion about dimerization significantly. Nevertheless, we believe that we have added error bars now at all required instances.

Something is odd in supp Figure 4D, where the chemical-shift perturbation is plotted onto the structure. The second one, with Ca, is red at many residues, but when looking at the raw data (panel B) this does not seem to match. For example, I see many red spheres in helix alpha5, but the CSPs are very small. As a comparison, in other panels (e.g. the first one, with Mg) I see much larger shifts than in the second panel, e.g. for residues 385-390, yet in the structure there is not a single red sphere. It is not a very good sign if the raw data are essentially decoupled from their final representation.

We agree that the representation in the previous version seemed a bit off. Therefore, we have rechecked all of the initial data and also have now included a second “window” to make it more clear that it is the adjacent loop, and not the α 5, that has the large CSPs besides the segment preceding helix α 7 (see Suppl. Figure S6).

Further comments for clarification:

Page 17 states: “Upon CuCl₂ addition, we observed an inhibition of the activity of HtrA2, but this effect could be partially alleviated by drastically increasing the activator peptide concentration, a tendency not observed for Zn²⁺ (Supplementary Fig. 7c–f). This behavior is consistent with our SEC-MALS data, which indicates that the suppression mechanism exhibited by Zn²⁺ on the HtrA2 activity differs from the Cu²⁺ inhibitory mechanism.”

I have a hard time following their reasoning. Besides the fact that I am not convinced by any of the activity data (Supp Fig 7C-F), as outlined above (Open technical points), I do not quite follow what they mean here. Zn leads to more dimerization according to Fig S5. And how is this in line with the finding that using more activator peptide can NOT lead to activation in the presence of Zn? This needs more explanations.

We appreciate the reviewer’s comments pointing us to this important aspect, which we did not appropriately address in the initial submission. A potentially inactive hexameric HtrA2 state was previously identified and described initially by the Kay lab (Toyama et al. PNAS 2021). The authors characterize the state in this publication as follows: The thermodynamic model of Fig. 5 describing oligomerization and ligand binding suggests that the formation of hexamers, leading to a sequestration of HtrA2 trimers in an inactive, proteolytically silent state, could potentially play an additional regulatory role in suppressing uncontrolled, nonspecific cleavage of substrates in the IMS.

Based on our observations of the inhibition with zinc we reasoned that the inhibitory effect might be due to the formation of this larger oligomeric species and a suppression of the needed trimer-hexamer equilibrium. As the hexamer was reported to have with lower affinity to substrates and activating peptides this would suggest that in the presence of zinc HtrA2 is potentially trapped in the hexameric or even larger oligomeric state showing no catalytic activity. To make this reasoning also clearer we have slightly adjusted the text when discussing the SEC-MALLS as well as the cleavage assay results:

Previously, a hexameric species of HtrA2 has been characterized as a closed inactive state with reduced substrate affinity compared to the canonical trimeric species of HtrA2²⁵. Based on this previous finding and the signal attenuation observed when titrating ZnSO₄ towards HtrA2-PDZ, we reasoned that the different divalent ions might influence the oligomeric equilibrium of HtrA2.

It has previously been suggested for other HtrA proteins that Zn²⁺ may increase oligomer stabilization and Zn²⁺ ions have been found to bind to the hexameric form of *Synechocystis* HtrA homolog A^{39,50}. In light of our SEC-MALS results indicating that the isolated PDZ domain is prone to dimerize in the presence of Zn²⁺ but not any other of the tested ions as well as the binding preference of activating peptides to trimeric rather than hexameric HtrA2, this provides a possible explanation for our observations in the proteolytic assays: HtrA2 potentially forms hexamers *via* the PDZ domain in the presence of Zn²⁺, preventing efficient binding of the activating peptide and thus keeping HtrA2 in an inactive state.

The Discussion section would greatly benefit from making clear and unambiguous references to what the authors mean when they refer to their findings. E.g. it is stated that there is a “switch region encompassing helix α5, where metals such as zinc have inhibitory effects”. As stated above, it is not clear on which data this assessment is based; there are no chemical-shift changes and no changes in CPMG data in α5 with metals. If the Discussion makes this point, it must be much clearer, in my opinion.

We agree with the reviewer that the discussion needed to be more to the point and in line with the experimental data. We believe we have addressed this point by the addition of the new data as well as the large rewrite of the whole manuscript.

The authors use the term “slow timescale”. This is sometimes seen in the literature but in my view is poor practice. A time scale can be long or short (and it needs to be specified relative to what it is long or short), but a time scale cannot be “slow”.

We agree with the reviewer that it is a rather sloppy usage of terms, for which we apologize missing it in the previous version, we adjusted the text accordingly.

Please state at the beginning of section “Elucidating the allosteric regulation of HtrA2S306A using methyl-TROSY” that this sample is trimeric and corresponds to the structure shown in Figure 1B, or clarify otherwise.

We followed the suggestion of the reviewer and added the following sentence at the beginning of this section:

For expanding our characterizations also to the full-length protein, we used the catalytically inactive trimeric HtrA2^{S306A} variant to suppress any eventual self-cleavage at the high protein concentrations needed for NMR studies ²⁵.

The following publication shall be cited in the context of the activation of HtrA proteins (and their PDZ domain) by peptides M. Meltzer et al., “Allosteric activation of HtrA protease DegP by stress signals during bacterial protein quality control,” *Angew. Chemie - Int. Ed.*, vol. 47, no. 7, pp. 1332–1334, 2008, doi: 10.1002/anie.200703273.

We completely agree with the reviewer and thank him/her for pointing us to this important publication which we have now included as reference 45:

It has been previously demonstrated that HtrA2 can be activated by the interaction between its PDZ domain and target substrates, or custom-made peptides designed for optimized interaction with the HtrA2-PDZ domain, similarly to bacterial DegS and DegP ^{43–45}.

It is very difficult for the reader to evaluate statements that the data are in “perfect agreement” with previous data. This is used e.g. on page 17: “Using the fluorescence data, we calculated a k_{cat}/K_M of 6,500 M⁻¹ s⁻¹ in perfect agreement with previous reports ⁴⁶” Please be specific what this means, and avoid words like “perfect agreement”, as it is certainly not “perfect”.

We agree with the reviewer and have rephrased instances where we refer to published data and in addition directly state the literature value as shown in appending example:

Employing a two-state ligand binding model, we determined the dissociation constant K_D to $3.18 \pm 0.9 \mu\text{M}$, with a dissociation rate k_{off} of $48.5 \pm 2.9 \text{ s}^{-1}$ (**Supplementary Fig. S12g**), in good agreement with a reported K_D of $7.5 \mu\text{M}$ between wild-type HtrA2 and a related 13-residue activating peptide ⁴⁷.

Make figures more readable:

It would be extremely helpful for the reader to systematically indicate the location of secondary structure elements in plots such as Supp Fig. 4B, 4G, 4H, 6C, 6F, 9A, 10B, and Fig. 2E, and possibly there are similar instances. I found it difficult to follow the arguments put forward in the text that make reference to certain structure elements, without being able to immediately see those elements in the plots. A reader who actually cares currently constantly needs to switch between several figures to understand the naming and location of residues/secondary structures. Similarly, it would also be tremendously useful to indicate the secondary structure element names in representations of the structure, such as in Fig. 4C and Supp Fig 4D (and maybe others). Additionally, I found it difficult to read the graphical representation of Fig 4C, because the spheres are floating freely in space. For example, I tried to find the contact between alpha1 and beta7, e.g. to M260, which is supposed to be in beta7. But in the figure the sphere called “M260” floats somewhere close to the helix, and I do not see any beta-strand in the vicinity.

We thank the reviewer for these highly appreciated and useful comments. We have thoroughly gone through all figures and tried to increase their readability. Further we reassembled large parts of the supplementary information and now directly guide the reader to the correct panel.

Figure panels such as Supp Fig 10 b are almost useless, because they are too small. What exactly can I learn e.g. about the intensity drop induced by the presence of Zn? I see a general drop everywhere (presumably the protein just aggregates, according to the main text), to a bar that is barely one millimeter long.

We have rearranged the figure in question (now Suppl. Fig. S18) and believe to adequately addressed this issue making the figure more readable.

If the authors want to make a point from these data, then they should be plotted such that they are visible, AND plot the effects on the structure. If not, then they could just as well be dropped. Supp Fig 10 has another problem: the color code of panel b is not clear. I can understand that the purple plot refers to Mg, the red one to Zn (well, I guess so), but then what are the two blue ones? The order of plots is clearly changed between panels A and B, and the two blue-tones in A are not the same ones as those in B, so it is impossible to know what is Cu and what is Ca. Based on the intensity drop, I assume that the third panel in B is the fourth panel of A (Cu).

We have now fixed the arrangement and have a consistent coloring scheme throughout the MS.

Please make more axis ticks on residue-wise plots. Currently there are ticks every 50 (!) residues e.g. in Figure 1. This makes it really tedious to find out the identity of residue belonging to the different bars. Please make sure that this is improved, eg Fig 1d 1f, 2a, and possibly others.

We followed the advice of the reviewer and hope to have now increased the readability by labeling every 25th residue.

Figure 4D and Supplementary Figures 9B and 9C are not clear to me. What we are seeing are NOE contacts between methyls, shown as lines in the Flare-plots. So far so good. And then we additionally see somehow (not clear to me whether this can be seen in the Flare plots) those residues that have chemical-shift perturbation upon addition of peptide. (Is this encoded in the color? Can one read in this figure which of the two methyls that is connected by a line has a significant CSP?) What this figure implies is that the NOE network changes because a given methyl group has a chemical-shift perturbation.

We agree that these figures were too complex in the previous version without additional explanation and the data for the NOE patterns in the peptide-bound state. We have now added detailed information in Supplementary Figure S16 besides the additional NOE network analysis in the peptide bound state, hopefully resolving this important point.

(Is this right?) But this implicit conclusion is not necessarily valid. Even if the structure does not undergo significant changes that would lead to disruption of contacts between methyls, one may have CSP. Chemical shifts are just rather complicated. To me it is unclear how the NOE network in the apo state (could basically be obtained from the crystal structure) together with CSPs (which do not directly translate to structure) can tell us anything about structural changes in the complex or of “allosteric coupling of the activating loops” (caption 4d).

We believe that this section now greatly benefitted from the additional NOE analysis in the peptide bound state and the valid criticism of the reviewer is now adequately addressed in the current version of the MS.

In plots of effects on the structure (e.g. Fig 4C), only the residues with significant effects are shown but not all residues. It would be useful to see where methyl probes are which do NOT have effects.

We have followed the advice of the reviewer and now show all methyl groups etc. in these types of analyses and clearly label residues which show no effect or the ones that could not be analyzed.

Figure 2C: the axis label is wrong. Either it is an exchange rate constant (with units of inverse seconds or inverse milliseconds), or it is an exchange time constants (in seconds or milliseconds). Now it says that it is a rate constant, but the unit is ms.

The population, plotted in the “ski” plot below needs error bars. I am not sure I understand the reason why the rate constant is shown as bars, and the populations as “skis”. Similarly, why not show bar plots in all other occurrences where currently these “skis” are used?

Thanks for pointing out this miss, which has now been fixed. Further for consistency we adjusted the style of the respective panel also to “skis”.

REVIEWER COMMENTS

Reviewer #1 (Remarks to the Author):

The authors addressed fully all my comments and suggestions. The manuscript has been thoroughly revised and describes novel data that provides a clear structural analysis of the allosteric activation of the trimeric HTRA2 protease.

Reviewer #2 (Remarks to the Author):

The authors have well addressed my initial concerns. The major rewriting and addition of significant new data, resulted in a very complete manuscript. Together the results provide significant and highly interesting new insight in the mechanism of the HtrA2 protease.

The authors provide a considerable amount of data, deposited most of that and give a thorough description of methods used and their analysis in a recommended manner.

Reviewer #3 (Remarks to the Author):

The revised version has corrected several of the significant flaws of the first version, and some new experiments have been added. Therefore, it has become better, but in my opinion still not as good as it could be.

General:

Reviewing and even reading this paper is a significant effort; the paper contains lots of experimental results, some of them seem not quite unambiguous (see below), and I think that especially for the non-NMR reader this paper is difficult to digest. The NMR expert, on the other hand, will see a few dangling ends, and some data do not seem to make sense.

Thus, I am not exactly enthusiastic about the paper; but I recognize the very substantial work load that went into the paper. It is a pity that it does not tell an easily understandable story, but it reads like a collection of data which, with the aid of interpretations which may or may not be right, builds a complex story.

The figures sometimes do not help following the flow of thoughts. For example, locating elements that seem to be central in some places (e.g. the beta strands 14 and 15) is very hard at later stages (e.g. Figures 5 & 6). And is the ms dynamics of that part (detected in the isolated PDZ domain) relevant in the context of the full-length protein? I had some questions of this kind, and it took me long time to jump back and forth to make my mind on what the story is.

In any case, I have limited myself here, after a significant amount of time spent on reviewing, to point out technical points where something either does not seem quite right or whether clarification is needed. I am sure I missed some technical points due to limited time.

I would also recommend the authors to give the paper to a non-NMR colleague, and see how well it is digested, and get some inspiration from there.

Technical points:

Page 6 describes the beta strand 14 and its dynamics: "...strand beta14, which in solution is slightly shifted and based on the obtained secondary chemical shifts appears to be only partially

populated (~50%) based on the extent of the secondary chemical shift values (Fig. 1d). In addition, this segment shows increased dynamics on the milli-second timescale as this region of the protein shows broadened signals in the [15N, 1H]-NMR spectrum although side-chain carbon resonances could be assigned (Supplemental Fig. S2d).” A few sentences further down, the authors describe the millisecond dynamics in beta-strand 14 (and supposedly beta-strand 15).

There are a few things that puzzled me:

First, it is unclear to me what is meant with the side-chain carbon resonance assignment. Apparently the backbone is not visible. How have the side chains then been assigned? And which carbons of the side chain have been assigned? Figure S2, to which the authors refer, explicitly says it is about methyl assignments. Do I understand it right that only the methyls are assigned? If so then it is unclear where the C-alpha and C-beta shifts (Figure 1d) come from. At this point of the manuscript at least, I was unable to understand this. It is also rather strange that the backbone 15N is not seen, but the C_alpha and C_beta are seen. In which experiment? Needs clarification.

Moreover, what is meant when it is said that the strand is shifted? I am guessing that the authors want to say that the residues that TALOS finds as being in a beta-strand are not exactly the same as those that form a strand in the crystal structure. Please clarify.

Then there is the statement about millisecond dynamics, which is inferred from the fact that the 15N-1H sites seem to be unobservable. It is stated that beta-strand 15 also has millisecond dynamics. (“This observation is further supported by a similar exchange process within adjacent strand β 15”). This statement is not sufficiently specific. Where can I see the “similar exchange process”?

In the Model-free analysis of dynamics there is no sign of ms exchange: R_{ex} in strand 15 clearly is not higher than in the rest of the protein (in fact it seems lower than in most places). This contradicts the statement of millisecond dynamics in strand 15. Even looking at the R_2 and $R_{1\rho}$ data in Figure S4 does not indicate ms motions in beta15. Maybe the authors anticipate, at this point of the manuscript, the CPMG results? But even there, it does not seem to me that beta15 is special, compared to other parts of the structure. Maybe the data coming later (e.g. CPMG) substantiate the claims, but at this point of the manuscript the argument is not convincing.

I would also like to get more details about the Model-free analysis. Figure 1f shows that many residues have millisecond contributions to relaxation (R_{ex}). It is well established that improper modelling of the rotational-diffusion tensor can lead to artefactual R_{ex} contributions. This might be the case here. There is a section that deals with this question, but I am not sure what to conclude from the plot (Figure S4b).

It is also a bit suspicious that the residue with a very low order parameter (ca. 360) also has the highest R_{ex} . I cannot exclude this, but most often residues that undergo large-amplitude motions, as seen in the order parameter here, do not show exchange contributions (because the fast motion averages the chemical shift already). Lastly, I do not quite understand why each residue has its own τ_c , which is a global parameter; and why is there at least one residue which has a τ_c value fitted but no S^2 value fitted? (the most N-terminal one)

The CPMG experiments (Figure 2c) point to the rather intriguing result that at higher temperature the exchange process gets slower. I think that from a physical point of view this does not make sense (negative activation free energy; Arrhenius equation). There are not many ways to explain this. One is that upon changing temperature there is some fundamental change in the process itself. The other option is that the fits are not reliable. Given this non-physical situation, I do not trust the populations either (which are provided without error estimate).

Along the same lines, I would expect that the methyl CPMG data (Figure S5) reveal the same exchange process. I strongly recommend to at least report the CPMG fit results of the methyls. The fits have already been performed anyhow – solid lines in Figure S5a. If the results do not fit the backbone data, then some explanations are needed.

Page 10. In discussing the methyl order parameters and the product of order parameter and

overall correlation time ($S^2 \cdot \tau_{\text{cor}}$; Figure 2e), it would help the reader to remind the reader that τ_{cor} is ca. 10-11 ns (Figure 1). Only then one can interpret the $S^2 \cdot \tau_{\text{cor}}$ values. If we trust that value, then one could actually also show order parameters of the methyl groups, which are easier to understand.

Unfortunately, when thinking of the order parameter data of Figure 2e, I ran into another non-physical situation. The order parameters of alanine residues are expected to be very similar to order parameters of the backbone amides. This is because the methyl axis of alanine (i.e. the CA-CB axis) experiences essentially the same motion as the CA-HA or N-H. Now, looking at e.g. Ala 402, 424, 430, their methyl order parameters are very low. For example, A424 has a value of $S^2 \cdot \tau_{\text{cor}}$ of 1 ns; factoring out τ_{cor} (10-11 ns), this means that its order parameter is ca. 0.1. Now let's look what the backbone reports: Figure 1f shows that the backbone order parameter is ca. 0.9 for this part. So clearly something must be wrong here.

In the discussion of the peptide-binding effects on the overall tumbling (page 25, line ca 537; Figure S17; methyl-detected values of $S^2 \cdot \tau_{\text{cor}}$), I see a weakness: when analysing the apo and peptide-bound states, different sets of residues have been used. Because the values of $S^2 \cdot \tau_{\text{cor}}$ have large fluctuations due to different side chain dynamics, using different subsets can induce very substantial differences. The authors state that the values cannot be compared, but nevertheless state that one can conclude that the lower average value point to a release ("decoupling") of the PDZ domain. I find this not very rigorous. The authors should compare only identical residues. Moreover, they shall use the opportunity to compare the data of Figure S16 with those of Figure 2e. Figure 2e shows only the PDZ domain, and therefore τ_{cor} is different; but to first approximation one can assume that the S^2 values are similar. Hence, one can use these different data sets as a sanity check.

As written above, I have very limited trust in the data of Figure 2e. So sanity checks are really required.

Page 25: "On average we observe Delta R^2_{eff} values of 8.5 μs , with helix alpha1, parts of the regulatory loops in particular the LA-loop, and helix alpha5 showing the largest extent of conformational exchange".

The unit does not make sense. Delta R^2_{eff} cannot have a unit of microseconds, but s^{-1} .

Figure S20 definitely needs error bars in order to know whether the differences are significant. Moreover, it is difficult for me to understand the authors' conclusion that binding of the peptide leads to a release of helix alpha1. Looking at Figure 6 d,e,f looks like the difference between d and e (apo vs metal bound) is larger than between e and f (metal bound vs peptide bound). Thus, one could state that peptide binding, just like metal binding, leads to a release of the PDZ domain, but not necessarily enhanced flexibility of alpha 1.

Smaller points:

Please add ticks and tick marks to the axis of Figure 1d and 1f. It is very hard to read with precision which data point belongs to which residue.

Page 9: "The R_1 rates show " (and other places) -> more correctly it should be called a rate constant, not a rate.

Page 16: "we also ran controls". Maybe "performed" would be better than "ran".

The labelling of methyl groups is termed e.g. "ILVMproR " or "MALVIproS". The terms "proR" and "proS" make sense only for Leu and Val, but not Ile or Thr or Met or Ala. It would help non-experts to clarify what is meant exactly.

Page 3: "of which the mature protein is lacking the 133 amino-terminal residues". I think that "lacks" is the more correct form here.

Page 5: "By combining this powerful technique with sophisticated paramagnetic relaxation enhancement..." I am not sure the term "sophisticated" is required here.

The title of Figure S2 is misleading for non-NMR readers. It says "chemical shift based structural prediction". What is meant is that from the chemical-shift assignments the secondary structure is inferred, with the help of statistical analysis (TALOS). Most non-NMR readers will associate "structural prediction" with sequence-based prediction à la AlphaFold.

REVIEWER COMMENTS **our comments**

We would like to thank the reviewers for their support and careful reading of the manuscript. Their very constructive comments were extremely helpful for us to improve the manuscript. We have now revised the manuscript according to the additional points raised by reviewer #3. Below is our detailed point-by-point response to the comments of reviewer #3.

Reviewer #1 (Remarks to the Author):

The authors addressed fully all my comments and suggestions. The manuscript has been thoroughly revised and describes novel data that provides a clear structural analysis of the allosteric activation of the trimeric HTRA2 protease.

We thank the reviewer for the appreciation of our additional efforts and the general appreciation of our work.

Reviewer #2 (Remarks to the Author):

The authors have well addressed my initial concerns. The major rewriting and addition of significant new data, resulted in a very complete manuscript. Together the results provide significant and highly interesting new insight in the mechanism of the HtrA2 protease.

The authors provide a considerable amount of data, deposited most of that and give a thorough description of methods used and their analysis in a recommended manner.

We are glad for being able to remove the concerns of this reviewer and appreciate his/her support.

Reviewer #3 (Remarks to the Author):

The revised version has corrected several of the significant flaws of the first version, and some new experiments have been added. Therefore, it has become better, but in my opinion still not as good as it could be.

General:

Reviewing and even reading this paper is a significant effort; the paper contains lots of experimental results, some of them seem not quite unambiguous (see below), and I think that especially for the non-NMR reader this paper is difficult to digest. The NMR expert, on the other hand, will see a few dangling ends, and some data do not seem to make sense.

Thus, I am not exactly enthusiastic about the paper; but I recognize the very substantial work load that went into the paper. It is a pity that it does not tell an easily understandable story, but it reads like a collection of data which, with the aid of interpretations which may or may not be right, builds a complex story.

We thank the reviewer for the appreciation of our efforts to improve the manuscript.

The figures sometimes do not help following the flow of thoughts. For example, locating elements that seem to be central in some places (e.g. the beta strands 14 and 15) is very hard at later stages (e.g. Figures 5 & 6). And is the ms dynamics of that part (detected in the isolated PDZ domain) relevant in the context of the full-length protein? I had some questions of this kind, and it took me long time to jump back and forth to make my mind on what the story is.

We thank the reviewer for pointing out this point. Therefore, we included labeling of these elements also in the later figures as well. As for the raised question regarding the ms dynamics we believe that the identified trigger function helix a5 is an important in the activation cascade.

In any case, I have limited myself here, after a significant amount of time spent on reviewing, to point

out technical points where something either does not seem quite right or whether clarification is needed. I am sure I missed some technical points due to limited time.

We highly appreciate the time this reviewer spent on our manuscript and the highly constructive nature of his/her valid criticism.

I would also recommend the authors to give the paper to a non-NMR colleague, and see how well it is digested, and get some inspiration from there.

We followed the advice of the reviewer and handed the MS to two of our colleagues with crystallography and biochemistry background, respectively, and rephrased a few instances.

Technical points:

Page 6 describes the beta strand 14 and its dynamics: "...strand beta14, which in solution is slightly shifted and based on the obtained secondary chemical shifts appears to be only partially populated (~50%) based on the extent of the secondary chemical shift values (Fig. 1d). In addition, this segment shows increased dynamics on the milli-second timescale as this region of the protein shows broadened signals in the [15N, 1H]-NMR spectrum although side-chain carbon resonances could be assigned (Supplemental Fig. S2d)." A few sentences further down, the authors describe the millisecond dynamics in beta-strand 14 (and supposedly beta-strand 15).

There are a few things that puzzled me:

First, it is unclear to me what is meant with the side-chain carbon resonance assignment. Apparently the backbone is not visible. How have the side chains then been assigned? And which carbons of the side chain have been assigned? Figure S2, to which the authors refer, explicitly says it is about methyl assignments. Do I understand it right that only the methyls are assigned? If so then it unclear where the C-alpha and C-beta shifts (Figure 1d) come from. At this point of the manuscript at least, I was unable to understand this. It is also rather strange that the backbone 15N is not seen, but the C_alpha and C_beta are seen. In which experiment? Needs clarification.

We apologize for being unclear here. As a matter of fact, we do not observe NH signals in the [15N, 1H]-NMR spectrum, but we could observe the whole carbon shifts, albeit weakly, in the [13C, 1H]-HSQC spectrum enabling us to assign them based on sidechain TOCSY type 3D spectra as well as NOESY data and thus giving access to the C α and C β shifts. We have rephrased and hope to have now clarified this section:

In addition, this segment shows broadened signals in the [15N, 1H]-NMR spectrum although the non-solvent exchangeable side-chain carbon resonances could be completely assigned (Supplemental Fig. S2d). This behavior could point to either elevated solvent exchange under the used conditions and/or dynamics on the milli-second timescale for this region of the protein, indicative of either structural heterogeneity and/or exchange processes on the micro- to millisecond NMR timescale

31

Moreover, what is meant when it is said that the strand is shifted? I am guessing that the authors want to say that the residues that TALOS finds as being in a b-strand are not exactly the same as those that form a strand in the crystal structure. Please clarify.

Yes, the reviewer is right it was our intention to state that the secondary chemical shifts regardless of the two types of analyses indicate that in solution the β -strand does not exactly overlap with the one reported in the crystal structure. We have adjusted the text to make this point clearer:

The main differences occur in strand β_{14} , which in solution, indicated by the secondary chemical shift analysis (Fig. 1d and Supplementary Fig. S2b), is slightly shifted, and

based on the extent of the obtained secondary chemical shift values appears to be only partially populated (~50%) (Fig. 1d).

Then there is the statement about millisecond dynamics, which is inferred from the fact that the ^{15}N - ^1H sites seem to be unobservable. It is stated that beta-strand 15 also has millisecond dynamics. ("This observation is further supported by a similar exchange process within adjacent strand β_{15} "). This statement is not sufficiently specific. Where can I see the "similar exchange process"?

We thank the reviewer for pointing out this miss. We now have adjusted the text using and provide Supplementary Figure S2 showing ^{15}N , ^1H -NMR spectrum signal intensities:

This observation is further supported by diminished signal intensity in the ^{15}N , ^1H -NMR spectrum for residues within adjacent strand β_{15} (Supplemental Fig. S2e), pointing to a similar process and suggesting that the interaction between these two strands is only transiently formed.

In the Model-free analysis of dynamics there is no sign of ms exchange: R_{ex} in strand 15 clearly is not higher than in the rest of the protein (in fact it seems lower than in most places). This contradicts the statement of millisecond dynamics in strand 15. Even looking at the R_2 and $R_{1\rho}$ data in Figure S4 does not indicate ms motions in beta15. Maybe the authors anticipate, at this point of the manuscript, the CPMG results? But even there, it does not seem to me that beta15 is special, compared to other parts of the structure. Maybe the data coming later (e.g. CPMG) substantiate the claims, but at this point of the manuscript the argument is not convincing.

We adjusted the text as indicated above and hope thus to have resolved the raised issue.

I would also like to get more details about the Model-free analysis. Figure 1f shows that many residues have millisecond contributions to relaxation (R_{ex}). It is well established that improper modelling of the rotational-diffusion tensor can lead to artefactual R_{ex} contributions. This might be the case here. There is a section that deals with this question, but I am not sure what to conclude from the plot (Figure S4b).

We thank the reviewer to pointing us to some inconsistencies here. Based on the points raised by the reviewer we have carefully checked our relaxation analysis and believe that we have now removed any remaining issues. We think that the tensor was accurately modelled (see below). In addition, we now provide in the supplementary additional details of the analysis of the backbone relaxation data in Supplementary Figure S1c, d:

c) Axially symmetric rotational diffusion tensor obtained by the backbone relaxation analysis using Tensor2 [3] employing the HtrA2-PDZ X-ray structure (PDB-ID: 1LCY). The table provides the determined diffusion tensor values in 10^{-7} s^{-1} . **d)** Residues assigned in the backbone relaxation analysis to the different dynamics model in the Model Free analysis [4,5]. The initial models (models 1–5) were described by Lipari and Szabo [4,5] as well as subsequently extended by Clore and co-workers (model 6) [5].

*Regarding the question about Figure S2b, we used the $R_1 * R_{2\beta}$ values to rule out if there is a large anisotropic and potentially incorrect causing elevated $R_{2\beta}$ rates instead of increase micro to millisecond dynamics as described previously (e.g. Kroenke et al. JACS 1998, Kneller and Bracken JACS 2002, Frey et al. Angewandte 2017). In the case of the HtrA2-PDZ domain these data show the same trend thus providing confidence in the used diffusion tensor (see also above) and point to the existence of micro- to millisecond dynamics as discussed in the main text.*

It is also a bit suspicious that the residue with a very low order parameter (ca. 360) also has the highest R_{ex} . I cannot exclude this, but most often residues that undergo large-amplitude motions, as seen in the order parameter here, do not show exchange contributions (because the fast motion averages the chemical shift already).

As outlined in the previous point we have based on the suggestions of the reviewer carefully checked the relaxation analysis and thus hope to have resolved these issues as stated in detail in the methods section:

Residues with an $R_{2(R1\rho)}/R_1$ rate quotient above 30 were removed from the analysis with Tensor2 and a prolate axially symmetric diffusion tensor was used to fit the relaxation data to the HtrA2-PDZ structure (PDB-ID: 1LCY).

Lastly, I do not quite understand why each residue has its own τ_c , which is a global parameter; and why is there at least one residue which has a τ_c value fitted but no S2 value fitted? (the most N-terminal one)

The reviewer is perfectly correct that the τ_c is a global parameter, we nevertheless choose to represent it in the chosen manner as we determined it from the ration of R_2 and $R_{2(R1\rho)}$. Thus, the average value is the value we report. We believe that the second part of the reviewers question about the residue with

an τ_c and no corresponding S^2 value has been resolved by the reanalysis of the data as outlined under the previous points.

The CPMG experiments (Figure 2c) point to the rather intriguing result that at higher temperature the exchange process gets slower. I think that from a physical point of view this does not make sense (negative activation free energy; Arrhenius equation). There are not many ways to explain this. One is that upon changing temperature there is some fundamental change in the process itself. The other option is that the fits are not reliable. Given this non-physical situation, I do not trust the populations either (which are provided without error estimate).

We must apologize for this mistake, but when reanalyzing and checking the whole backbone relaxation data we realized that the panels for 298 and 283 K were unintentionally swapped, thus the data shows only a slight temperature-dependent modulation which now also makes sense from a physical point of view. Furthermore, we now provide the requested error bars in the Figure 2c.

Along the same lines, I would expect that the methyl CPMG data (Figure S5) reveal the same exchange process. I strongly recommend to at least report the CPMG fit results of the methyls. The fits have already been performed anyhow – solid lines in Figure S5a. If the results do not fit the backbone data, then some explanations are needed.

We agree with the reviewer that we should have addressed this important point. We provide the fits of the CPMG data on Mendeley and have slightly altered the text to clarify that we believe to observe the same process in both methyl and backbone datasets:

As we had already obtained indications of specific line-broadening in a 2D [$^{13}\text{C}, ^1\text{H}$]-NMR spectrum for some methyl resonances, e.g. I373 and V364 (**Supplementary Fig. S2a**), we used a multiple quantum (MQ) CPMG relaxation dispersion experiment³⁷. The obtained CPMG relaxation dispersion profiles report on the micro- to millisecond dynamics and showed increased rate constants for residues in the $\alpha 5$ to $\beta 15$ region as well as for residues in the $\beta 16/\alpha 7$ with a $16.6 \pm 0.4\%$ minor state population on a 1.19 ± 0.07 ms time scale (**Fig. 2f** and **Supplementary Fig. S5a, b**). The obtained data for both, the protein amide backbone as well as the methyl groups, indicate that the PDZ domain is experiencing micro- to millisecond dynamics, indicative of the same underlying process. These dynamics are manifested within the $\alpha 5$ helix and the adjacent linker region connecting $\alpha 5$ and $\beta 15$, pointing to a possible importance of these inherent local dynamics for the regulation of HtrA2 activation and its protease activity.

Page 10. In discussing the methyl order parameters and the product of order parameter and overall correlation time ($S^2 \cdot \tau_{\text{auc}}$; Figure 2e), it would help the reader to remind the reader that τ_{auc} is ca. 10-11 ns (Figure 1). Only then one can interpret the $S^2 \cdot \tau_{\text{auc}}$ values. If we trust that value, then one could actually also show order parameters of the methyl groups, which are easier to understand. Unfortunately, when thinking of the order parameter data of Figure 2e, I ran into another non-physical situation. The order parameters of alanine residues are expected to be very similar to order parameters of the backbone amides. This is because the methyl axis of alanine (i.e. the CA-CB axis) experiences essentially the same motion as the CA-HA or N-H. Now, looking at e.g. Ala 402, 424, 430, their methyl order parameters are very low. For example, A424 has a value of $S^2 \cdot \tau_{\text{auc}}$ of 1 ns; factoring out τ_{auc} (10-11 ns), this means that its order parameter is ca. 0.1. Now let's look what the backbone reports: Figure 1f shows that the backbone order parameter is ca. 0.9 for this part. So clearly something must be wrong here.

We are grateful to the reviewer to pointing this crucial issue out as he/she is perfectly correct that there was something off. We rechecked our data and realized that while assembling the bar plots shown in Figure 2e the panels were unintendedly swapped (top panel was actually the bottom panel) indicating

that something is off. In the now correct bar plot the mentioned alanine residues show the correct values close to the maximum $S^2_{axis} \cdot \tau_C$ values obtained above ~ 6.5 ns.

In the discussion of the peptide-binding effects on the overall tumbling (page 25, line ca 537; Figure S17; methyl-detected values of $S^2_{axis} \cdot \tau_C$), I see a weakness: when analysing the apo and peptide-bound states, different sets of residues have been used. Because the values of $S^2_{axis} \cdot \tau_C$ have large fluctuations due to different side chain dynamics, using different subsets can induce very substantial differences. The authors state that the values cannot be compared, but nevertheless state that one can conclude that the lower average value point to a release (“decoupling”) of the PDZ domain. I find this not very rigorous. The authors should compare only identical residues. Moreover, they shall use the opportunity to compare the data of Figure S16 with those of Figure 2e. Figure 2e shows only the PDZ domain, and therefore τ_C is different; but to first approximation one can assume that the S^2 values are similar. Hence, one can use these different data sets as a sanity check. As written above, I have very limited trust in the data of Figure 2e. So sanity checks are really required.

We completely agree that we should have done such sanity checks and apologize for this miss. Based on the correction of Figure 2e as outlined above we mainly focused on the comparison between apo and peptide bound state and rephrased this section:

In the apo state we obtained values of 38 ± 19 ns and 37 ± 24 ns for the protease and PDZ domain (Supplementary Figure S17a, c), respectively, in good agreement with a previous study²⁵ showing that the domains are tightly coupled in the apo state. Analyzing HtrA2^{S306A} in the presence of the DD-PDZOpt peptide yielded decreased values of 29 ± 20 ns and 24 ± 21 ns for the protease and PDZ domain, respectively (Supplementary Fig. S17b, d). It must be noted that due to line-broadening and signal overlap fewer resonances could be analyzed compared to the apo state thus the difference is slightly less pronounced if the same residues are compared: 34 ± 13 ns and 35 ± 19 ns for protease domain and PDZ in the apo-state in comparison to 29 ± 20 ns and 24 ± 21 ns in the peptide bound state (Supplementary Fig. S17d). Nevertheless, the obtained values are in line with the decoupling of the PDZ-domain as evidenced by the on average lower values for this domain. Although these values indicate an enhanced flexibility of this domain, values around 6.5 ns, as obtained for the isolated PDZ domain (Fig. 2d, e), would be expected for a completely decoupled movement, pointing to a still restricted mobility after detachment of the PDZ.

We also adjusted Supplementary Fig. 17d to directly show the comparison between apo and holo state:

d) $S^2_{axis} \cdot \tau_C$ -values of peptide bound HtrA2^{S306A} plotted against the HtrA2 amino acid sequence. Secondary structure elements are displayed on the top. Average values and the respective standard deviations for the protease domain and the PDZ domain are indicated. For comparison, values for corresponding residues in the apo state and the average values are indicated in faint blue and grey, respectively.

Page 25: “On average we observe ΔR_{2eff} values of 8.5 μ s, with helix α_1 , parts of the regulatory loops in particular the LA-loop, and helix α_5 showing the largest extent of conformational

exchange”.

The unit does not make sense. Delta R2eff cannot have a unit of microseconds, but s⁻¹.

The reviewer is perfectly correct, and we have now fixed this error.

Figure S20 definitely needs error bars in order to know whether the differences are significant. Moreover, it is difficult for me to understand the authors' conclusion that binding of the peptide leads to a release of helix alpha1. Looking at Figure 6 d,e,f looks like the difference between d and e (apo vs metal bound) is larger than between e and f (metal bound vs peptide bound). Thus, one could state that peptide binding, just like metal binding, leads to a release of the PDZ domain, but not necessarily enhanced flexibility of alpha 1.

We have based on the suggestion of the reviewer added error bars and show the intensity of the apo state in the other states panel to provide are more intuitive connection:

Supplementary Figure S20. Paramagnetic relaxation enhancement (PRE) of a spin label attached to HtrA2 at position S145C. a–c) PRE data for HtrA2^{S306A, S145C} in buffer (blue) compared to HtrA2^{S306A, S145C} in the presence of CaCl₂ (dark blue) and HtrA2^{S306A, S145C} in the presence of DD-PDZOpt (yellow). The PRE is shown as the ratio V_{ox}/V_{red} , where V_{ox} is the cross-peak intensity before reduction of the spin label and V_{red} is the peak intensity after reduction of the spin label. Secondary structural elements and regulatory loops are indicated. The dashed line indicates where the V_{ox}/V_{red} ratio equals 1. Errors were estimated based on the spectral noise. Secondary structure elements and regulatory loops are indicated on top of each panel. For reference the apo data is additionally shown in faint blue also in panels b and c.

Smaller points:

Please add ticks and tick marks to the axis of Figure 1d and 1f. It is very hard to read with precision which data point belongs to which residue.

We apologize for missing to adjust the labelling of Figure 1f in the previous round of revisions. It has been adjusted to tick marks every 25th residue, to match the tick mark spacing on the other figures throughout the whole manuscript. Figure 1d already has this interval, so we did not adjust it further.

Page 9: "The R1 rates show " (and other places) -> more correctly it should be called a rate constant, not a rate.

We adjusted the wording throughout the MS according to the reviewer's suggestions.

Page 16: "we also ran controls". Maybe "performed" would be better than "ran".
"Ran" has been substituted with "performed".

The labelling of methyl groups is termed e.g. "ILVMproR" or "MALVIproS". The terms "proR" and "proS" make sense only for Leu and Val, but not Ile or Thr or Met or Ala. It would help non-experts to clarify what is meant exactly.

We followed the suggestion of the reviewer and explained the proS and proR meaning at their respective first occurrences.

By using previously reported assignments for ILVM^{proR} (methyl group labeling of Ile- δ_1 , Leu- δ_1 (proR), Val- γ_1 (proR) and Met- ϵ) HtrA2^{S306A} 25,30 as well as our PDZ side-chain assignment, we were able to complete the sequence specific assignment to ~92% of the methyl groups of a MALVI^{proS} (methyl group labeling of Met- ϵ , Ala- β , Leu- δ_2 (proS), Val- γ_2 (proS) and Ile- δ_1) labeled HtrA2^{S306A} (**Supplementary Fig. S15**).

Page 3: "of which the mature protein is lacking the 133 amino-terminal residues". I think that "lacks" is the more correct form here.

This grammar error has been corrected.

Page 5: "By combining this powerful technique with sophisticated paramagnetic relaxation enhancement..." I am not sure the term "sophisticated" is required here.

We followed the suggestion of the reviewer and removed "sophisticated".

The title of Figure S2 is misleading for non-NMR readers. It says "chemical shift based structural prediction". What is meant is that from the chemical-shift assignments the secondary structure is inferred, with the help of statistical analysis (TALOS). Most non-NMR readers will associate "structural prediction" with sequence-based prediction à la AlphaFold.

We agree with the reviewer, therefore the title has been edited to "Methyl side chain assignment and chemical shift-based prediction of secondary structure elements" to hopefully be clearer.

REVIEWERS' COMMENTS

Reviewer #3 (Remarks to the Author):

I thank the authors for their patience to chase the last details or swapped data, and congratulate them for this study, which contains a great amount of data.
I do not have further concerns, and recommend publication of this nice study.

REVIEWER COMMENTS **our comments**

Reviewer #3 (Remarks to the Author):

I thank the authors for their patience to chase the last details or swapped data, and congratulate them for this study, which contains a great amount of data.

I do not have further concerns, and recommend publication of this nice study.

We would like to thank the reviewer for his persistence helping us to greatly improve our manuscript. We are glad that we could remove all remaining concerns of this reviewer and highly appreciate his/her kind words and support.